# Co-Homology of Differential Forms and Feynman Diagrams



**Sergio Luigi Cacciatori** [1,2,*], **Maria Conti** [1,2] **and Simone Trevisan** [1,2]

1 Dipartimento di Scienza ed Alta Tecnologia, Università degli Studi dell'Insubria, 22100 Como, Italy;
mconti@uninsubria.it (M.C.); strevisan@uninsubria.it (S.T.)
2 INFN, Sezione di Milano, Via Celoria 16, 20133 Milan, Italy
* Correspondence: sergio.cacciatori@uninsubria.it

**Abstract:** In the present review we provide an extensive analysis of the intertwinement between Feynman integrals and cohomology theories in light of recent developments. Feynman integrals enter in several perturbative methods for solving non-linear PDE, starting from Quantum Field Theories and including General Relativity and Condensed Matter Physics. Precision calculations involve several loop integrals and an onec strategy to address, which is to bring them back in terms of linear combinations of a complete set of integrals (the master integrals). In this sense Feynman integrals can be thought as defining a sort of vector space to be decomposed in term of a basis. Such a task may be simpler if the vector space is endowed with a scalar product. Recently, it has been discovered that, if these spaces are interpreted in terms of twisted cohomology, the role of a scalar product is played by intersection products. The present review is meant to provide the mathematical tools, usually familiar to mathematicians but often not in the standard baggage of physicists, such as singular, simplicial and intersection (co)homologies, and hodge structures, that are apt to restate this strategy on precise mathematical grounds. It is intended to be both an introduction for beginners interested in the topic, as well as a general reference providing helpful tools for tackling the several still-open problems.

**Keywords:** Feynman integrals; twisted cohomology; intersection theory

## 1. Introduction

Feynman diagrams are introduced in the context of quantum interacting field theory, as a graphical representation of the solution of a system of first order differential equations, admitting a path-ordered exponential expression. Usually, the matrix of the system is composed of two terms: one identifying the solution in the absence of interactions, i.e., the free solution, and a second one, carrying information on the interaction, treated as a perturbation to the free evolution, and characterized by the strength of the interaction, i.e., the coupling constant, considered as a small quantity. The perturbative expansion of the path-ordered exponential, obtained by a series expansion in the coupling constant, gives rise to the Dyson series, containing an infinite sequence of iterated integrals, whose iteration number increases with the perturbative order: at any given order, and hence for any given power of the coupling constant, the integrands are formed by an ordered product of functions-to better say, distributions-, representing the free evolution (the propagators), and the insertion of interaction terms (the vertices).

Dyson series can be used to describe the evolution of physical systems, whose dynamics follow the Volterra-type model, within quantum as well as classical physics. Therefore, the predictive power of a theoretical model aiming to describe the dynamics of physical systems on a wide spectrum of physical scales, from microscopic such as colliding elementary particles to macroscopic such as coalescing astrophysical binary systems, may depend on our ability to evaluate Feynman integrals, also known as solving systems of differential equations.

Hamiltonian and Lagrangian carry information about the free and the interactive dynamics, and the basic rules to build Feynman graphs can be systematically derived from them. In particular, the interaction between two elementary entities experiencing the *presence* of each other through the mediation of a third entity can be described like a scattering event. Therefore, quantities such as the impact parameter, the cross section, the scattering angle, or the interaction potential turn out to be related to the scattering amplitude, which ultimately admits a representation in terms of Feynman graphs.

"Perturbation theory means Feynman diagrams" [1], yet the diagrammatic approach is not limited to the perturbative regime: "Perturbation theory is a very useful device to discover very useful equations and properties that may hold true even if the perturbation expansion fails" [2].

More modern approaches based on analyticity and unitarity, so called on-shell and unitarity-based methods [3–6], make use of the factorization properties of scattering amplitudes (exposed by using complex variables to build suitable combinations of energy and momenta of the interacting objects) in order to more efficiently group the contributing Feynman diagrams and exploit recursive patterns, hard to identify within the pure diagrammatic approach. In this case, the symmetries, which do not necessarily hold for the individual diagrams, and which are inherited from the lower-order amplitudes, yield novel representations of the scattering amplitudes (see f.i. [7,8]).

Therefore, scattering amplitudes, at any given order in perturbation theory, can be canonically built out of linear combination of Feynman graphs and equivalently out of products (convolutions) of lower order amplitudes. Independently of the strategy adopted for their generation, the evaluation of scattering amplitudes beyond the tree-level approximation requires the evaluation of multivariate Feynman integrals.

Dimensional regularization played a crucial role in the formal mathematical developments of gauge theories and of Feynman integrals. Exploiting the analytic continuation in the space-time dimensions $d$ of the interacting fields, it is possible to modify the number of integration variables in order to *stabilize* otherwise ill-defined (mathematically non-existing) integrals emerging in the evaluation of quantities that ultimately have to be compared with numbers coming from (physically existing) experiments.

Within the dimensional regularization scheme, Feynman integrals are not independent functions. They obey relations that can be established at the integrand level, namely among the integrands related to different graphs, systematized in the so-called integrand decomposition method for scattering amplitudes [9–16], as well as relation, that hold, instead, just upon integration. The latter are contiguity relations known as integration-by-parts (IBP) identities [17], which play a crucial role in the evaluation of scattering amplitudes beyond the tree-level approximation. Process by process, IBP identities yield the identification of an elementary set of integrals, the so-called *master integrals* (MIs), which can be used as a basis for the decomposition of multi-loop amplitudes [18]. MIs are special integrals, namely elementary Feynman integrals that admit a graphical representation (in terms of products of scalar propagators and scalar interaction vertices). At the same time, IBP relations can be used to derive differential equations [19–26], finite difference equations [27,28], and dimensional recurrence relations [29,30] obeyed by MIs. The solutions of those equations are valuable methods for the evaluation of MIs for those cases where their direct integration might turn out to be prohibitive (see, e.g., [31–33]).

The study of Feynman integrals, the systems of differential equations they obey, and the iterated integral representation of their solution [34–36] have been stimulating a vivid interplay and renovated interest between field theoretical concepts and formal mathematical ideas in Combinatorics, Number Theory, Differential and Algebraic Geometry, and Topology (see, e.g., [37–49]).

The geometric origin of the analytic properties of Feynman integrals finds its roots in the application of topology to the S-matrix theory [50–52]. In more recent studies, cohomology played an important role for identifying relations among Feynman integrals and to expose deeper properties of scattering amplitudes [39,41,45,46,53–73]

In this editorial, we elaborate on the recently understood vector space structure of Feynman integrals [59–66] and the role played by the intersection theory for twisted de Rham (co)-homology to understand it.

As observed in Reference [59], after looking at Feynman integrals as generating a vector space, one can see *intersection numbers* of differential forms [74] as a sort of scalar product over it. From this viewpoint, the intersection products with a basis of MIs mimic the projection of a vector into a basis. For example, using intersection projections for 1-forms applied to integral representations of the Lauricella $F_D$ functions allowed one to easily re-derive continuity relations for such functions and the decomposition in terms of MIs for those Feynman integrals on maximal cuts that admit a representation as a one-fold integral [59,61]. For more general cases, when one has to deal with multifold integral representations [61,63,65], the *multivariate* intersection numbers have been introduced [46,75–82]. For the case of meromorphic $n$-forms, an iterative method for the determination of intersection numbers was proposed in [60] and successively refined in [63,65,66]. The only simple case is for logarithmic (dlog) differential forms, which bring simple poles only, whose intersection numbers can be computed by employing the global residue theorem [66].

Within this approach, the number of MIs, proven to be finite [83], is the dimension of the vector space of Feynman integrals [63], and corresponds to the dimension of the homology groups [53], or equivalently of the cohomology group [59,61,63,65], and can be related to topological quantities such as the number of critical points [53], Euler characteristics [84–87], as well as to the dimension of quotient rings of polynomials, for zero dimensional ideals, in the context of computational algebraic geometry [65].

Another interesting consequence of intersection numbers is about their underlying geometrical meaning, which leads to determining linear relations, equivalent to IBP relations, and quadratic relations for Feynman integrals [59,61,63,65], called *twisted Riemann periods relations* (TRPR) [74] since they represent a twisted version of the well-known bilinear Riemann relations. Some of such quadratic relations were already noticed by using number-theoretic methods to Feynman calculus and have given rise to conjectures [54–56,88,89], whose proof has been given only recently [70,71], while other bilinear relations, proposed in [90], have yet to be understood in the light of the TRPR.

As stated above, in this work, we first concentrate on the basic aspects leading to the definition of a vector space of Feynman integrals; then we move onto the mathematical description ( starting from an elementary point of view) of the instruments needed in order to tackle the geometrization program just described. Finally, we devote some time to addressing the problem of the actual computation of intersection numbers. More precisely, in Section 2, we mostly describe the Baikov representation of Feynman integrals and its role in uncovering the underlying cohomological structure, while in Section 3, we consider the vector structure of Feynman integrals—including bilinear identities—and also precisely describe how the number of MIs can be computed. In Section 4, we provide an elementary illustration of cohomologies, while in Section 5, we highlight the link between cohomology theory and integration theory. In Section 6, we give an extensive lookout at the advanced mathematical constructions behind. Finally, in Section 7 we make some explicit examples of practical techniques adopted to compute intersection numbers. The three appendices include some technical details.

## 2. Feynman Integral Representation

We aim at describing the properties of (scalar) Feynman integrals, representing the most general type of integrals appearing in the evaluation of Scattering Amplitudes, left over after carrying out the spinor and the Lorentz algebra (spinor-helicity decomposition, Dirac–Clifford gamma algebra, form factor decomposition), generically indicated as

$$I_{\nu_1,\cdots,\nu_N} = \int \prod_{i=1}^{L} \frac{d^D q_i}{\pi^{D/2}} \prod_{a=1}^{N} \frac{1}{D_a^{\nu_a}}. \tag{1}$$

In the classical literature, the evaluation of Feynman integrals is carried out by direct integration, in position and/or momentum-space representation, making use of Feynman, or equivalently Schwinger, parameters. More advanced methods make use of differential equations as an alternative computational strategy, which turns out to be very useful whenever the direct integration becomes prohibitive, for instance, due to the number of the physical scales in the scattering reaction (number of particles and/or masses of particles). In this work, we would like to approach the multi-loop Feynman calculus in a different fashion from the direct integration, making use of novel properties that emerge when Feynman integrals are cast in suitable parametric representation, such as the so called *Baikov representation* [91] (see also [92] for review). First, we observe that the integration variables involved in the integral (1) are the usual $L$ loop momenta $q_i$, which are not Lorentz invariants. Baikov representation consists in a change of variables in which the new integration variables are actually Lorentz invariants: that is, the independent scalar products one can build, using the $L$ loop momenta $q_i$ and $E$, independent external momenta $p_j$. Using these ideas, one can put the Feynman Integrals in the following form, called Baikov representation:

$$I_{\nu_1,\cdots,\nu_M} = K \int_\Gamma B^\gamma \prod_{a=1}^M \frac{dz_a}{z_a^{\nu_a}}. \tag{2}$$

For a proof, see Appendix A.

Before getting more into depth of the meaning behind Equation (2), an observation is necessary. By comparing the original integral with Equation (2), one observes that the number of integration variables changes from $LD$ to $M$. When we perform the projection (A4) of each 4-momentum onto the space generated by the vectors coming next, it is actually clear that this process cannot continue indefinitely, as all the vectors are certainly not independent if they lie in the physical 4D space. The decomposition we describe in Equation (A4) is to be thought of in an abstract (sufficiently large) dimension $D$. Since the final expression is an analytic function of $D$, we get the physically meaningful result via an analytic continuation down to $D = 4$. This discussion can be also summarized by saying that we are implicitly using dimensional regularization to make sense of the expression (1), which is obviously divergent in $D = 4$.

The representation in Equation (2) highlights new properties of the original integral (1) and allows us to study its topological structure as Aomoto-Gel'fand integral [59]. In fact, extending the integral in Equation (2) into the complex space, it takes the form

$$I = K \int_C u(\vec{z})\phi(\vec{z}), \tag{3}$$

where $K$ is constant prefactor, $u = B^\gamma$ is a multivalued function such that $u(\partial C) = 0$ and $\phi$ is an $M$-form

$$\phi \equiv \hat{\phi} d^M z = \frac{dz_1 \wedge \cdots \wedge dz_M}{z_1^{\nu_1} \cdots z_M^{\nu_M}}. \tag{4}$$

Because of the Stokes theorem, given a certain $(M-1)$-form $\xi$ the following identity holds:

$$\int_C d(u\xi) = \int_{\partial C} u\xi = 0, \tag{5}$$

as $u\xi$ is integrated along $\partial C$ where $u$ vanishes. Equation (5) can also be rewritten as

$$\int_C d(u\xi) = \int_C (du \wedge \xi + ud\xi) = \int_C u(\underbrace{d\log u}_{\omega} \wedge + d)\xi \equiv \int_C u\nabla_\omega \xi = 0. \tag{6}$$

Equation (6) states that, because of the introduction of the connection $\nabla_\omega = d + \omega\wedge$ where $\omega \equiv d\log u$, it holds that

$$\int_C u\phi = \int_C u(\phi + \nabla_\omega \xi), \tag{7}$$

as the second term in the right side gives a null contribution. Equation (7) identifies an equivalence class, addressed as

$$_\omega\langle\phi| := \quad \phi \sim \phi + \nabla_\omega\xi. \tag{8}$$

This equivalence class defines *twisted cohomologies* (twisted because the derivative involved is not simply $d$ as in the de Rham cohomology but it is the covariant derivative $\nabla_\omega$). Representatives of a class are called *twisted cocycles*.

In this fashion, the integral $I$,

$$I = \int_C u\phi \tag{9}$$

arises as a *pairing* between the twisted cycle $|C]$ and the twisted cocycle $\langle\phi|$. For ease of notation, the subscript $\omega$ is understood and restored when needed.

Aomoto-Gel'fand integrals admit linear and quadratic relations that can be used to simplify the evaluation of scattering amplitudes. In particular, linear relations can be exploited to express any integral as a linear combination of an independent set of functions, called *master integrals* (MIs). The decomposition of Feynman integrals in terms of MIs was proposed in [17] and later systematized in [18], and represents the most common computational technique for addressing multi-loop calculus nowadays. The novel insights we elaborate on in this work allow us to explore the underpinning vector space structure obeyed by Aomoto-Gel'fand integrals in order to investigate the properties of Feynman integrals making use of co-homological techniques. Accordingly, the decomposition of any generic integral $I = \int_C u\phi \equiv \langle\phi|C]$ in terms of a basis of MIs, say $J_i$, can be achieved in a twofold approach:

$$I = \sum_{i=1}^{\nu} c_i \int_C u\, e_i = \sum_{i=1}^{\nu} c_i\, J_i \tag{10}$$

or

$$I = \sum_{i=1}^{\nu} c_i \int_{C_i} u\,\phi = \sum_{i=1}^{\nu} c_i\, J_i\,. \tag{11}$$

The former decomposition involves a basis formed by independent equivalence classes $\{e_i\}$ of the underlying twisted cohomology, while the latter involves a basis formed by independent equivalence classes $\{C_i\}$ of the twisted homology. Remarkably, the dimension of the twisted homology and co-homology spaces is the same.

Let us finally remark that although Baikov representation turned out to be useful to uncover the cohomological structure of Feynman integrals [59], there is no commitment to necessarily use it. Other parametric representations, such as Feynman-Schwinger [93], $n$-dimensional polar coordinates [94], and Lee-Pomeransky representation [53], to name a few, can be equivalently used [64]. In fact, the integral in (1) can be cast in the form [53]

$$I_{\nu_1,\cdots,\nu_N} = \frac{\Gamma(D/2)}{\Gamma((L+1)D/2 - |n|)\prod_a\Gamma(\nu_a)} \int_0^\infty \cdots \int_0^\infty \frac{dz_a}{z_a^{1-\nu_a}} \mathcal{G}^{-D/2}\,, \tag{12}$$

with $|n| = \sum_a \nu_a$ and $\mathcal{G} = \mathcal{F} + \mathcal{U}$, where $\mathcal{F}$ and $\mathcal{U}$ are the Symanzik polynomials. The latter are defined as

$$\mathcal{F} = \det\{A\}C - (A^{\text{adj}})_{ij}B_iB_j\,, \quad \mathcal{U} = \det\{A\}\,, \tag{13}$$

with $A$, $B$, and $C$ being the matrices that appear in the decomposition of the denominators, as

$$D_a = A_{a,ij}q_iq_j + 2B_{a,ij}q_ip_j + C_a\,, \tag{14}$$

where

$$A_{ij} = \sum_a z_a A_{a,ij}\,, \quad B_i = \sum_a z_a B_{a,ij}p_j\,, \quad C = \sum_a z_a C_a\,. \tag{15}$$

We observe that, although the integral in Equation (12) has the structure of Aomoto-Gel'fand integrals Equation (9), the polynomial $\mathcal{G}$ does not vanish on the boundary of the integration domain; therefore, surface terms can emerge in in the (co)-homology decomposition. It turns out that these extra terms can be related to integrals belonging to simpler sectors, i.e., with fewer denominators than the ones in the original diagram.

### 3. The Twisted Cohomology Vector Space

*3.1. Vector-Space Structure*

To study the co-homology of dimensionally regulated Feynman integrals, we consider integrals of the form

$$I = \int_{\mathcal{C}_R} u(\mathbf{z})\, \varphi_L(\mathbf{z}) = \langle \varphi_L | \mathcal{C}_R], \tag{16}$$

regarded as a pairing between $\langle \varphi_L |$ and the function $u(\mathbf{z})$, integrated over the contour $|\mathcal{C}_R]$. In particular, $u(\mathbf{z})$ is a multivalued function, $u(\mathbf{z}) = \mathcal{B}(\mathbf{z})^{\gamma}$ (or $u(\mathbf{z}) = \prod_i, \mathcal{B}_i(\mathbf{z})^{\gamma_i}$), with

$$\mathcal{B}(\partial \mathcal{C}_R) = 0\,. \tag{17}$$

The pairing so defined is not strictly speaking a pairing of forms but of equivalence classes of $n$-forms, such that two differential forms in the same class differ by covariant derivative-terms whose contribution under integration over a contour vanishes. Let us see how this works.

*3.2. Dual Cohomology Groups*

Let $\xi_L$ be an $(n-1)$-differential form and $\mathcal{C}_R$ an integration contour such that (17) holds true. Thus, we can use Stokes theorem to write

$$0 = \int_{\partial \mathcal{C}_R} u\, \xi_L = \int_{\mathcal{C}_R} d(u\, \xi_L) = \int_{\mathcal{C}_R} u \left( \frac{du}{u} \wedge +d \right) \xi_L = \int_{\mathcal{C}_R} u\, \nabla_\omega\, \xi_L = \langle \nabla_\omega \xi_L | \mathcal{C}_R]\,, \tag{18}$$

where

$$\nabla_\omega = d + \omega \wedge, \qquad \omega = d\log u. \tag{19}$$

As a consequence, we immediately get

$$\langle \phi_L | \mathcal{C}_R] = \langle \phi_L | \mathcal{C}_R] + \langle \nabla_\omega \xi_L | \mathcal{C}_R] = \langle \phi_L + \nabla_\omega \xi_L | \mathcal{C}_R]\,, \tag{20}$$

which means that the forms $\varphi_L$ and $\varphi_L + \nabla_\omega \xi_L$ give the same result upon integration, as stated above, and we can consider them as equivalent for the purpose of computing intersections. This is an equivalence relation and we can say that the two forms are in the same equivalence class, writing

$$\phi_L \sim \phi_L + \nabla_\omega \xi_L\,. \tag{21}$$

Of course, the equivalence classes of $n$-forms defined by the equivalence relation (21), which we will denote with $\langle \varphi_L |$, define a vector space, called the *twisted cohomology group* $H_\omega^n$.

Analogously, every time two $n$-dimensional contours differ by a boundary, they have the same boundary, so if (17) holds for one of them it holds for the other one. Using again the Stokes theorem, it follows that when used as integration contours for a closed $n$-form $u\phi_L$, they give the same result and, again, can be considered equivalent. We indicate the equivalence classes of integration contours so obtained with $|\mathcal{C}_R]$. They define the *twisted homology group* $H_n^\omega$, which is indeed an abelian group identifiable with a vector space after tensorization with $\mathbb{R}$.

Let us consider, now, what happens when using $u^{-1}$ in the integral definition instead of $u$. Let us define a *dual* integral

$$\tilde{I} = \int_{\mathcal{C}_L} u(\mathbf{z})^{-1} \, \phi_R(\mathbf{z}) = [\mathcal{C}_L | \phi_R \rangle \,, \tag{22}$$

as a pairing between the dual twisted cycle $[\mathcal{C}_L|$ and the dual twisted cocycle $|\phi_R\rangle$. It is clear by construction that what we did before can be repeated in the same way just replacing the covariant derivative with

$$\nabla_{-\omega} = d - \omega \wedge, \qquad \omega = d \log u. \tag{23}$$

Then, we immediately get an equivalence relation for dual twisted cocycles, analogous to (21),

$$\varphi_R \; \sim \; \varphi_R + \nabla_{-\omega} \tilde{\xi}_R \tag{24}$$

which defines equivalence classes denoted by $|\varphi_R\rangle$. These equivalence classes define the *dual* vector space $(H_\omega^n)^* = H_{-\omega}^n$.

As mentioned earlier, an equivalence relation among dual integration contours may also be considered, yielding to the identification of the dual homology vector space $(H_n^\omega)^* = H_n^{-\omega}$, referred to as the *dual twisted homology group*, whose elements are denoted by $[\mathcal{C}_L|$.

### 3.3. Number of Master Integrals

We have shown that we can recognize a vector space structure in Feynman integrals, so that a given integral may be expressed in some basis of the twisted cohomology: if one is able to compute the coefficients of the decomposition, which we discuss in Section 7, the computation will be reduced to the evaluation of some fixed master integrals. All this reasoning would be useless if the dimension of such a vector space turned out to be infinite. Luckily, the number of the independent MIs, which we refer to as $\nu$, is known to be actually finite [83]. Moreover, it is known how to compute $\nu$ in practice, as we show in this section.

There are actually many interpretations of $\nu$: here, we focus mainly on the interpretation given by Lee and Pomeransky [53]. Firstly, $\nu$ is the number of the many independent equivalence classes in the associated cohomology group. Due to the Poincaré duality between cohomology and homology classes, it is equivalent to study the dimension of the homology space, which consists in counting how many non-homotopically contractible closed paths exist in the space of integration contours.

We consider a particularly simple case in order to get a clear understanding of the usual reasoning followed when counting independent cycles. Once the associated Baikov representation (2) is derived, it is appropriate to perform a maximal cut (for a more detailed discussion on cuts performed on Feynman integrals in the context of Baikov representation, we recommend taking a look at [95,96]): out of the $M$ total denominators, we put to 0 the $N$ original ones corresponding to physical propagators. Of course, putting a propagator on mass shell is the same as performing a residue, which reduces the level of complexity of the integral. We consider the case where the integral along the maximal cut reduces to one with a single variable:

$$I = \int \frac{dz_1}{z_1^{\nu_1}} B(z_1)^\gamma. \tag{25}$$

We stress that, by taking the maximal cut, all the original physical propagators were eliminated, so in Equation (25), the variable $z_1$ is related to one of the fake propagators introduced in Section 2: hence we can consider $\nu_1 < 0$ so it does not introduce any additional singularities. When looking at how many types of non-equivalent integration contours one can build, it is clear that the topology of the space must be taken into account. Suppose that $B(z_1)$ has $m$ distinct zeros: the power $\gamma$ introduces in the $z_1$ complex space $m$ cuts starting from each zero point and going to infinity. Supposing that the integrand

is well-behaved at infinity (if this were not true, then the whole integral $I$ would not converge), we can connect at infinity the $m$ paths one can draw around the $m$ cuts. The resulting closed path is actually contractible in a single point; hence, only $m - 1$ paths are independent (Figure 1).

Qualitatively, notice that if $m$ is the order of the polynomial $B(z_1)$, then $m - 1$ is the order of the polynomial $\partial_{z_1} B$, and hence it is related to the number of zeros of $\partial_{z_1} B$. Getting back to the notation

$$I = \int_C u\phi\,, \tag{26}$$

where $u = B^\gamma$, it is equivalent to the number of solutions of

$$\omega = d\log u = \gamma(\partial_{z_1} B/B)dz_1 = 0\,, \tag{27}$$

called the number of *proper zeros*. Equation (27) suggests a deep connection between the number $\nu$ of MIs and the number of critical points of $B$.

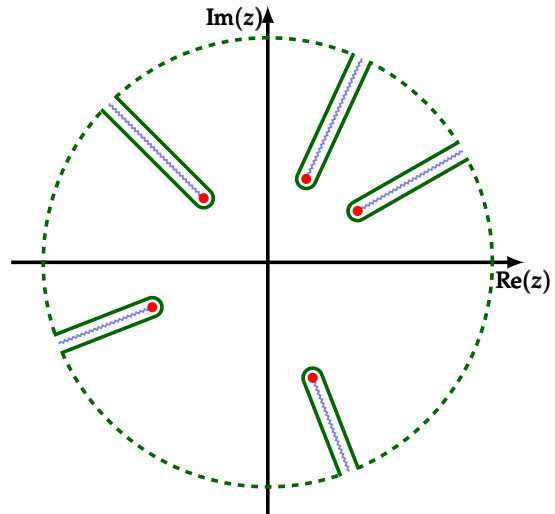

**Figure 1.** Complex plane with $m = 5$ cuts (undulating blue curves). Each cut is encircled by a path going to infinity while never crossing any cut. Dashed green lines connect at infinity the full green lines and overall create a closed path that is clearly contractible in 0.

As shown more extensively in [53], this connection is actually much more general: given an integral of the form (26), in which $\phi$ is a holomorphic $M$-form and $u$ is a multivalued function such that $u(\partial C) = 0$, then the number of Master Integrals is

$$\nu = \text{ number of solutions of the system } \begin{cases} \omega_1 = 0 \\ \vdots \\ \omega_n = 0 \end{cases}, \tag{28}$$

where

$$\omega = d\log u(\vec{z}) = \sum_{i=1}^n \partial_{z_i} \log u(\vec{z})dz_i = \sum_{i=1}^n \omega_i dz_i. \tag{29}$$

Summing up, the number $\nu$ of MIs, which is the dimension of both the cohomology and homology groups thanks to the Poincaré duality, is equivalent to the number of proper critical points of $B$, which solve as $\omega = 0$. We mention that $\nu$ is also related to another geometrical object: the Euler characteristic $\chi$ [53,87]. It is found that is linked to $\chi(P_\omega)$, where $P_\omega$ is a projective variety defined as the set of poles of $\omega$, through the relation [63]

$$\nu = \dim H^n_{\pm\omega} = (-1)^n(n + 1 - \chi(P_\omega)). \tag{30}$$

While we do not delve into the details of this particular result, we highlight how, once again, $\nu$ relates the physical problem of solving a Feynman integral as a geometrical one.

### 3.4. Linear and Bilinear Identities

The bases of dual twisted cocycles $\{e_i\} \in H_\omega^n$ and $\{h_i\} \in H_{-\omega}^n$, as well as the bases of the dual twisted cycles $|\mathcal{C}_{R,i}] \in H_n^\omega$ and $[\mathcal{C}_{L,i}| \in H_n^{-\omega}$, can be used to express the identity operator in the respective vector spaces. In particular, in the cohomology space, the identity resolution reads as,

$$\sum_{i,j=1}^{\nu} |h_i\rangle \left(\mathbf{C}^{-1}\right)_{ij} \langle e_j| = \mathbb{I}_c \tag{31}$$

where we define the *metric matrix*

$$\mathbf{C}_{ij} = \langle e_i|h_j\rangle \,, \tag{32}$$

whose elements are *intersection numbers* of the twisted basic forms. We can do the same in the homology space, where the resolution of the identity takes the form

$$\sum_{i,j=1}^{\nu} |\mathcal{C}_{R,i}] \left(\mathbf{H}^{-1}\right)_{ij} [\mathcal{C}_{L,j}| = \mathbb{I}_h \,, \tag{33}$$

with the metric matrix now given by

$$\mathbf{H}_{ij} = [\mathcal{C}_{L,i}|\mathcal{C}_{R,j}] \,, \tag{34}$$

in terms of intersection numbers of the basic twisted cycles.

Linear and bilinear relations for Aomoto-Gel'fand-Feynman integrals, as well as the differential equations and the finite difference equation they obey are a consequence of the purely algebraic application of the identity operators defined above: this is the simple observation made in [59], yielding what we consider as the profound developments reached in the studies [65].

In fact, the decomposition of any generic integral $I = \langle \varphi_L|\mathcal{C}_R]$ in terms of a set of $\nu$ MIs $J_i = \langle e_i|\mathcal{C}_R]$ reads as

$$I = \sum_{i=1}^{\nu} c_i J_i \tag{35}$$

and can be understood as coming from the underpinning decomposition of differential forms as

$$\langle \phi_L| = \sum_{i=1}^{\nu} c_i \langle e_i| \,, \tag{36}$$

(because the integration cycle is the same for all the integrals of Equation (35)). Likewise, the decomposition of a *dual* integral $\tilde{I} = [\mathcal{C}_L|\varphi_R\rangle$ in terms of a set of $\nu$ *dual* MIs $\tilde{J}_i = [\mathcal{C}_L|h_i\rangle$

$$\tilde{I} = \sum_{i=1}^{\nu} \tilde{c}_i \tilde{J}_i \tag{37}$$

becomes

$$|\phi_R\rangle = \sum_{i=1}^{\nu} \tilde{c}_i |h_i\rangle. \tag{38}$$

By applying the identity operator $\mathbb{I}_c$ to the forms $\langle \phi_L |$ and $| \phi_R \rangle$,

$$\langle \phi_L | = \langle \phi_L | \mathbb{I}_c = \sum_{i,j=1}^{\nu} \langle \phi_L | h_j \rangle \left( \mathbf{C}^{-1} \right)_{ji} \langle e_i | , \tag{39}$$

$$| \phi_R \rangle = \mathbb{I}_c | \phi_R \rangle = \sum_{i,j=1}^{\nu} | h_i \rangle \left( \mathbf{C}^{-1} \right)_{ij} \langle e_j | \phi_R \rangle , \tag{40}$$

the coefficients $c_i$, and $\tilde{c}_i$ of the linear relations in Equations (36) and (38) read as

$$c_i = \sum_{j=1}^{\nu} \langle \phi_L | h_j \rangle \left( \mathbf{C}^{-1} \right)_{ji} , \tag{41}$$

$$\tilde{c}_i = \sum_{j=1}^{\nu} \left( \mathbf{C}^{-1} \right)_{ij} \langle e_j | \phi_R \rangle . \tag{42}$$

The latter two formulas, dubbed *master decomposition formulas* for twisted cycles [59,61], imply that the decomposition of any (dual) Aomoto-Gel'fand-Feynman integral can be expressed as linear combination of (dual) master integrals is an algebraic operation, similar to the decomposition/projection of any vector within a vector space that can be executed by computing intersection numbers of twisted de Rham differential forms (cocycles).

Alternatively, by using the identity operator $\mathbb{I}_h$ in the homology space, one obtains the decomposition of (dual) twisted cycles $| \mathcal{C}_R |$ and $| \mathcal{C}_L |$ in terms of (dual) bases,

$$| \mathcal{C}_R ] = \sum_i a_i | \mathcal{C}_{R,i} ] , \qquad \text{and} \qquad [ \mathcal{C}_L | = \sum_i \tilde{a}_i [ \mathcal{C}_{L,i} | , \tag{43}$$

where

$$a_i = \sum_{j=1}^{\nu} \left( \mathbf{H}^{-1} \right)_{ij} [ \mathcal{C}_{L,j} | \mathcal{C}_R ] , \qquad \text{and} \qquad \tilde{a}_i = \sum_{i=1}^{\nu} [ \mathcal{C}_L | \mathcal{C}_{R,j} ] \left( \mathbf{H}^{-1} \right)_{ji} . \tag{44}$$

They imply the decomposition of the (dual) integrals, $I = \langle \varphi_L | \mathcal{C}_R ]$ and $\tilde{I} = [ \mathcal{C}_L | \varphi_R \rangle$, in terms of MIs, $J_i'$ and $\tilde{J}_i'$,

$$I = \langle \varphi_L | \mathcal{C}_R ] = \sum_{i=1}^{\nu} a_i J_i , \qquad \text{and} \qquad \tilde{I} = [ \mathcal{C}_L | \varphi_R \rangle = \sum_{i=1}^{\nu} \tilde{a}_i \tilde{J}_i , \tag{45}$$

with

$$J_i' = \langle \varphi_L | \mathcal{C}_{R,i} ] , \qquad \text{and} \qquad \tilde{J}_i' = [ \mathcal{C}_{L,i} | \varphi_R \rangle , \tag{46}$$

where the MIs are characterised by the independent elements of the homology bases.

In the above formulas, $\mathbf{C}$ and $\mathbf{H}$ are $(\nu \times \nu)$-matrices of intersection numbers, which, in general, differ from the identity matrix. For orthonormal intersections of basic cocycles, $\langle e_i |$ and dual-forms $| h_i \rangle$, and basic cycles, $| \mathcal{C}_{R,i} ]$ and their dual $[ \mathcal{C}_{L,i} |$, they turn into the unit matrix, hence simplifying the decomposition formulas in Equations (41), (42), and (44). As usual, one can use the orthonormalization Gram-Schmidt procedure in order to get an orthonormal basis with respect to the intersection product. For 1-forms, it is possible to construct orthonormal bases in a direct way starting from the expression of $\omega$ [61].

We can now get the quadratic identities simply by inserting the identity operators $\mathbb{I}_c$ and $\mathbb{I}_h$ in the pairing between the twisted cocyles or cycles,

$$\langle \phi_L | \phi_R \rangle \;=\; \sum_{i,j=1}^{\nu} \langle \phi_L | \mathcal{C}_{R,i} \rangle \left( \mathbf{H}^{-1} \right)_{ij} [\mathcal{C}_{L,j} | \phi_R \rangle \tag{47}$$

$$[\mathcal{C}_L | \mathcal{C}_R] \;=\; \sum_{i,j=1}^{\nu} [\mathcal{C}_L | h_i \rangle \left( \mathbf{C}^{-1} \right)_{ij} \langle e_j | \mathcal{C}_R] \, . \tag{48}$$

These are the *Twisted Riemann's Period Relations* (TRPR) [74]; see also Equation (151).

For applications of twisted intersection numbers to bilinear relations and to Gel'fand-Kapranov–Zelevinski systems, see [54–56,70,71,82,88,89,97–99].

## 4. Pictorial (Co)Homology

Before providing a geometric interpretation of Feynman-type integrals, we want to recall some fundamental facts of topology, homology and cohomology, only at an intuitive level, for those who need a refresher on these concepts.

### *4.1. The Euler Characteristic*

One of the most known topological facts is probably the Euler theorem, Reference [100], relating the numbers of faces, edges, and vertices in a tessellation of a compact simply connected region of the plane, such as an electricity grid or a Feynman diagram without external legs: the number $v$ of vertices minus the number $j$ of edges plus the number $f$ of faces is always equal to 1,

$$v - j + f = 1, \tag{49}$$

independently of the details of the tessellation. This can be quite easily understood as follows. The rules of the game are that any face is simply connected and has a one-dimensional boundary that is the union of edges and vertices (no lines or vertices are internal to a face); any edge is a simple line ending at two vertices (which may coincide); and no vertices belong to an internal point of an edge.

With these rules, given a simply connected region, the simple tessellation we can do is with just one face (covering the whole region), one edge (the whole boundary cut in a point), and one vertex (the cut point). Therefore, in this case, $v - j + f = 1$. Now, suppose it has given a tessellation. We can modify it by adding an edge. This can be done in four different ways:

- The new edge starts and ends in already existing (and possibly coincident) vertices (Figure 2). In this case, the new edge cuts a face in two, and the new tessellation has $v' = v, j' = j + 1, f' = f + 1$ so that $v' - j' + f' = v - j + f$;

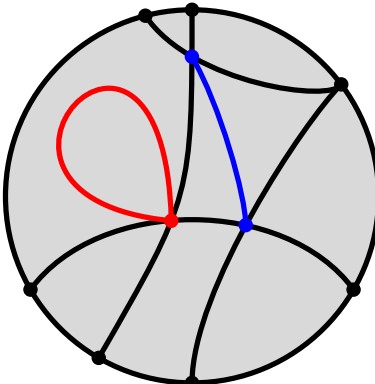

**Figure 2.** The red edge starts and ends at the same vertex, while the blue one belongs between two vertices.

- The new edge extends from an already existing vertex to a new vertex attached to an existing edge (Figure 3). The new edge separates a face into two faces and the new vertex separates the old edge into two edges. Therefore, the new tessellation has $v' = v + 1$, $j' = j + 2$, $f' = f + 1$, so that $v' - j' + f' = v - j + f$;

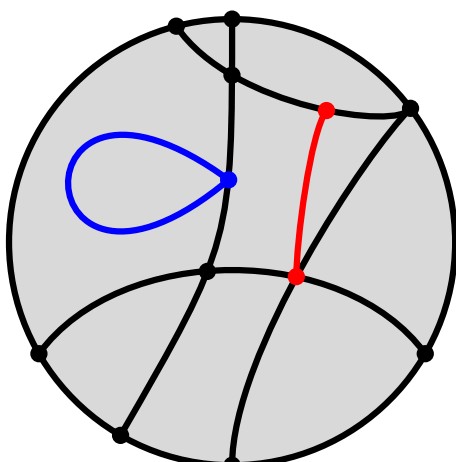

**Figure 3.** The blue edge starts and ends at the same new vertex, while the red one starts from an old vertex and ends in a new one.

- The new edge starts from and ends to a unique new vertex, inserted inTO an old edge (Figure 3). The new edge cuts a face into two, and the new vertex cuts the old edge into two. Therefore, the new tessellation has $v' = v + 1$, $j' = j + 2$, $f' = f + 1$, so that $v' - j' + f' = v - j + f$;
- the new edge extends from a new vertex to a different new vertex attached to two different or the same existing edge (Figure 4). The new edge separates a face into two faces. The new vertices separate two edges into two parts each or a unique edge in three. Therefore, the new tessellation has $v' = v + 2$, $j' = j + 3$, $f' = f + 1$, so that $v' - j' + f' = v - j + f$.

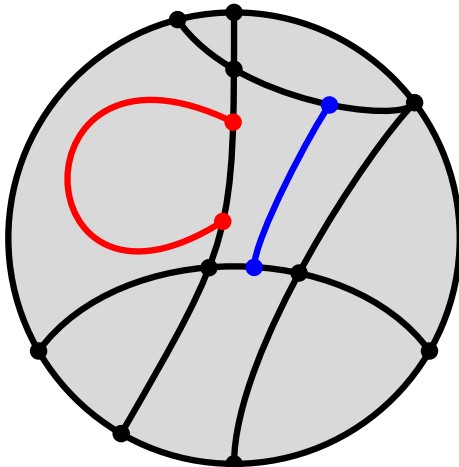

**Figure 4.** The red edge lies between two different new vertices on the same edge, while the blue one lies between two new vertices belonging on two different edges.

Therefore, we see that $\chi := v - j + f$ is invariant, and since any tessellation can be obtained from the most elementary one by acting with these operations, we see that $\chi = 1$. What happens if in place of a simply connected region we consider a region with one hole? We can again consider a tessellation for it and prove that $\chi = v - j + f$ is invariant. This follows from the fact that if we close the hole by adding the face corresponding to it, we get a simply connected region with $f' = f + 1$, $v' = v$, $j' = j$, so that

$$1 = v' - j' + f' = v - j + f + 1, \quad \Rightarrow \quad \chi = v - j + f = 0. \tag{50}$$

Changing the topology, the number $\chi$ is changed. From the same reasoning, we see that if we consider $k$ holes, then

$$\chi = 1 - k. \tag{51}$$

Having understood the mechanism, we can go beyond and see that $\chi$ does not change if we deform the given region a bit, without changing its topology. For example, we can deform a simply connected region $S$ to cover part of a sphere. However, if we close this surface to cover the sphere, the topology changes and what we do to $(v, j, k)$ of the original piece of surface is to add a face so that for the sphere (Figure 5),

$$\chi_{S^2} = \chi_S + 1 = 2. \tag{52}$$

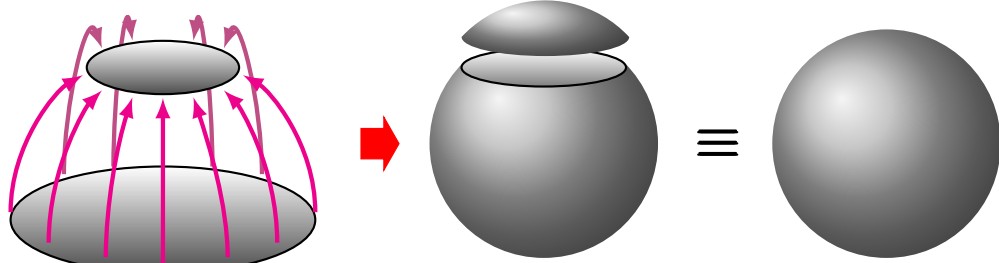

**Figure 5.** A sphere is obtained by adding a face to a disc: the Euler characteristic increases by 1.

Each time we make a hole on the sphere, we diminish $\chi_{S^2}$ by 1. What does it change if we pass from a sphere to a torus? We can get a torus from a sphere in the following way. First, we can cut two spherical caps out of the sphere (say along the arctic and antarctic

polar circles). What remains is a deformed piece of a cylinder, and since it is like a sphere with two holes it has $\chi = 0$. The torus is then obtained by gluing two such pieces along the circles (Figure 6). Since both have $\chi = 0$, we see that for the torus $T^2$ we have

$$\chi_{T^2} = 0. \tag{53}$$

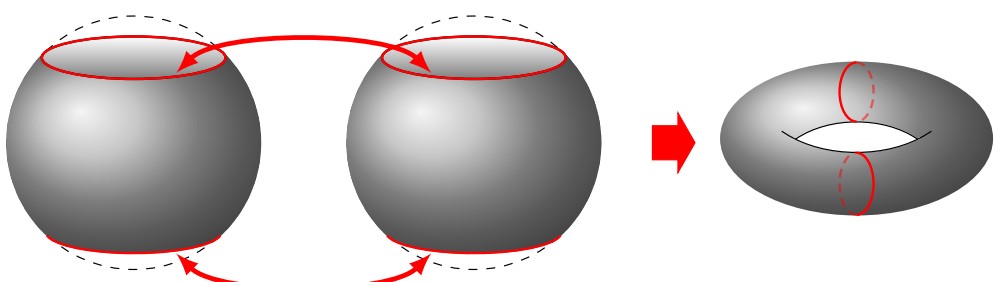

**Figure 6.** A torus is obtained by gluing two cylinders along the boundaries.

If we practice $k$ holes on the torus we get $\chi = -k$. If we want to pass to a surface of genus 2, we have to practice two holes in the torus surface and glue the extremities of a piece of cylinder to the holes. The holes diminish $\chi$ by two, while the cylinder is harmless, so $\chi = -2$ in this case. With the same construction we see that for a surface $K_{g,k}$ of genus $g$ and $k$ holes on the surface we have

$$\chi_{K_{g,k}} = 2 - 2g - k. \tag{54}$$

This topological number is called the *Euler characteristic of the surface*.

### 4.2. Simplicial (Co)homology

The simplest tessellation we can think of for a surface is a triangulation. The surface is then equivalent to the union of triangles, which are equivalent to two-dimensional simplexes (the convex hull generated by three non-aligned points in $\mathbb{R}^N$). A triangle has the union of three segments (and three points) as boundary. Each segment is a one-dimensional simplex and each point a zero-dimensional simplex. Two points are the boundary of a 1-simplex. This way, we see the surface as a collection of simplexes glued together. If the triangulation is fine enough, we can guarantee that two simplexes of given dimension meet at most at a simplex of lower dimension (a face). Moreover, all sub-simplexes of any simplex re also in the collection. Such a collection is called a *simplicial complex*. Thus, we can see a surface as a simplicial complex.

Suppose we want to compute the Euler characteristic starting from it. We have to count the number of faces, edges, and vertices. Now, for example, let us consider a face. It is a filled triangle whose boundary is an empty triangle, so it has one face, three edges, and three vertices. If we squash the triangle along an edge until it collapses over the other two edges, the face disappears and we are left with two edges and three vertices (Figure 7).

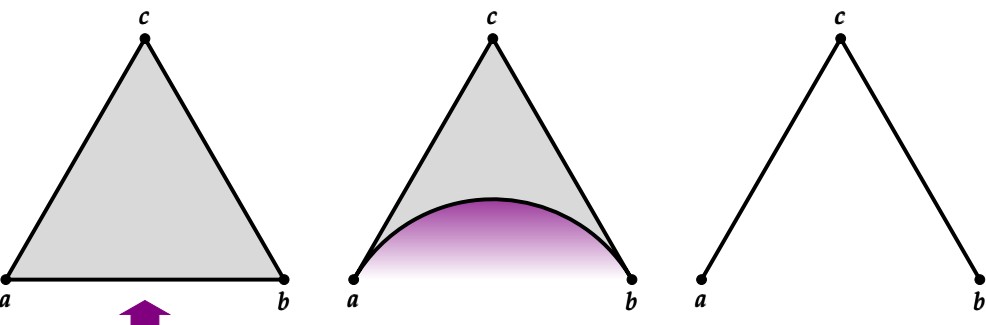

**Figure 7.** The edge *ab* is squashed until the triangle reduces to two edges.

However, $\chi$ is left invariant. The idea is then to say that the squashed edge is equivalent to the other two. If we orient the triangle, call it $(abc)$, so that it has a well-defined traveling direction, what we have done can be reformulated by saying that $(ab)$ can be squashed to $(ac) \cup (cb)$ after elimination of the face. In practice one realizes this by saying that the boundary of the 2-simplex $(abc)$ is

$$\partial(abc) = (bc) - (ac) + (ab), \tag{55}$$

where the sign respects the orientation, and stating that *if a loop is a boundary, it is trivial*: $\partial(abc) \sim 0$, which means $(ac) \sim (bc) + (ab)$. Furthermore, if we take a union of such triangles, we see that we can eliminate the common edges (intersections between simplexes) without changing $\chi$, so as to obtain a simply connected polygon $P$ that is a boundary of a region. Again, we will get that one of the edges of the polygon is equivalent to the sum (union) of the remaining ones. It is easy to check that this can be written as

$$P = \sum_j T_j, \tag{56}$$

$$\partial P = \sum_j \partial T_j \sim 0, \tag{57}$$

where $T_j$ are the simplexes comprising $P$. However, it could happen that the polygon so obtained is not simply connected but has a number of holes. The best we can say this way is that, after collapsing the triangles, the "more external boundary" of the polygon is equivalent to the sum of boundaries of the internal holes; see Figure 8.

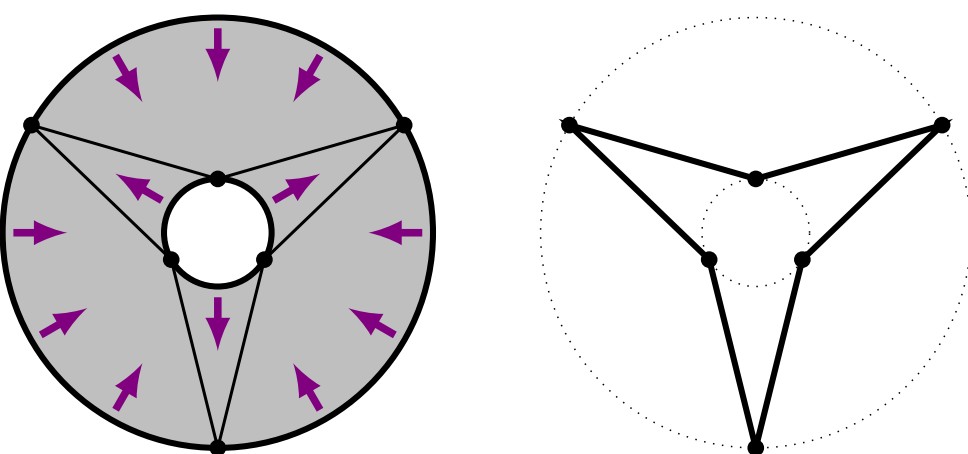

**Figure 8.** All triangles are squashed so the ring is reduced to a closed path homotopic to a circle.

Strictly speaking, however, these are not really boundaries, since there is indeed a hole and not a face that they are boundaries of.[1] This construction shows that in place of

counting all edges, one is interested in counting how much unions of edges forming closed paths are independent in the above sense (so unions of closed paths are not boundaries).

The same construction can be done for counting vertices. If two vertices $(a)$ and $(b)$ are the two tips of an edge, then, in place of counting two vertices and one edge, we can just count one vertex and zero edges, without changing $\chi$. In this case, we can restate this by saying that $(a) \sim (b)$

$$\partial(ab) = (b) - (a) \tag{58}$$

and saying that, again, boundaries are trivial: $\partial(ab) \sim 0$, $(a) \sim (b)$. Notice that if a union $L = \sum_j e_j$ of edges forms a closed path than $\partial L = 0$. In particular, then, $\partial(\partial P) = 0$ for any polygon $P$. Finally, we notice that if a connected surface has a boundary, then by collapsing triangles starting from the boundary, one can always reduce to zero the number of faces, while if it has no boundary (like a sphere or a torus), then we can reduce the number of faces to one. This is exactly the number of closed surfaces that are not boundaries (obviously for dimensional reasons, things could change if we worked, for example, in three dimensions). We can summarize such discussion as follows. We call a $j - chain$ the finite union of $j$-dimensional simplexes, with relative coefficients:

$$c = \sum_{a=1}^{N} b_a s_a, \qquad \dim(s_a) = j. \tag{59}$$

The sign of $b_a$ establishes the orientation, and its modulus is the "number of repetitions". Therefore, the set of chains is a linear space over $\mathbb{Z}$. We say that $c$ is closed if $\partial c = 0$ and that it is exact if $c = \partial b$ for $b$ a $(j+1)$-chain. We are then interested in counting the closed chains that are independent with respect to the relation that two close chains are equivalent if they differ by an exact chain. This space is called the $j$-th simplicial homology group

$$H_j(S, \mathbb{Z}), \tag{60}$$

of the surface $S$. Then, we can define the *Betty numbers* $b_j = \dim H_j$ so that the above reasoning leads us to the result

$$\chi_S = \sum_{j=0}^{2} (-1)^j b_j. \tag{61}$$

This result does not change if we change the coefficients allowing $b_a$ to take values in $\mathbb{K} = \mathbb{Q}, \mathbb{R}, \mathbb{C}$. In this case, the Homology groups become vector spaces. Furthermore, we can consider the cohomology groups by replacing simplexes $s_k$ with continuous maps ($j$-cochain)

$$\sigma_k : s_k \to \mathbb{K}, \tag{62}$$

and the boundary operators $\partial$ with the coboundary operators $\delta$, sending $j$-cochains in $(j+1)$-cochains, defined by

$$\delta(\sigma_j)(c_{j+1}) \equiv \sigma_j(\partial c_{j+1}). \tag{63}$$

This defines the cohomology groups $H^j(S, \mathbb{K})$. It follows that $H^j(S, \mathbb{K})$ is the linear dual of $H_j(S, \mathbb{K})$, so that there are isomorphic as vector spaces.

*4.3. Morse Theory*

Another way of understanding the topology of a differentiable manifold $M$ is to use properties of functions $f : M \to \mathbb{R}$. Assume these functions are smooth, with non

-degenerate isolated critical points. A critical point is a point $p$ where $df(p) = 0$. It is non-degenerate if its Hessian is different from zero. We will not delve here into the details (see [101] for these), but we just want to show how such function may capture the topological properties by looking at the explicit example of a torus. Referring to Figure 9, let consider the map that at $p$ associates the quote $z = f(p)$. The critical points are the points where the horizontal plane is tangent to the surface. We see that there are four critical points: a maximum, a minimum, and two saddle points.

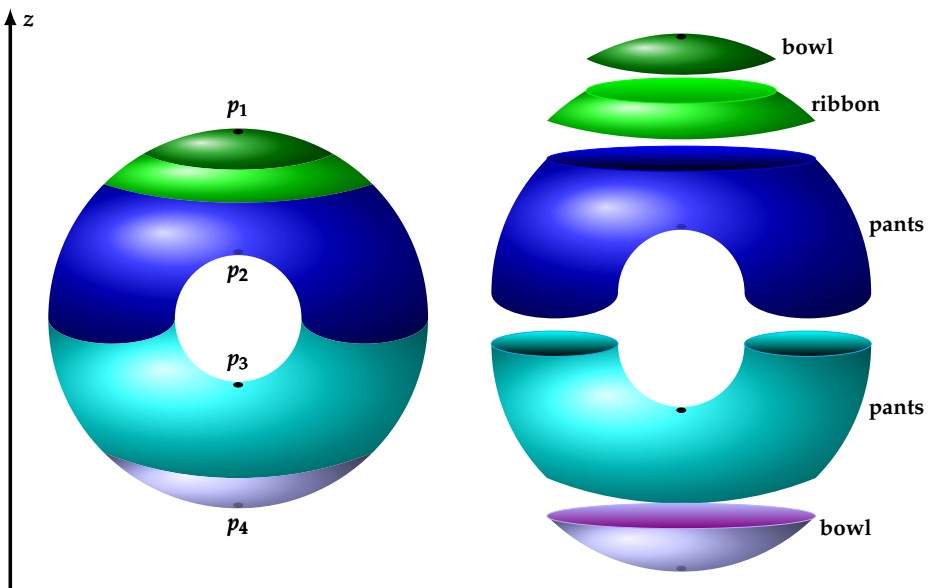

**Figure 9.** A torus is cut in pieces with different topologies. Notice that the ribbon can be pasted to the bowl or to the pants without changing the topology of either the bowl or the pants: cutting away pieces not containing critical points does not change the topology.

Let us see how this function can give us a new way to reconstruct the surfaces starting from pieces. Starting from the above, assume that we use the horizontal plane to cut the surface. First, we will meet the top $p_1$ of the surface, corresponding to the maximum. If we cut a little bit below, we get a small bowl facing down. Then, let us move below. If we cut before meeting the second critical point, we cut out a sort of ribbon, whose boundaries are homologically equivalent. This will occur each time we take two cuts not containing any critical point in between. Then, we move even below until meeting the saddle point. If we cut a little bit below, we get a piece that looks like a pair of pants. If we go even more below, until passing the second saddle point $p_3$, we get a second pair of pants (with the legs upside).[2] Going below the point $p_4$ we are left with a second bowl. Forgetting the trivial pieces, we see that we can reconstruct the surface by gluing together the shapes obtained after cutting horizontally around the critical points. A maximum gives a bowl down, any saddle point gives a pair of pants, and a minimum gives a bowl up. Each of these shapes is understood by looking at the signature of the Hessian at each critical point. Starting from above, we have just to look at the principal ways to go down (Figure 10):

- At $p_1$, the Hessian is negative, so it has two principal directions going down (the eigenvectors). We say that $p_1$ has Morse index 2.
- At $p_2$ and $p_3$, the Hessian is indefinite, it has only one negative eigenvalue at which it corresponds a descending direction. We say that $p_2$ and $p_3$ have Morse index 1.
- At $p_4$, the Hessian is positive. There are not descending directions and we say that $p_4$ has Morse index 0.

If $m(n)$ is the number of critical points with Morse index $n$, then we can define the Euler number associated to our surface $S$ ($M$ stays for Morse):

$$\chi_M(S) = \sum_{n=0}^{2} (-1)^n m(n). \tag{64}$$

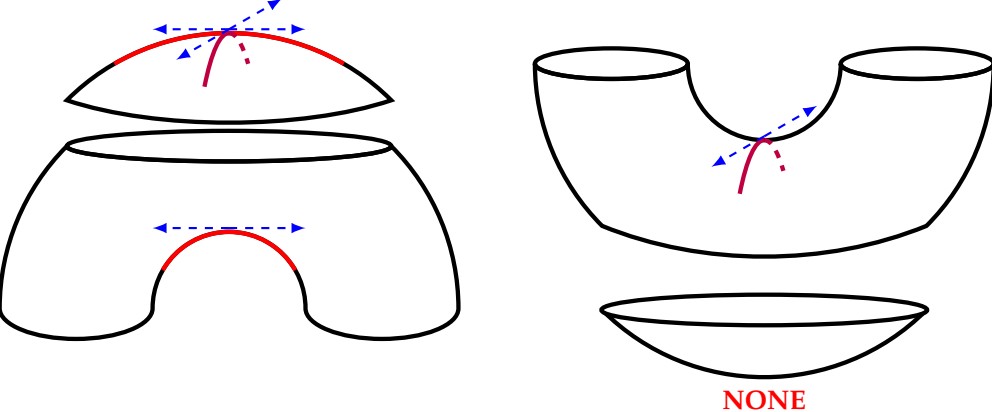

**Figure 10.** Principal descending directions from the critical points at any piece.

Notice that in our example we get $\chi_M(S) = 0$. If we deform the surface without changing the topology, like in Figure 11, the number of critical points can change, but $\chi_M(S)$ remains unchanged.

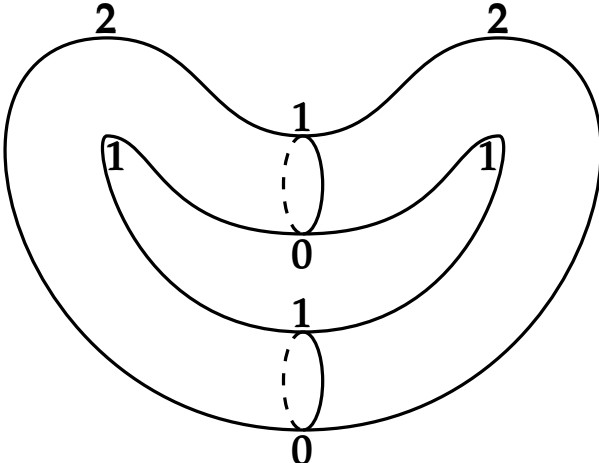

**Figure 11.** Deforming the torus does not changes the Morse index.

It is a topological invariant and, in this case, it is zero, equal to the Euler characteristic of the torus. It is possible to prove that $\chi_M(S)$ is always equal to the Euler characteristic, in any dimension, for any closed compact manifold. In the next picture, we can see that, indeed, for a sphere $S^2$, we get $\chi_M(S^2) = 2$ and for a surface $\Sigma_g$ of genus $g$, one has $\chi_M(\Sigma_g) = 2 - 2g$.

### 4.4. Cellular (Co)homology

Another way to look at homology (and then cohomology by duality, as usual) is to look for a cellular decomposition of our surface, [102]. This means that we have to obtain the surface (up to deformations preserving the topology) by gluing discs of different dimensions. A disc is meant to be a full ball in a given dimension, so a zero-dimensional disc is a point and a one-dimensional disc is a segment. The rule is that one starts from

the lower dimensional discs and then glues the boundaries of the successive dimensional discs to the lower dimensional structure obtained previously. For example, one can get a sphere starting from a 0-cell and a 2-cell, gluing the boundary of the two cell to the point (Figure 12).

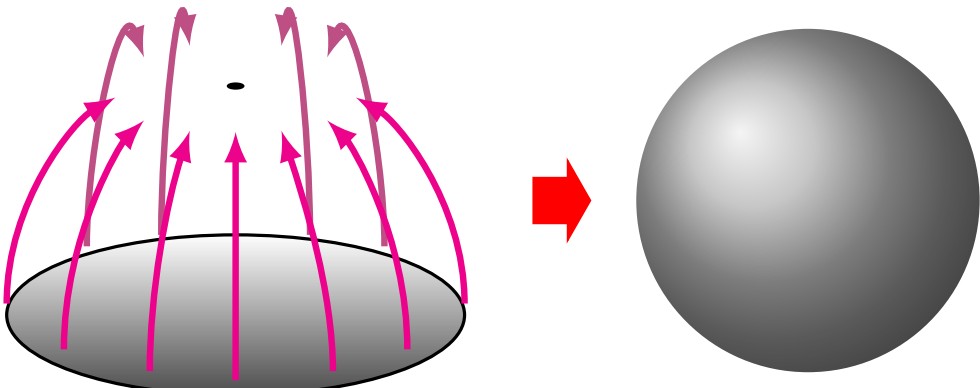

**Figure 12.** A cellular decomposition of the sphere: a disc is glued to a point.

Alternatively, starting from a point, a segment and two discs, one can do so by gluing the boundaries of the segments to the point (getting $S^1$) and gluing the boundaries of the two 2-cells to $S^1$ (Figure 13).

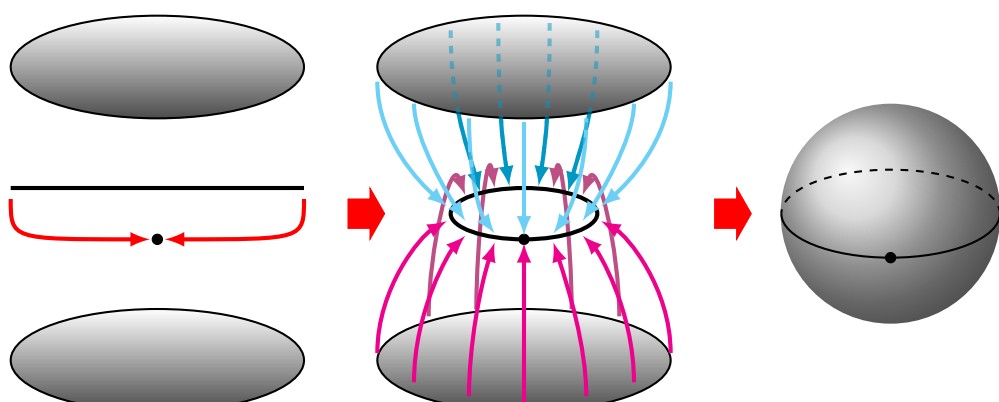

**Figure 13.** A different cellular decomposition of the sphere.

These are different constructions of $S^2$ gluing discs, and we can think many others. The point is that if we indicate with $d(n)$ the numbers of cells of dimension $n$, then we get for the sphere

$$\chi_{CW}(S^2) := \sum_{n=0}^{2} (-1)^n d(n) = 2 \tag{65}$$

independently of the construction. Notice that, again, it coincides with the Euler characteristic, and, again, it is not by chance but it is a general result: the cellular decomposition gives us another way to compute the Euler characteristic. In Figure 14, it is shown how to get a genus $g$ surface $\Sigma_g$ by gluing a point, $2g$ segments and a 2-disc.

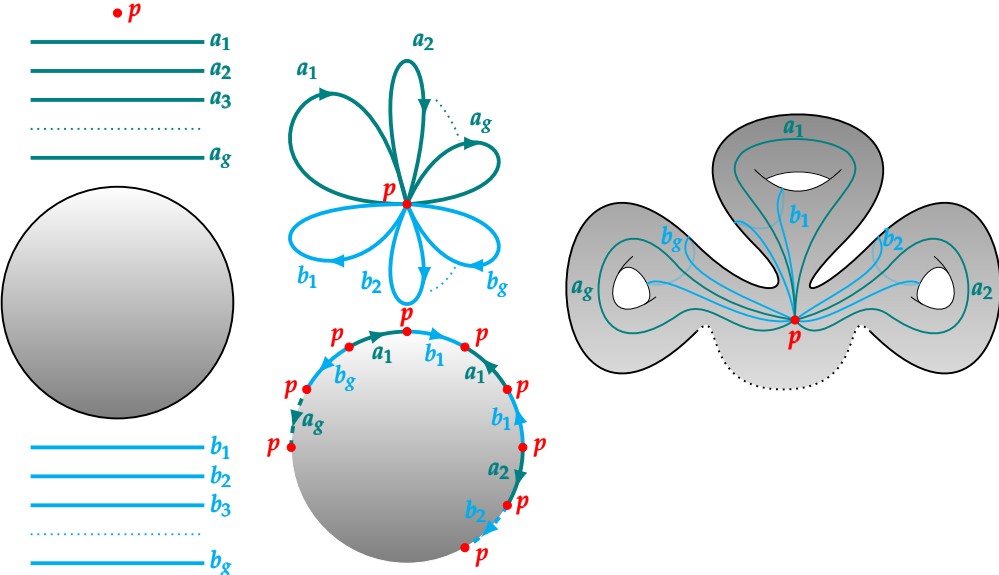

**Figure 14.** A cellular decomposition of a genus $g$ surface. All the boundaries of the $2g$ one dimensional cells are glued to a point, giving a one-dimensional skeleton. Then the boundary of the two-dimensional disc is glued to the skeleton, gluing the red points to p and the remaining parts to the corresponding lines (two segments for each curve) by respecting the drawn orientations.

Again, we see that

$$\chi_{CW}(\Sigma_g) = \chi_E(\Sigma_g) = \chi_M(\Sigma_g). \tag{66}$$

*4.5. de Rham Cohomology*

Finally, we want just to recall what is probably the most commonly known cohomology, which is the de Rham cohomology; see Reference [103]. On a smooth surface, we can have 0-forms (functions), 1-forms, and 2-forms. The external differential maps $k$-forms into $(k+1)$-forms. A $k$-form $\phi$ is said to be closed if $d\phi = 0$, while it is called exact if $\phi = d\psi$ for $\psi$ a $(k-1)$-form. An exact form is also closed since $d^2 = 0$. This is an obvious consequence of the Schwartz's lemma for double derivatives. Therefore, exact forms generate a subspace of closed forms. The de Rham cohomology group of degree $k$ is the space of closed $k$-forms identified up to exact forms. If $\Omega^k(S)$ are the $k$-forms:

$$H^k_{dR}(S) = \{\omega \in \Omega^k(S)|d\omega = 0\} / \sim, \tag{67}$$

where $\omega_1 \sim \omega_2$ if and only if $\omega_1 - \omega_2 = d\lambda$ for some $\lambda \in \Omega^{k-1}(S)$. It turns out that, under reasonable hypotheses, $H^k_{dR}(S)$ has finite dimension as a real vector space. Its dimension $b_k = \dim H^k_{dR}(S)$ is called the $k$-th Betty number of $S$ and is a topological invariant. For $k = 0$, the closed forms are the locally constant functions. They are closed but not exact, so $H^0_{dR}(S)$ is the space of locally constant functions. If $S$ is connected, any constant function is just identified by its value, so $H^0_{dR}(S) = \mathbb{R}$ and $b_0 = 1$. On a bidimensional surface, any 2-form is closed. Suppose that $S$ is connected and closed (i.e., has no boundary) and that $\omega_1 \sim \omega_2$ are two equivalent 2-forms. Then, $\omega_2 - \omega_1 = d\psi$ for $\psi$ a 1-form, and using Stokes theorem:

$$\int_S \omega_2 - \int_S \omega_1 = \int_S d\psi = \int_{\partial S} \psi = 0, \tag{68}$$

so two equivalent forms have the same integral. The opposite is also true, and we get that for a closed connected surface, $b_2 = 1$. It is difficult but possible to prove that if the surface has genus $g$, then $b_1 = 2g$. Since $b_j$ are invariants, it follows that

$$\chi_{dR}(S) := \sum_{n=0}^{2} (-1)^n b_n \tag{69}$$

is also an invariant. We see that if $S = \Sigma_g$, we get

$$\chi_{dR}(\Sigma_g) = 2 - 2g = \chi_{CW}(\Sigma_g) = \chi_E(\Sigma_g) = \chi_M(\Sigma_g). \tag{70}$$

Once more, this is a general result. We can now stop here, we will discuss a little bit further about the relation of the de Rham cohomology with other cohomologies in a more general setting (where we will also see a further cohomology, the singular cohomology).

## 5. Integrals and Cohomologies

While calculus with functions of one real variable gives us the impression of a strict relation between differential and integral calculus, it is pretty evident that when passing to more variables, or just to one complex variable, things change drastically, and the theory of integration looks deeply different. Indeed, the sensitivity of integration to global questions quickly becomes manifest. Integration looks more related to (co)homology and Hodge theory. In this section, we summarize some facts we probably will have to consider in a full program of investigation about the geometry underlying Feynman integrals. Of course, we cannot be exhaustive, but our aim is to be at least suggestive.

A convenient way to see it is to pass through complete elliptic integrals. These are one-dimensional integrals whose structure resembles that of Feynman integrals, hiding the main characterizations we need, and are also the origin of larger dimensional integration theory, starting from Poincaré and developed by Lefschetz and Hodge [104–108].

### 5.1. Elliptic Integrals

Consider the elliptic integrals of first and second kind, Reference [109]:

$$K(k) = \int_0^1 \frac{dx}{\sqrt{(1-x^2)(1-k^2x^2)}} = \frac{1}{2} \int_{-1}^1 \frac{dx}{\sqrt{(1-x^2)(1-k^2x^2)}}, \tag{71}$$

$$E(k) = \int_0^1 \frac{\sqrt{1-k^2x^2}}{\sqrt{1-x^2}} \, dx = \frac{1}{2} \int_{-1}^1 \frac{\sqrt{1-k^2x^2}}{\sqrt{1-x^2}} \, dx. \tag{72}$$

It is known that these integrals cannot be expressed in terms of elementary functions of $k$. In both cases, the last integral extends between two branch points of the integrands. There are four branch points, at $z_\pm = \pm 1$ and $\tilde{z}_\pm = \pm k^{-1}$, $k \neq \pm 1$. As the integrand is a multivalued function, we can extend it to a single-valued function on a Riemann surface. To this end, we can first extend $x$ to $z$ on the complex plane, with two cuts, one from $-1$ to 1, the other from $-k^{-1}$ to $k^{-1}$. Notice that in both cases, the integrand is regular at infinity, so we can think at it as defined on the Riemann sphere $\mathbb{P}^1$ with the two given cuts. If we try to cross one of the cuts, the function flips the sign and the only way to keep the function single-valued is to assume that we end up to a second copy of the cut Riemann sphere, where at the doubled point it takes the same value as on the original sphere, but with the opposite sign. With this standard construction, we end up with a well-defined single-valued function on a two-dimensional surface obtained by gluing the edges of each cut of a sphere to the edges of the corresponding cut on the second sphere. This is topologically a torus, that is, a Riemann surface of genus two or, from the complex point of view, an elliptic curve; Figure 15.

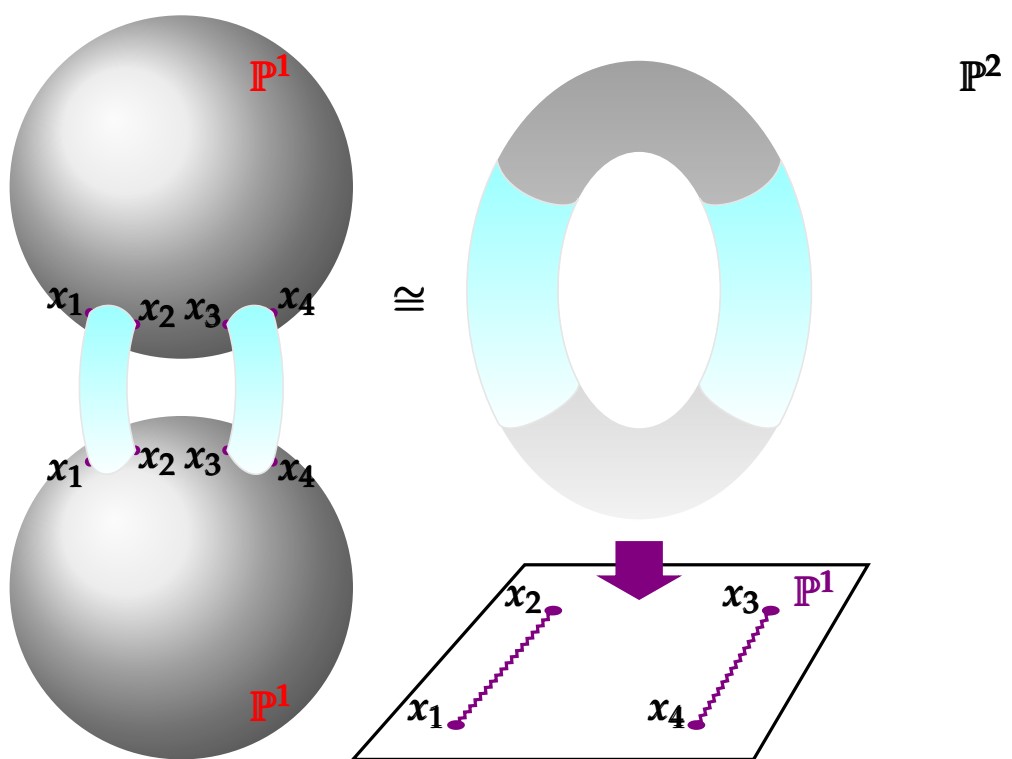

**Figure 15.** Two double-cut $\mathbb{P}^1$ planes glued along the cuts are topologically equivalent to a torus.

We see that the integrals can then be thought as along the edge of the cut where the root is positive. If we go back integrating along the other edge of the cut, we get the same result. Since the whole cut traveled this way is a closed path $\gamma$ along the elliptic curve, we can write

$$K(k) = \frac{1}{4} \oint_{\gamma} \frac{dx}{\sqrt{(1-x^2)(1-k^2x^2)}}, \tag{73}$$

$$E(k) = \frac{1}{4} \oint_{\gamma} \frac{\sqrt{1-k^2x^2}}{\sqrt{1-x^2}} \, dx. \tag{74}$$

Here, $\gamma$ is one of the generators of the first homology group of the elliptic curve. In this form, the elliptic integrals looks like a 1-form integrated over a closed cycle. Since the first homology group of an elliptic curve is two-dimensional, there is a second independent curve, which we can represent, for example, as the path going from $z_+$ to $\tilde{z}_+$ on one of the spheres and coming back from the other sphere. This second path, say $\beta$, intersects $\gamma$ transversally at a point. After having chosen an orientation, we can define the intersection product $\beta \cdot \gamma = 1$, meaning that the velocities $\dot{\beta}$ and $\dot{\gamma}$ at the intersection point form a basis for the tangent space correctly oriented. Of course this means that $\gamma \cdot \beta = -1$. However, in place of considering the same integral changing $\gamma$ with $\beta$, let us first consider the other canonical integrals $K(k')$ and $E(k')$, with $k' = \sqrt{1-k^2}$. These integrals are related to the previous ones by the Legendre's quadratic relation

$$K(k)E(k') + E(k)K(k') - K(k)K(k') = \frac{\pi}{2}. \tag{75}$$

To understand the meaning of this quadratic relation, it is convenient to pass to a more canonical description of the integrals of first and second kind. Consider an integral of the form

$$I_1 = \int_{x_1}^{x_2} \frac{dx}{\sqrt{P(x)}}, \tag{76}$$

where $P$ is a fourth-order polynomial with simple roots, among which there are $x_1$ and $x_2$ (not necessarily real). Since the integrand is not single-valued, this integral depends on the specification of the path connecting the two roots. However, it is clear that we can work exactly as before, so that we can write[3]

$$I_1 = \frac{1}{2} \int_\gamma \frac{dx}{y}, \tag{77}$$

where $\gamma$ is a closed path on the genus one complex line defined by the equation

$$y^2 = P(x) \tag{78}$$

in $\mathbb{P}^2$.[4] Notice that in the form (76), we are working just on a $\mathbb{P}^1$ projected component $\mathbb{P}^2 \mapsto \mathbb{P}^1$ (this is the reason why the branch points appear). On $\mathbb{P}^1$, there is the action of the Möbius fractional group $PGL(2, \mathbb{C})$, which allows us to move three points in any desired position. It works as follows. Let $z$ be the inhomogeneous coordinate on $\mathbb{P}^1$, and

$$A = \begin{pmatrix} a & b \\ c & d \end{pmatrix} \in GL(2, \mathbb{C}). \tag{79}$$

Its action on $z$ is thus defined by

$$Az = \frac{az + b}{cz + d}. \tag{80}$$

For $\lambda \in \mathbb{C}_*$, we se that $\lambda A$ gives the same action as $A$, so it reduces to an action of the projective group $PGL(2, \mathbb{C})$. By moving one of the roots to infinity, it is always possible to bring the integral in the form $I_1 = \chi J_1$, where

$$J_1 = \int_\gamma \frac{dx}{y}, \tag{81}$$

where now the elliptic curve is defined by the Weierstrass normal form[5] [110]

$$y^2 = 4x^3 - g_2 x - g_3. \tag{84}$$

Another choice is to bring it in the Legendre normal form

$$y^2 = (1 - x^2)(1 - k^2 x^2), \tag{85}$$

that is, relating it to the complete elliptic integral of the first kind. We are interested in the Weierstrass form now. Associated with it, the elliptic integrals of the second kind can be written in the form

$$J_2 = \int_\gamma \frac{x\,dx}{y}. \tag{86}$$

Let us now quickly see how $J_1$ and $J_2$ are related to the theory of Weierstrass $\wp$ and $\zeta$ elliptic functions. For $\omega_1$ and $\omega_2$, two complex numbers such that

$$\tau := \frac{\omega_2}{\omega_1} \tag{87}$$

has strictly positive imaginary part, one defines the Weierstrass gamma function $\wp$ as [111]

$$\wp(z) = \frac{1}{z^2} + \sum_{(m,n)\in\mathbb{Z}_0^2} \left( \frac{1}{(z - m\omega_1 - n\omega_2)^2} - \frac{1}{(m\omega_1 + n\omega_2)^2} \right), \tag{88}$$

where $m, n$ are relative integers with $(m, n) \neq (0, 0)$. It is biperiodic, with periods $\omega_1$ and $\omega_2$, and satisfies the differential equation

$$(\wp')^2 = 4\wp^3 - g_2\wp - g_3, \tag{89}$$

with

$$g_2 = \sum_{(m,n)\in\mathbb{Z}_0^2} \frac{60}{(m\omega_1 + n\omega_2)^4}, \tag{90}$$

$$g_3 = \sum_{(m,n)\in\mathbb{Z}_0^2} \frac{140}{(m\omega_1 + n\omega_2)^6}. \tag{91}$$

The Weierstrass $\zeta$ function is

$$\zeta(z) = \frac{1}{z} + \sum_{(m,n)\in\mathbb{Z}_0^2} \left( \frac{1}{z - m\omega_1 - n\omega_2} + \frac{1}{m\omega_1 + n\omega_2} + \frac{z}{(m\omega_1 + n\omega_2)^2} \right). \tag{92}$$

It is then clear that $\zeta'(z) = -\wp(z)$. However, $\zeta(z)$ is not biperiodic, but it satisfies the relations

$$\zeta(z + \omega_j) = \zeta(z) + 2\eta_j, \quad \eta_j = \zeta(\frac{\omega_j}{2}), \quad j = 1, 2, 3, \tag{93}$$

with $\omega_3 = \omega_1 + \omega_2$. In this way, the Legendre's quadratic relation takes the form

$$\eta_1\omega_2 - \omega_1\eta_2 = i\pi, \tag{94}$$

which can be easily proven after integrating $\zeta$ along a fundamental parallelogram centered at 0, with fundamental periods as edges [111]. This means that the rectangle is $(abcd)$, with

$$(a, b, c, d) = (-\frac{\omega_1 + \omega_2}{2}, \frac{\omega_1 - \omega_2}{2}, \frac{\omega_1 + \omega_2}{2}, \frac{-\omega_1 + \omega_2}{2}), \tag{95}$$

see Figure 16.

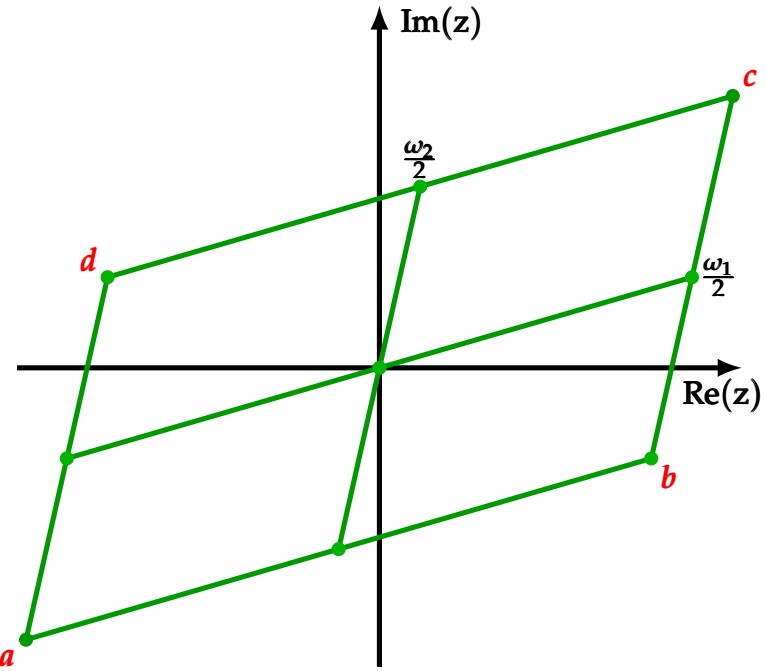

**Figure 16.** Fundamental parallelogram.

Now, in the fundamental region, the function $z$ has a unique simple pole, located in $z = 0$, with residue 1. Therefore, we can write

$$2\pi i = \int_{(abcd)} \zeta(z)dz = \int_a^b \zeta(z)dz + \int_b^c \zeta(z)dz + \int_c^d \zeta(z)dz + \int_d^a \zeta(z)dz. \tag{96}$$

On the other hand, we have that

$$(d,c) = (a + \omega_2, b + \omega_2), \qquad (b,c) = (a + \omega_1, d + \omega_1), \tag{97}$$

which imply

$$\int_b^c \zeta(z)dz = \int_{a+\omega_1}^{d+\omega_1} \zeta(z)dz = \int_a^d \zeta(t+\omega_1)dt = \int_a^d (\zeta(t) + 2\eta_1)dt = \int_a^d \zeta(t)dt + 2\eta_1(d-a)$$

$$= \int_a^d \zeta(t)dt + 2\eta_1\omega_2, \tag{98}$$

and

$$\int_c^d \zeta(z)dz = -\int_d^c \zeta(z)dz = -\int_{a+\omega_2}^{b+\omega_2} \zeta(z)dz = -\int_a^b \zeta(t+\omega_2)dt = -\int_a^b (\zeta(t) + 2\eta_2)dt$$

$$= -\int_a^b \zeta(t)dt - 2\eta_2(b-a) = -\int_a^b \zeta(t)dt - 2\eta_2\omega_1. \tag{99}$$

Inserted in (96) we get

$$2\pi i = 2\eta_1\omega_2 - 2\eta_2\omega_1, \tag{100}$$

which is the assertion.

Finally, it is also possible to prove (but we omit the proof) that

$$\omega_1 = J_1, \qquad \eta_1 = \frac{1}{2}J_2, \tag{101}$$

$$\omega_2 = \tilde{J}_1, \qquad \eta_2 = \frac{1}{2}\tilde{J}_2, \tag{102}$$

where $\tilde{J}_k$ are obtained by replacing $\gamma$ with $\beta$, so we can further rewrite the Legendre's relation in the form

$$\tilde{J}_1 J_2 - \tilde{J}_2 J_1 = 2\pi i. \tag{103}$$

It is in this form that the Legendre's relation has its deepest geometrical meaning.

*5.2. Riemann's Bilinear Relations and Intersections*

To interpret the above result, we pass to a more general situation and consider an oriented Riemann surface $\Sigma$ of genus $g > 0$ (the surface of a donut with $g$ holes or a sphere with $g$ handles). It is well known, from Reference [102], Lemma 3.14, that we can realize it from a $4g$-gone with edges, in clockwise order, $a_1 b_1 a_1^{-1} b_1^{-1} \cdots a_g b_g a_g^{-1} b_g^{-1}$, where $a^{-1}$ means the edge is oriented counterclockwise, gluing each edge with its "inverse", according to the orientation. Any $a_j$ and any $b_j$ is a closed curve and they satisfy the intersection product relations

$$a_j \cdot a_k = b_j \cdot b_k = 0, \qquad a_j \cdot b_k = -b_k \cdot a_j = \delta_{jk}. \tag{104}$$

These curves are homotopically distinct and are a (canonical) basis for the first homology group $H_1(\Sigma, \mathbb{Z}) \simeq \mathbb{Z}^{2g}$. Given a closed 1-form $\omega$ on $\Sigma$, one defines its periods relative to the given basis as

$$\pi_j^a(\omega) := \oint_{a_j} \omega, \qquad \pi_j^b(\omega) := \oint_{b_j} \omega. \tag{105}$$

Given two closed forms $\omega_1$ and $\omega_2$, using Stokes's theorem and closures, it is not difficult to prove that the following identity is always true

$$\int_\Sigma \omega_1 \wedge \omega_2 = \sum_{j=1}^g \left[ \pi_j^a(\omega_1)\pi_j^b(\omega_2) - \pi_j^b(\omega_1)\pi_j^a(\omega_2) \right]. \tag{106}$$

**Proof.** Assume that $\Sigma$ is a closed oriented Riemann surface of genus $g$. Now, let us notice that if we cut $\Sigma$ along the representatives $a_j$ and $b_j$, we get the $4g$-gon $\tilde{\Sigma}$ associated with the word $a_1 b_1 a_1^{-1} b_1^{-1} \cdots a_j b_j a_j^{-1} b_j^{-1} \cdots a_g b_g a_g^{-1} b_g^{-1}$, see Figure 17

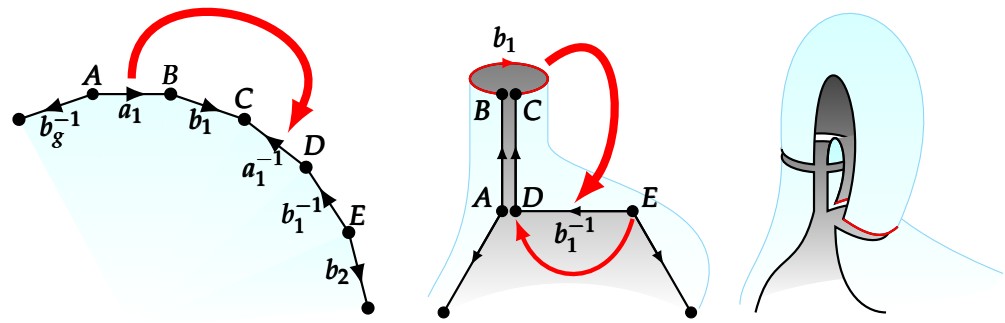

**Figure 17.** Riemann surfaces from the fundamental $4g$-gon.

This polygon is simply connected, so any closed form restricted to it is exact. In particular, this is true for $\omega_2$, which has the form $\omega_2 = d\phi_2$, and for $\phi_2$, a well defined

function on $\tilde{\Sigma}$ (but not on $\Sigma$). More precisely, we can fix a point $o$ in $\tilde{\Sigma}$ (for example the center). Since the polygon is convex, any $x \in \tilde{\Sigma}$ is connected to $o$ by a segment, and we can write

$$\phi_2(x) \equiv \int_o^x \omega_2, \tag{107}$$

where the integration is intended along the segment. Now, if $\omega_2$ is exact and $\omega_1$ is closed (indeed exact too on the simply connected domain), then $\omega_1 \wedge \omega_2$ is also exact. Indeed, the 1-form $\Omega = -\phi_2 \omega_1$ has the property that $d\Omega = \omega_1 \wedge \omega_2$, and is well defined on $\tilde{\Sigma}$. Finally, since as sets

$$\Sigma - \bigcup_{j=1}^g (a_j \bigcup b_j) = \tilde{\Sigma} - \partial\tilde{\Sigma}, \tag{108}$$

together with $\bigcup_{j=1}^g (a_j \bigcup b_j)$ and $\partial\tilde{\Sigma}$, are subsets of vanishing measure of $\Sigma$ and $\tilde{\Sigma}$, respectively, we have

$$\int_\Sigma \omega_1 \wedge \omega_2 = \int_{\Sigma - \bigcup_{j=1}^g (a_j \bigcup b_j)} \omega_1 \wedge \omega_2 = \int_{\tilde{\Sigma} - \partial\tilde{\Sigma}} \omega_1 \wedge \omega_2 = \int_{\tilde{\Sigma}} \omega_1 \wedge \omega_2 = \int_{\tilde{\Sigma}} d\Omega = \int_{\partial\tilde{\Sigma}} \Omega$$

$$= -\int_{\partial\tilde{\Sigma}} \phi_2 \omega_1 = -\int_{\partial\tilde{\Sigma}} \omega_1(x) \int_o^x \omega_2, \tag{109}$$

where the main trick has been to rewrite the integral over a region where the integrand is exact, in order to be able to use the Stokes theorem. Since the boundary $\partial\tilde{\Sigma}$ is the (oriented) union of the edges of the polygon, we then have

$$\int_\Sigma \omega_1 \wedge \omega_2 = -\sum_{j=1}^g \left( \int_{a_j} \omega_1(x) \int_o^x \omega_2 + \int_{b_j} \omega_1(x) \int_o^x \omega_2 + \int_{a_j^{-1}} \omega_1(x) \int_o^x \omega_2 + \int_{b_j^{-1}} \omega_1(x) \int_o^x \omega_2 \right). \tag{110}$$

Now, consider the integrals

$$I_j^a \equiv \int_{a_j} \omega_1(x) \int_o^x \omega_2 + \int_{a_j^{-1}} \omega_1(x) \int_o^x \omega_2. \tag{111}$$

The forms $\omega_1$ and $\omega_2$ are well-defined on $\Sigma$, so they take the same values on the identified points of $a_j$ and $a_j^{-1}$. If $y(x)$ is the point on $a_j^{-1}$ which identifies with $x \in a_j$ in gluing $\tilde{\Sigma}$ to form $\Sigma$, then, noticing that $a_j^{-1}$ is oriented in the opposite sense of $a_j$, we can write

$$I_j^a = \int_{a_j} \omega_1(x) \left( \int_o^x \omega_2 - \int_o^{y(x)} \omega_2 \right) = \int_{a_j} \omega_1(x) \left( \int_o^x \omega_2 + \int_{y(x)}^o \omega_2 \right). \tag{112}$$

The sum of the two integrals of $\omega_2$ is just equivalent to the integral along a curve from $y(x)$ to $x$, which is homotopically equivalent to $b_j$ traveling in the opposite orientation (from $a_j^{-1}$ to $a_j$); see Figure 18.

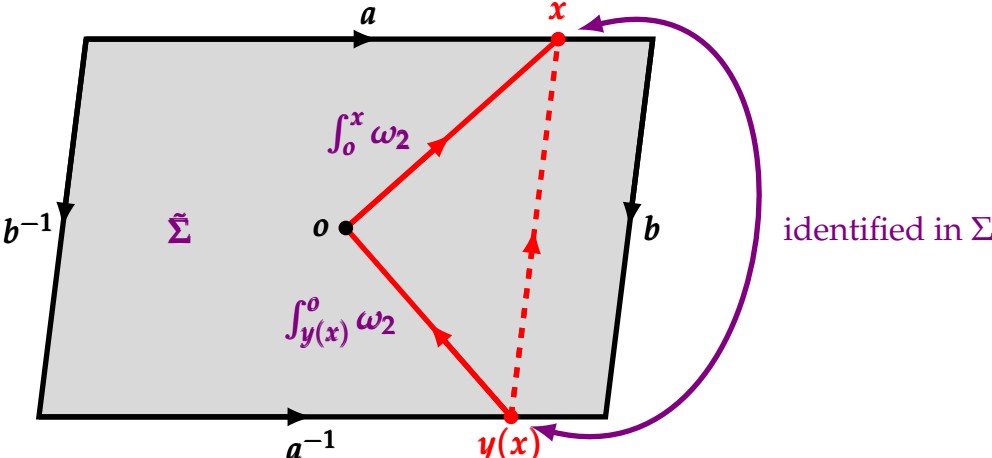

**Figure 18.** Construction of the integral $I_b$ in case $g = 1$.

Therefore,

$$\int_o^x \omega_2 + \int_{y(x)}^0 \omega_2 = -\int_{b_j} \omega_2 = -\pi_j^b(\omega_2) \tag{113}$$

is independent of $x$. So, we can write

$$I_j^a = -\pi_j^b(\omega_2) \int_{a_j} \omega_1(x) = -\pi_j^b(\omega_2)\pi_j^a(\omega_1). \tag{114}$$

Let us now see what changes for the remaining integrals

$$I_j^b \equiv \int_{b_j} \omega_1(x) \int_o^x \omega_2 + \int_{b_j^{-1}} \omega_1(x) \int_o^x \omega_2. \tag{115}$$

Reasoning as above, if $y(x)$ is the point on $b_j^{-1}$ identified with $x$ on $b_j$ when constructing $\Sigma$ from $\tilde{\Sigma}$, we can write

$$I_j^b = \int_{b_j} \omega_1(x) \left( \int_o^x \omega_2 + \int_{y(x)}^o \omega_2 \right). \tag{116}$$

Now, the path $y(x) \to o \to x$ on $\Sigma$ becomes a path homotopic to $a_j^{-1}$ traveled in the opposite direction, so it is homotopic to $a_j$ and, therefore

$$\int_o^x \omega_2 + \int_{y(x)}^o \omega_2 = \pi_j^a(\omega_2). \tag{117}$$

Thus,

$$I_j^b = \int_{b_j} \omega_1(x)\pi_j^a(\omega_2) = \pi_j^a(\omega_2)\pi_j^b(\omega_1). \tag{118}$$

Since

$$\int_\Sigma \omega_1 \wedge \omega_2 = -\sum_{j=1}^g (I_j^a + I_j^b), \tag{119}$$

we get the assertion. $\quad\square$

This wonderful formula, known as Riemann's bilinear relations, gives (103) as a very particular case and provides the geometric interpretation we were looking for. Indeed, it leads to the concept of intersection number between closed forms, or, as we will see in a moment, between cohomology cocycles. The transversality of the homology curves is now replaced by the fact that in any given point, $\omega_1 \wedge \omega_2$ is non-degenerate as a bilinear (or multilinear) map on the tangent space to $\Sigma$. The directions where $\omega_1$ vanishes define locally a curve. Similarly, $\omega_2$ defines locally a second curve. These two curves meet transversally in the given point. These curves are essentially the Poincaré duals of the forms, and this naive description shows its role in providing a geometrical interpretation of the intersection number between forms. When $\Sigma$ is provided by a Riemannian metric $\boldsymbol{h}$, then, in local coordinates, we can write

$$\omega_1 \wedge \omega_2 = \omega_{1j}\omega_{2k}dx^j \wedge dx^k = \omega_{1j}\omega_{2k}\frac{1}{\sqrt{h}}\varepsilon^{jk}\sqrt{h}dx^1 \wedge dx^2, \tag{120}$$

where $\varepsilon^{jk}$ is the usual Levi-Civita tensor density, and Einstein's summation convention is adopted. We set

$$w_2^j \equiv \omega_{2k}\frac{1}{\sqrt{h}}\varepsilon^{jk}, \tag{121}$$

the Hodge dual of $\omega_2$ and identify $\sqrt{h}dx^1 \wedge dx^2 = dV$ as the volume element on $\Sigma$. Finally, since closed forms are locally exact, in a small enough neighborhood we can write $\omega_1 = d\phi_1$ for a potential $\phi_1$. Notice that the Poincaré dual of $\omega_1$ is thus locally represented by the level curves $\phi_1 = const$. In the given neighborhood, we get

$$\omega_1 \wedge \omega_2 = \partial_j\phi_1 w_2^j \, dV = \partial_j(\phi_1 w_2^j) \, dV, \tag{122}$$

where we used the fact that $\omega_2$ closed implies $\partial_j w_2^j = 0$. If as a small neighborhood we take a small rectangle $Q$ with edges $a, b, c, d$ with $a$ and $c$ along level curves for $\phi_1$ (with values $\kappa_a$ and $\kappa_c$), while $b$ and $d$ are orthogonal, then

$$\int_Q \omega_1 \wedge \omega_2 = \int_Q \partial_j(\phi_1 w_2^j) \, dV = \kappa_c \int_c n_j^c w_2^j ds + \kappa_a \int_a n_j^a w_2^j ds, \tag{123}$$

$n_j^c$ and $n_j^a$ being the external normals to $c$ and $a$, respectively. This integral is thus determined by the flux of the vector field $w_2^j$, Hodge dual to $\omega_2$, through the equipotential surfaces (lines) of $\omega_1$. Notice that even in an extremal case when the rectangle can be extended until wrapping the surface so that[6] $a = c$, then we would get

$$\int_Q \omega_1 \wedge \omega_2 = (\kappa_c - \kappa_a) \int_c n_j^c w_2^j ds, \tag{124}$$

in general with $\kappa_c - \kappa_a \neq 0$ despite $c$ and $a$ coinciding, since $\phi_1$ is not monodromic and takes a different value after wrapping $\Sigma$ along a closed loop. Indeed, in this case $(\kappa_c - \kappa_a)$ is nothing but the period of $\omega_1$ along the closed path. In general, it is not possible to extend even $c$ to a closed path, and our naive analysis then stops here. However, it is clear that its correct formulation is given exactly by the Riemann's bilinear formula, which generalizes this vision in terms of fluxes (of the Hodge dual of $\omega_2$) through the Poincaré dual of $\omega_1$.

Now, it is time to be a little bit more precise. If we modify $\omega_1$ by an exact form $df$, its periods do not change[7]. On the other hand, since $\omega_2$ is closed too,

$$(\omega_1 + df) \wedge \omega_2 = \omega_1 \wedge \omega_2 + d(f\omega_2), \tag{125}$$

and by Stokes's theorem and the fact that $\Sigma$ has no boundary, we see that the exact term does not contribute to the l.h.s. integral. The same holds if we modify $\omega_2$ by an exact form.

This means that this formula depends only on the cohomology classes of $\omega_1$ and $\omega_2$. From (125), we also see that if we consider the cohomology class of $\omega_1 \wedge \omega_2$, then, it depends only on the classes $[\omega_1]$ and $[\omega_2]$ and not specifically on the forms. So, this gives rise to a well-defined bilinear antisymmetric map

$$\cup : H^1(\Sigma) \times H^1(\Sigma) \longrightarrow H^2(\Sigma), \ ([\omega_1],[\omega_2]) \mapsto [\omega_1] \cup [\omega_2] := [\omega_1 \wedge \omega_2], \tag{126}$$

called the *cup product*. The above formula can then be written as

$$[\omega_1] \cdot [\omega_2] \equiv \int_\Sigma [\omega_1] \cup [\omega_2] = \sum_{j=1}^{g} \left[ \pi_j^a([\omega_1]) \pi_j^b([\omega_2]) - \pi_j^b([\omega_1]) \pi_j^a([\omega_2]) \right]. \tag{127}$$

The l.h.s. formula is called the *intersection product* of the two classes (or forms). This is because it is in a sense a dual of the intersection formula in homology. Indeed, let us consider the dual basis to the canonical one, that is, a set of closed one-forms $\omega_j^a$, $\omega_j^b$, $j = 1,\dots,g$ such that

$$\pi_j^a(\omega_k^a) = \delta_{jk}, \qquad \pi_j^a(\omega_k^b) = 0, \tag{128}$$

$$\pi_j^b(\omega_k^a) = 0, \qquad \pi_j^b(\omega_k^b) = \delta_{jk}. \tag{129}$$

Thus,

$$[\omega_j^a] \cdot [\omega_k^a] = [\omega_j^b] \cdot [\omega_k^b] = 0, \tag{130}$$

$$[\omega_j^a] \cdot [\omega_k^b] = -[\omega_k^b] \cdot [\omega_j^a] = \delta_{jk}, \tag{131}$$

which has exactly the same form of the intersection products among the elements of the homology basis. This is our final interpretation of the Legendre's bilinear Formula (103): it gives intersection product between the first kind and the second kind differentials. In general, after selecting a basis $b_j$, $j = 1,\dots,2g$ of $H^1(\Sigma)$, the matrix $\boldsymbol{c}$ with elements

$$c_{jk} = b_j \cdot b_k \tag{132}$$

is invertible. Any other element $[\omega]$ in $H^1(\Sigma)$ can be written as

$$[\omega] = \sum_{j=1}^{2g} k^j b_j, \tag{133}$$

with $k_J \in \mathbb{C}$. Therefore,

$$[\omega] \cdot b_k = \sum_{j=1}^{2g} k^j c_{jk} \tag{134}$$

so that

$$k^j = \sum_{k=1}^{2g} [\omega] \cdot b_k c^{kj}, \tag{135}$$

where $c^{kj}$ are the matrix elements of the inverse matrix $\boldsymbol{c}^{-1}$. On the other hand, for $u = a,b$, the periods of $[\omega]$ are

$$\pi_j^u([\omega]) = \sum_{k=1}^{2g} k^k \pi_j^u(b_k) = \sum_{l=1}^{2g} \sum_{k=1}^{2g} ([\omega] \cdot b_l) c^{lk} \pi_j^u(b_k). \tag{136}$$

Thus, we see that after the intersection product is given, for a path $\gamma$ homologically equal to $\gamma = \sum_{j=1}^{g}(m_a^j a_j + m_b^j b_j)$, $m_u^j \in \mathbb{Z}$, our integral is given by

$$\int_\gamma \omega = \sum_{u=a,b}\sum_{j=1}^{g}\sum_{l=1}^{2g}\sum_{k=1}^{2g} m_u^j([\omega]\cdot b_l)c^{lk}\pi_j^u(b_k). \tag{137}$$

So, the only integrals we have to compute are $\pi_j^u(b_k)$, which we can call the "master integrals." Notice that this expression resembles the decomposition Formulas (45).

At this point we may wonder how much all this can be generalized to higher dimensions. To get some insight, let us now present some very general considerations, without any pretense of completeness, but just with the aim to outline the main concepts one would need to deepen, referring to the appropriate literature.

**Remark 1.** *In the literature, Riemann's bilinear relations usually are not exactly (106), but rather one of its consequences ([102], Th.12.2): if $\omega_j$, $j = 1, 2$ are two holomorphic forms on a genus g closed oriented Riemann surface $\Sigma$, and $a_j, b_j$, $j = 1, \ldots, g$ a canonical basis of $H_1(\Sigma)$, then the following relations are true:*

$$\sum_{j=1}^{g}(\pi_j^a(\omega_1)\pi_j^b(\omega_2) - \pi_j^a(\omega_2)\pi_j^b(\omega_1)) = 0 \tag{138}$$

$$\mathrm{Im}(\sum_{j=1}^{g}\overline{\pi_j^b(\omega_1)}\pi_j^a(\omega_1)) > 0, \tag{139}$$

*where $\bar{z}$ means the complex conjugate of z.*

### 5.3. A Twisted Version

We want now to consider a situation that is a little bit more involved. In place of real or complex valued forms over our Riemann surface $\Sigma_g$, let us consider twisted differential forms taking value in a line bundle $\mathcal{L}$. This is a bundle whose fibres are copies of $\mathbb{C}_* = \mathbb{C} - \{0\}$. To it we can associate a dual line bundle $\mathcal{L}^\vee$, which has the following property: if $\psi$ and $\psi^\vee$ are sections of $\mathcal{L}$ and $\mathcal{L}^\vee$, respectively, then, $\psi(z)\psi^\vee(z) \equiv f(z)$ defines a function $f : \Sigma_g \to \mathbb{C}$. In particular, if $u(z)$ is a section of $\mathcal{L}^\vee$, then $u(z)^{-1}$ is a section of $\mathcal{L}$. With this in mind, we say that $\phi_L$ is a $\mathcal{L}$-valued twisted $k$-form if for any given section $u$ of $\mathcal{L}^\vee$, then $u\phi_L$ is a $\mathbb{C}$-valued $k$-form. We will call $\phi_L$ a left $k$-form, and the space of left $k$-forms is

$$\Omega^k(\Sigma_g, \mathcal{L}) = \Omega^k(\Sigma_g, \mathbb{C}) \otimes \mathcal{L}, \tag{140}$$

and we write

$$\Omega_L^* = \bigoplus_{k=0}^{2} \Omega^k(\Sigma_g, \mathcal{L}). \tag{141}$$

We say that $\mathcal{L}$ is flat if it admits a global section. When it happens, it is clear that $\mathcal{L}^\vee$ is also flat. Assume that $u$ is a global section of $\mathcal{L}^\vee$: we call it *the twist*. Given a twist, on $\Omega_L^*$ we can put a structure of ring as follows. For $\phi_1, \phi_2 \in \Omega_L^*$, we define

$$\phi_1 \wedge_u \phi_2 := u\phi_1 \wedge \phi_2, \tag{142}$$

in harmony with the condition $(u\phi_1) \wedge (u\phi_2) = u(\phi_1 \wedge_u \phi_2)$, required by the mapping $\Omega^k(\Sigma_g, \mathcal{L}) \to \Omega^k(\Sigma_g, \mathbb{C})$, $\phi \mapsto u\phi$. Notice that $\Omega^0(\Sigma_g, \mathcal{L})$ are the sections of $\mathcal{L}$. $u$ also allows us to introduce a connection on $\mathcal{L}$ that induces a covariant derivative

$$\nabla_\omega : \Omega^k(\Sigma_g, \mathcal{L}) \longrightarrow \Omega^{k+1}(\Sigma_g, \mathcal{L}), \quad , i.e., \quad d(u\phi_L) = u\nabla_\omega \phi_L. \tag{143}$$

Concretely,

$$\nabla_\omega \phi \equiv d + \omega \wedge \phi, \qquad \omega = \frac{du}{u}. \tag{144}$$

It is a flat connection, since it satisfies $\nabla_\omega^2 = 0$ as one can easily check. This allows us to say that $\phi$ is closed if $\nabla_\omega \phi = 0$ and exact if $\phi = \nabla_\omega \psi$ for $\psi$ a left form of lower rank. Notice that these definitions imply

$$\nabla_\omega(\phi_L \wedge_u \tilde{\phi}_L) = (\nabla_\omega \phi_L) \wedge_u \tilde{\phi}_L + (-1)^k \phi_L \wedge_u (\nabla_\omega \tilde{\phi}_L) - \omega \wedge \phi_L \wedge_u \tilde{\phi}_L, \tag{145}$$

if $\phi_L$ has degree $k$.

In a similar way, we can define the right forms as the twisted forms with value in $\mathcal{L}^\vee$, simply by exchanging $\mathcal{L} \leftrightarrow \mathcal{L}^\vee$, $u \leftrightarrow 1/u$, $\omega \leftrightarrow -\omega$. A right form will be called $\phi_R$, when necessary, to distinguish it from a left form. Of course, a right form is closed if $\nabla_{-\omega}\phi_R = 0$ and exact if $\phi_R = \nabla_{-\omega}\psi_R$.

A first interesting fact is that the standard wedge product between a left form and a right form satisfies

$$\begin{aligned}
d(\phi_L \wedge \phi_R) &= d\phi_L \wedge \phi_R + (-1)^{k_L}\phi_L \wedge d\phi_R = (\nabla_\omega - \omega\wedge)\phi_L \wedge \phi_R + (-1)^{k_L}\phi_L \wedge (\nabla_{-\omega} + \omega\wedge)\phi_R \\
&= \nabla_\omega \phi_L \wedge \phi_R + (-1)^{k_L}\phi_L \wedge \nabla_{-\omega}\phi_R - \omega \wedge \phi_L \wedge \phi_R + \omega \wedge \phi_L \wedge \phi_R \\
&= \nabla_\omega \phi_L \wedge \phi_R + (-1)^{k_L}\phi_L \wedge \nabla_{-\omega}\phi_R,
\end{aligned} \tag{146}$$

where we used that $\phi_L \wedge \omega = (-1)^{k_L}\omega \wedge \phi_L$. In particular, if $\phi_L$ and $\phi_R$ are covariantly closed, then, $\phi_L \wedge \phi_R$ is closed in the canonical sense. Now, suppose that $\phi_L$ and $\phi_R$ are 1-forms, well defined over the Riemann surface $\Sigma_g$. As in the previous sections, we can cut it to $\tilde{\Sigma}_g$, the fundamental $4g$-gone, which is simply connected and, indeed, star shaped with respect to an internal point $o$. Therefore, for any $z \in \tilde{\Sigma}_g$, we can take the segment $[o, z]$ and consider the function

$$\psi_L(z) = e^{-\int_o^z \omega(t)} \int_o^z e^{\int_o^t \omega(s)} \phi_L(s). \tag{147}$$

It is well defined on $\tilde{\Sigma}_g$ and satisfies

$$\nabla_\omega \psi_L(z) = \phi_L(z). \tag{148}$$

Formula (146) then shows that

$$d(\psi_L \phi_R) = \phi_L \wedge \phi_R, \tag{149}$$

so $\phi_L \wedge \phi_R$ is exact on $\tilde{\Sigma}_g$. Proceeding as in the proof of (106), we can write

$$\int_{\Sigma_g} \phi_L \wedge \phi_R = \int_{\tilde{\Sigma}_g} \phi_L \wedge \phi_R = \int_{\tilde{\Sigma}_g} d(\psi_L \phi_R) = \int_{\partial\tilde{\Sigma}_g} \psi_L \phi_R = \int_{\partial\tilde{\Sigma}_g} e^{-\int_o^z \omega(t)} \int_o^z e^{\int_o^t \omega(s)} \phi_L(t)\phi_R(z), \tag{150}$$

and following the same exact procedure, we finally get [74]

$$\int_{\Sigma_g} \phi_L \wedge \phi_R = \sum_{j=1}^g \left[ \left( \int_{a_j} e^{\int_o^t \omega(s)} \phi_L(t) \right) \left( \int_{b_j} e^{-\int_o^t \omega(s)} \phi_R(t) \right) \right.$$
$$\left. - \left( \int_{b_j} e^{\int_o^t \omega(s)} \phi_L(t) \right) \left( \int_{a_j} e^{-\int_o^t \omega(s)} \phi_R(t) \right) \right], \tag{151}$$

which is the twisted version of (106).



### 5.4. Perversities and Thimbles

Now, we can address the main point. While we previously related our integrals just to the cohomology of the Riemann surface defined by the polynomial, the situation is more involved in the general case of a Feynman integral. The point is that the section $u$ is represented by the Baikov polynomial $\mathcal{B}$ as $u = \mathcal{B}^\gamma$. We assume $\gamma \in \mathbb{R} - \mathbb{Z}$. This section has zeros (or singular points if $\gamma < 0$) in the zeros of $\mathcal{B}$, so it does not define a global section of a line bundle over the whole domain of the polynomial, but just in the complement of its zeros. Furthermore, if the integration domain goes towards infinity, it must be chosen so that the integral converges (so along paths where $\mathcal{B}$ goes to 0 or to $\infty$, according to whether $\gamma$ is positive or negative). This looks like it says that the form becomes trivial around infinity, to make sense to the integral. This leads us to think that the right (co)homology to be considered is the one of $\mathbb{C}^n - Z$ relative to infinity, where $Z$ is the zero locus of the polynomial. This is indeed the result of Francis Pham, who proved that the right homology is the relative homology $H_n(\mathbb{C}^n - Z, B)$, $B$ being a relative neighborhood of infinity: $B = \{x \in \mathbb{C}^n \,||\mathcal{B}(x)| > N\}$ where $N$ is "large enough" [112,113].

The point is that one has to manage the homological tools when singularities are present. Let us consider as an illustrative example the integral

$$\int_\Gamma \frac{dz}{(z^3 + z)^\alpha},$$ (152)

$\alpha \notin \mathbb{Z}$. The integral has to converge, so $\Gamma$ has to avoid the zeros of the polynomial, but can reach the infinity if $\alpha > \frac{1}{3}$ and the opposite if $\alpha < \frac{1}{3}$. The zeros of the polynomials are branch points, and we have to fix branch cuts to fix the integrand. Moreover, the map $\mathcal{B} : \mathbb{C} \to \mathbb{C}$ has two critical points, which are the points where the gradient vanishes. Let us briefly see why this could be interesting. If we want to have an intuition on the cohomology to be used, we can start thinking that it is not expected to depend on the exact value of $\alpha$. Therefore, after rewriting the integral in the form

$$\int_\Gamma e^{-\alpha \log(z^3 + z)} dz,$$ (153)

we can estimate it for very large $\alpha$. This can be done by means of the saddle point method, so that the integral is dominated by the saddle points of the exponent, which are indeed the critical points of $\mathcal{B}$. Depending on $\alpha$, one has then to follow the steepest descent or ascent lines, in such a way that the integrand goes to zero. For example, if we write $\mathcal{B}(\zeta_+) = |\mathcal{B}(\zeta_+)| e^{i\phi_+}$, where $\zeta_+$ is a critical point, then

$$\mathcal{B}(\zeta_+)^{-\alpha} = e^{-\alpha \log |\mathcal{B}(\zeta_+)| - i\alpha\phi_+},$$ (154)

and we see that the path passing from the saddle point is conveniently chosen so that the phase is constant so as to avoid oscillations. Thus, the best choice is the paths having the constant phase, fixed by the critical point. Then, one chooses the direction such to move from the critical value to infinity or to zero, depending on the case if the path is allowed to reach infinity or zero (in our example, depending on whether $\alpha > 1/3$ or $\alpha < 1/3$). Notice that, having fixed the phase, choosing a direction to move away from the critical point, the absolute value of the integrand moves monotonically toward 0 or $\infty$. Furthermore, since we have to move along these paths, we have to choose the cuts so that they do not coincide with (part of) the constant-phase integration paths. Now that we have got hint, let us go back to the explicit case in some detail.

The branch points are $z_0 = 0$, $z_\pm = \pm i$. The critical points solve $3z^2 + 1 = 0$, so that they are $\zeta_\pm = \pm \frac{i}{\sqrt{3}}$. If we coordinatize the target space of $\mathcal{B}$ with $t$, then the there are three images of the branch points and the critical points in the target space:

$$t_0 \equiv \mathcal{B}(z_i) = 0, \qquad \tau_\pm = \mathcal{B}(\zeta_\pm) = \pm \frac{2i}{3\sqrt{3}}.$$ (155)

The phases determined by $\tau_{\pm}$ are thus $\phi_{\pm} = \pm\frac{\pi}{2}$. Therefore, we choose the cuts with phase 0. Writing $z = x + iy$, this means

$$\mathcal{B}(z) = x^3 - 3y^2x + x + i(-y^3 + 3x^2y + y), \tag{156}$$

so that the cuts are defined by $-y^3 + 3x^2y + y = 0$, $x^3 - 3y^2x + x > 0$. These are

$$y = 0, \qquad x > 0; \tag{157}$$
$$y = \sqrt{3x^2 + 1}, \qquad x < 0;$$
$$y = -\sqrt{3x^2 + 1}, \qquad x < 0, \tag{158}$$

see Figure 19. In a similar way, we can determine the descent/ascent paths from the critical points.

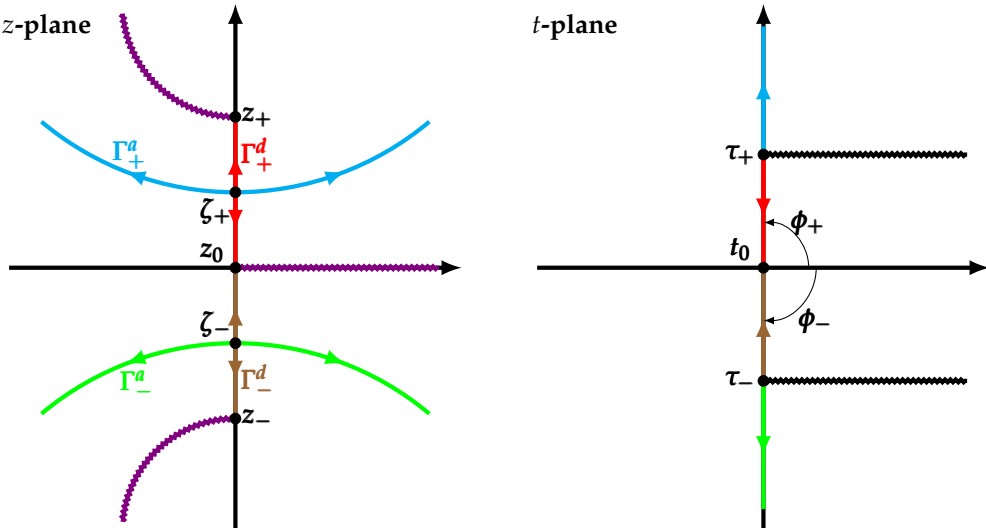

**Figure 19.** Cuts, ascent paths and descent paths.

Through $\zeta_{\pm}$ the paths $\Gamma_{\pm}$ are defined by the phase $\phi_{\pm} = \pm\frac{\pi}{2}$, which correspond to $x^3 - 3y^2x + x = 0$, $\pm(-y^3 + 3x^2y + y) > 0$. These give for $\Gamma_+$ the paths

$$x = 0, \qquad 0 < y < 1, \qquad \text{(descent path, } \Gamma_+^d\text{);} \tag{159}$$
$$y = \sqrt{\frac{x^2+1}{3}}, \qquad \text{(ascent path, } \Gamma_+^a\text{),} \tag{160}$$

for $\Gamma_-$ the paths

$$x = 0, \qquad -1 < y < 0, \qquad \text{(descent path, } \Gamma_-^d\text{);} \tag{161}$$
$$y = -\sqrt{\frac{x^2+1}{3}}, \qquad \text{(ascent path, } \Gamma_-^a\text{).} \tag{162}$$

These are curves from infinity to infinity around the cuts (ascent paths), or from a zero to another one (descent paths). They are called *Lefschetz thimbles*, for a reason that will become clear soon. Now we want to understand a little better what we have done just inspired by saddle point methods. To this hand, let us consider $\mathcal{B}$ as a map defining a fibration over $\mathbb{C}$. The fiber over $t$ is generically given by a set $\mathcal{B}^{-1}(t)$ of 3 points. The exceptions are the critical points where the polynomial has double roots, so the fiber is made by just 2 points. For the generic fiber at $t$ (which is zero dimensional), the cohomology is $H^0(\mathcal{B}^{-1}(t), \mathbb{Z}) = \mathbb{Z}^3$, and is invariant when $t$ varies, so we are in trouble if we move to a critical point. Therefore, to manage this difficulty, it is convenient to restrict this "fibration"

of cohomology groups to $\mathbb{C}_\tau \equiv \mathbb{C} - \{\tau_+, \tau_-\}$ and look at what happens if we move around the critical values. To see this, notice that if $\lambda := e^{\frac{2\pi i}{3}}$ is a primitive third root of 1, then

$$\mathcal{B}^{-1}(t) = \begin{pmatrix} \sqrt[3]{\frac{t}{2} + \sqrt{\frac{t^2}{4} + \frac{1}{27}}} + \sqrt[3]{\frac{t}{2} - \sqrt{\frac{t^2}{4} + \frac{1}{27}}} \\ \lambda\sqrt[3]{\frac{t}{2} + \sqrt{\frac{t^2}{4} + \frac{1}{27}}} + \lambda^2\sqrt[3]{\frac{t}{2} - \sqrt{\frac{t^2}{4} + \frac{1}{27}}} \\ \lambda^2\sqrt[3]{\frac{t}{2} + \sqrt{\frac{t^2}{4} + \frac{1}{27}}} + \lambda\sqrt[3]{\frac{t}{2} - \sqrt{\frac{t^2}{4} + \frac{1}{27}}} \end{pmatrix}, \tag{163}$$

from which we can easily see what happens to its (co)homology when we move on a loop around the critical points: since $\sqrt{\frac{t^2}{4} + \frac{1}{27}}$ vanishes at $t = \tau_\pm$, it changes sign in moving along the loop, and we see that the upper component is invariant, while the lowest components are interchanged. Therefore, on $V_t \equiv H^0(\mathcal{B}^{-1}(t), \mathbb{Z}) = \mathbb{Z}^3$, the monodromy action (the action of moving around the points) around $\tau_\pm$ is given by the matrices

$$M_\pm = \begin{pmatrix} 1 & 0 & 0 \\ 0 & 0 & 1 \\ 0 & 1 & 0 \end{pmatrix}. \tag{164}$$

The badness of the fibration is measured by these (here coincident) monodromy matrices. The point is that $V_t$ has fixed points given by the eigenspaces of $M_\pm$ corresponding to the eigenvalue 1. These define the spaces of invariants

$$V^{M_\pm} \simeq \mathbb{Z} \begin{pmatrix} 1 \\ 0 \\ 0 \end{pmatrix} \oplus \mathbb{Z} \begin{pmatrix} 0 \\ 1 \\ 1 \end{pmatrix}. \tag{165}$$

Their complement are the spaces of coinvariants

$$V_{M_\pm} \simeq \mathbb{Z} \begin{pmatrix} 0 \\ 1 \\ -1 \end{pmatrix}. \tag{166}$$

On them, the monodromies act as minus the identity. The strategy for capturing the right cohomology is to restrict the fibration of cohomologies, keeping only the coinvariant parts and the information of the action of the monodromies. Notice that these are irreducible non-trivial representations of $\pi_1(\mathbb{C}_\tau)$, the first homotopy group (generated by the loops around the critical values). This "almost fibration" with specification of monodromy constrained structures at singular points substantially describes what is called a perverse sheaf. The family of coinvariant (co)homology elements can be then interpreted as follows. In the homology of $\mathcal{B}^{-1}(t)$, they can be thought as

$$0 \cdot \left( \sqrt[3]{\frac{t}{2} + \sqrt{\frac{t^2}{4} + \frac{1}{27}}} + \sqrt[3]{\frac{t}{2} - \sqrt{\frac{t^2}{4} + \frac{1}{27}}} \right) + p \cdot \left( \lambda\sqrt[3]{\frac{t}{2} + \sqrt{\frac{t^2}{4} + \frac{1}{27}}} + \lambda^2\sqrt[3]{\frac{t}{2} - \sqrt{\frac{t^2}{4} + \frac{1}{27}}} \right)$$

$$- p \cdot \left( \lambda^2\sqrt[3]{\frac{t}{2} + \sqrt{\frac{t^2}{4} + \frac{1}{27}}} + \lambda\sqrt[3]{\frac{t}{2} - \sqrt{\frac{t^2}{4} + \frac{1}{27}}} \right) \tag{167}$$

with $p \in \mathbb{Z}$. The generator is for $p = 1$. Then when $t$ varies from a singular point to infinity (or to 0), we see that these points collapse at the critical point, collapsing to zero also as an element of $H_0(\mathcal{B}^{-1}(t), \mathbb{Z})$, since we get the same point with total coefficient $1 - 1 = 0.$[8] These correspond to the thimbles and, are called vanishing cycles. Notice that the pair of points are 0-dimensional cycles. For a polynomial in $n$ variables, one gets that the preimages of $t$ are generically $\mathcal{B}^{-1}(t) \simeq TS^{n-1}$, the total space of the tangent bundle of

a sphere, which are homologically equivalent to spheres $S^{n-1}$. The (perverse) cohomology generated as such is represented by a union of spheres when $t$ varies, for example, from a critical value to $\infty$ along a line $\gamma$. The result is homotopic to a cylinder $S^{n-1} \times \gamma$, whose face at the end corresponding to the critical value collapses to a point. This may be thought to have the shape of a thimble.

Of course, a rigorous version of this construction in a general setting requires the use of more sophisticated mathematical tools.

## 6. Some General Constructions

In this section, we want to discuss some relations between Feynman integrals, homologies and cohomologies. Even if we avoid entering into details, we illustrate the main ideas on how to deal with singularities by passing to simplistic (co)homologies, while our original problem remains the one of computing integrals of differential forms. Therefore, we will start by recalling some relations among the main (co)homologies we will need to consider in the successive subsections. We will not provide notions but rather only rough ideas and some literature. However, we hope to make this section at least readable.

### 6.1. On Cohomologies and de Rham Theorem

In general, for a given space, there can be several associated homologies depending on the structures one is considering. We are of course interested in the de Rham cohomology, which is well defined on smooth manifolds as the quotient of closed forms with exact forms. However, there are other (co)homologies that are of interest for concrete calculations. For the definitions reported below and any further lecture, we refer to [102,114].

#### 6.1.1. Singular (Co)homology

One of the most general is *singular homology*. A $k$-simplex $\Delta_k$ is the smallest convex subset of $\mathbb{R}^N$ (N > k) generated by $k+1$ points in general position (so it has dimension $k$). If $x_j$, $j = 0, \ldots, k$ are the generating points, it is convenient to use the notation $\Delta_k = \{x_0, \ldots, x_k\}$. Then, the $j$-th face of the simplex is the $(k-1)$-simplex $\Delta_{k-1}^j = \{x_0 \ldots, \hat{x}_j, \ldots, x_n\}$, where the *hat* means omission of the point. If $X$ is a topological space, then a *singular $k$-simplex* in $X$ is a pair $(\Delta_k, \phi)$, where $\phi : \Delta_k \to X$ is a continuous map. A singular $k$-chain $c_k$ is a formal finite combination of singular $k$-simplexes $c_k = \sum_\alpha a_\alpha (\Delta_{k,\alpha}, \phi_\alpha)$, $a_\alpha \in \mathbb{K}$, where $\mathbb{K}$ is an abelian group, for us, $\mathbb{K} = \mathbb{Z}, \mathbb{Q}, \mathbb{R}, \mathbb{C}$. They form the space $C_k^{sing}(X)$ of singular chains. The boundary of a singular simplex $(\Delta_k, \phi)$ is the singular $(k-1)$-chain

$$\partial(\Delta_k, \phi) = \sum_{j=0}^{k} (-1)^j (\Delta_{k-1}^j, \phi|_{\Delta_{k-1}^j}). \tag{168}$$

This defines by $\mathbb{K}$-linear extension, a sequence of $\mathbb{K}$-linear maps

$$\partial_k : C_k^{sing}(X) \longrightarrow C_{k-1}^{sing}(X), \quad c_k \mapsto \partial c_k. \tag{169}$$

It follows that $\partial_{k-1} \circ \partial_k = 0$ and one can define the *singular homology groups*

$$H_k^{sing}(X, \mathbb{K}) = \frac{\ker(\partial_k)}{\mathrm{Im}(\partial_{k+1})}. \tag{170}$$

Similarly, we can define a singular $k$-cochain as a linear map

$$\psi^k : C_k^{sing}(X) \longrightarrow \mathbb{K}, \tag{171}$$

that form the $\mathbb{K}$-linear space $C^k_{sing}(X)$, on which acts the coboundary operator

$$\delta^k : C^k_{sing}(X) \longrightarrow C^{k+1}_{sing}(X), \quad ,i.e., \quad \delta^k \psi^k(c_{k+1}) \equiv \psi^k(\partial_{k+1}c_{k+1}), \ \forall c_{k+1} \in C^{sing}_{k+1}(X). \tag{172}$$

From these, one defines the *singular cohomology groups*

$$H^k_{sing}(X, \mathbb{K}) = \frac{\ker(\delta^k)}{\mathrm{Im}(\delta^{k-1})}. \tag{173}$$

**Remark 2.** *Replacing simplexes with hypercubes, one gets the singular cubic (co)homology. However, this gives a non-normalized cohomology [102] (also called a generalized cohomology), that is, a cohomology that does not satisfy the condition $H_k(p) = 0$ for $k > 0$ and $p$ a point.*

### 6.1.2. Simplicial (co)homology

This is less general and is first defined for simplicial complexes. A simplicial complex $\Xi$ is a collection of simplexes such that if a simplex is in the complex, then all its faces are also in, and, moreover, two simplexes in the collection can intersect in one and only one common sub-face of a given dimension. It is called finite if composed of a finite number of simplexes, while it is locally finite if for any point there is a small neighbourhood intersecting just a finite number of simplexes. Given a simplicial complex $\Xi$, one defines its *simplicial k-chains* as the formal combinations $c_k = \sum_\alpha a_\alpha \Delta_{k,\alpha}$, $a_\alpha \in \mathbb{K}$, which define the space $C^{simp}_k(\Xi)$. Defining the boundary

$$\partial \Delta_k = \sum_{j=0}^{k} (-1)^j \Delta^j_{k-1}, \tag{174}$$

we get linear maps

$$\partial^s_k : C^{simp}_k(\Xi) \longrightarrow C^{simp}_{k-1}(\Xi), \quad c_k \mapsto \partial c_k, \tag{175}$$

which satisfy $\partial_{k-1} \circ \partial_k = 0$ and allow to define the simplicial homology groups

$$H^{simp}_k(\Xi, \mathbb{K}) = \frac{\ker(\partial^s_k)}{\mathrm{Im}(\partial^s_{k+1})}. \tag{176}$$

Similarly, we can define a simplicial *k*-cochain as a linear map

$$\xi^k : C^{simp}_k(\Xi) \longrightarrow \mathbb{K}, \tag{177}$$

that forms the $\mathbb{K}$-linear space $C^k_{simp}(\Xi)$, on which the following the coboundary operator acts:

$$\delta^k_s : C^k_{simp}(\Xi) \longrightarrow C^{k+1}_{simp}(\Xi), \quad ,i.e., \quad \delta^k_s \xi^k(c_{k+1}) \equiv \xi^k(\partial_{k+1}c_{k+1}), \ \forall c_{k+1} \in C^{simp}_{k+1}(\Xi). \tag{178}$$

From these, one defines the *simplicial cohomology groups*

$$H^k_{simp}(\Xi, \mathbb{K}) = \frac{\ker(\delta^k_s)}{\mathrm{Im}(\delta^{k-1}_s)}. \tag{179}$$

This construction can then be extended to the case of a topological space $X$ that admits triangulations. If a given triangulation $T$ can be homotopically deformed to become

homeomorphic to a simplicial complex $\Xi_T$, then we can define the simplicial homology groups of $X$ as

$$H_k^{simp}(X, \mathbb{K}) = H_k^{simp}(\Xi_T, \mathbb{K}), \qquad H_{simp}^k(X, \mathbb{K}) = H_{simp}^k(\Xi_T, \mathbb{K}). \tag{180}$$

These are well defined, since it can be proved that the resulting groups are independent of the triangulation $T$. An important result is that the following isomorphisms (for spaces admitting simplicial homology) hold true:

$$H_k^{sing}(X, \mathbb{K}) \simeq H_k^{simp}(X, \mathbb{K}), \qquad H_{sing}^k(X, \mathbb{K}) \simeq H_{simp}^k(X, \mathbb{K}). \tag{181}$$

For a proof, see [102,114].

### 6.1.3. De Rham Theorem

Now we can state the following important result ([102,114–118]):

**Theorem 1** (de Rham Theorem)**.** *If $X$ is a smooth manifold and $\mathbb{K} = \mathbb{R}, \mathbb{C}$, then*

$$H_k^{deRham}(X, \mathbb{K}) \simeq H_k^{sing}(X, \mathbb{K}) \simeq H_k^{simp}(X, \mathbb{K}), \tag{182}$$

$$H_{deRham}^k(X, \mathbb{K}) \simeq H_{sing}^k(X, \mathbb{K}) \simeq H_{simp}^k(X, \mathbb{K}). \tag{183}$$

An important remark for us is that the same theorem can be extended by weakening the hypothesis and assuming that $X$ admits singularities and more in general a (Whitney) stratification or the structure of a stratifold. See [119] and references therein.

This result allows us to move to simplicial (co)homology to simplify abstract constructions as we did in Section 5. However, it is also helpful for simplifying certain computations like the one of Betti number and the Euler characteristic. Here we recall that the Betti numbers of $X$ are the numbers

$$b_k = \dim H_{deRham}^k(X, \mathbb{K}), \tag{184}$$

while the Euler characteristic of $X$ is the number

$$\chi(X) = \sum_{j=0}^{N} (-1)^j b_j, \tag{185}$$

where $N = \dim X$.

Finally, we remark that further cohomologies can be introduced, like for example relative (co)homologies $H_k(X, B, \mathbb{K})$, cellular (co)homologies, Čech and sheaf (see [102]), or generalized (co)homologies, such as $K$-theories or (co)bordisms. We demand any further reading to the standard literature.

### 6.2. Cup Products and Intersections on Smooth Manifolds

In the example of closed one-forms on a compact closed Riemann surface $\Sigma$, we have seen that the cup product $\cup : H^1(\Sigma) \times H^1(\Sigma) \to H^2(\Sigma)$ can be interpreted as the intersection product between two one-forms, in a sense, dual to the intersection product among homological curves (1-cycles), which is probably more intuitive. Now, we want to understand how much this notion can be generalized. To this hand we will refer to the lecture in [120].

Let $\mathcal{M}$ be an $m$-dimensional oriented compact smooth manifold. Furthermore, we assume that $S_1$ and $S_2$ are two smooth submanifolds of dimensions $m_1$ and $m_2$, respectively. We say that they have transversal intersection if $N = S_1 \cap S_2 \neq \varnothing$, and for any $x \in N$ the union of the embeddings of $\iota_j : T_x S_j \hookrightarrow T_x \mathcal{M}$ is the whole $T_x \mathcal{M}$. In particular, this implies that $m_1 + m_2 - m = m_N \geq 0$, which is the dimension of the intersection. We have to take

into account the orientation. We say that the intersection is oriented on a given connected component $\bar{N}$ of $N$ if it is possible to choose a basis $\boldsymbol{v}_1, \ldots \boldsymbol{v}_{m_N} \in T_x N$, $x \in \bar{N}$ such that:

- We can complete it to an oriented basis $\boldsymbol{w}_1, \ldots, \boldsymbol{w}_{m_1-m_N}, \boldsymbol{v}_1, \ldots \boldsymbol{v}_{m_N} \in T_x S_1$ according to the orientation of $S_1$;
- We can complete it to an oriented basis $\boldsymbol{v}_1, \ldots \boldsymbol{v}_{m_N}, \boldsymbol{z}_1, \ldots, \boldsymbol{z}_{m_2-m_N} \in T_x S_2$ according to the orientation of $S_2$;
- $\boldsymbol{w}_1, \ldots \boldsymbol{w}_{m_1-m_N}, \boldsymbol{v}_1, \ldots \boldsymbol{v}_{m_N}, \boldsymbol{z}_1, \ldots, \boldsymbol{z}_{m_2-m_N}$ is an oriented basis for $T_x M$ according to the orientation of $M$.

See Figure 20 for a pictorial representation.

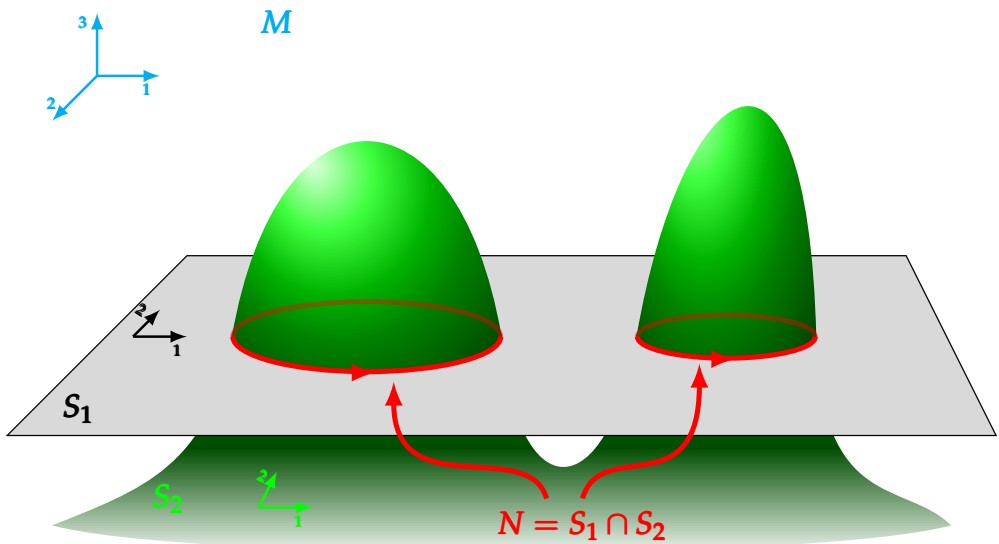

**Figure 20.** Intersection between two oriented surfaces $S_1$ and $S_2$ in a three-dimensional oriented manifold $M$.

It is clear that it depends only on the connected component and not on the specific point. Therefore, $\boldsymbol{v}_1, \ldots \boldsymbol{v}_{m_N}$ defines the orientation of the intersection. We can consider the homology class $[S_1 \cap S_2]$ associated with $N$, in the sense of simplicial homology.[9] It can be shown that it depends only on the homology classes of $S_j$, so it gives a map $H_{m_1}(M, \mathbb{Z}) \times H_{m_2}(M, \mathbb{Z}) \to H_{m_1+m_2-m}(M, \mathbb{Z})$, each time $m_1 + m_2 - m \geq 0$.

In particular, when $m_1 + m_2 = m$, $N$ is just the union of a set of isolated points. This set is finite since $M$ is compact and defines an element of $H_0(M, \mathbb{Z}) \simeq \mathbb{Z}$ (since $M$ is connected). Thus, we can associate the intersection product with a number representing such an element. This can be done as follows. If $\sigma(x)$ is the sign associated with $x \in N$, we set

$$S_1 \cdot S_2 := \sum_{x \in N} \sigma(x). \tag{186}$$

It is well defined, since $N$ is finite, and it can be shown to depend on the homology classes only. So, it shows the connection with the intersection product we met in the case of a Riemann surface.

However, let us stay general for the moment. There is another interesting map to be considered in simplicial (co)homology in order to go ahead: the *cap product*,

$$\frown : H^j(M, \mathbb{Z}) \times H_m(M, \mathbb{Z}) \longrightarrow H_{m-j}(M, \mathbb{Z}), \quad (\mu, u) \mapsto \mu \frown u, \tag{187}$$

defined in such the way that for any given $\nu \in H^{m-j}(M,\mathbb{Z})$, we must have $\nu(u \frown \mu) \equiv (\nu \cup \mu)(u)$. Then, let $[M] \in H_m(M,\mathbb{Z})$ be the class generated by $M$, also called the *fundamental class*. Fixing $u = [M]$, we get a map

$$\star : H^j(M,\mathbb{Z}) \longrightarrow H_{m-j}(M,\mathbb{Z}), \quad \mu \mapsto \star\mu := \mu \frown [M]. \tag{188}$$

Then, this theorem holds ([121], Th. 1.1.3):

**Theorem 2** (Poincaré duality). *If M is a connected, closed, and oriented topological manifold, then $\star : H^j(M,\mathbb{Z}) \longrightarrow H_{m-j}(M,\mathbb{Z})$ is an isomorphism for all integers j.*

The most important consequence of this theorem, at least as regards the applications we are interested in, is the following theorem ([120], Th. 1.1):

**Theorem 3.** *Assume M is a compact, connected, closed smooth manifold and let $S_j$ smooth submanifolds of dimension $m_j$, $j = 1, 2$ that (up to an homological deformation) have transversal intersection. Then, it holds that*

$$\star[S_1] \cup \star[S_2] = \star[S_1 \cap S_2]. \tag{189}$$

For a proof, see [120]. This theorem thus states that cup and cap products are dual under the Poincaré map. So, it provides a remarkable generalization of the results discussed for Riemann surfaces. We described it in the case of simplicial homology, but it clearly works as well if we tensorize with $\mathbb{K} = \mathbb{R}, \mathbb{C}$, and then we can use the isomorphism $H^*_{simp}(M,\mathbb{K}) \simeq H^*_{dR}(M,\mathbb{K})$ to get it for the case of the de Rham cohomology (where the cup product is induced by the wedge product), which better matches with the previous section.

However, we are not yet satisfied this, since we had to make use of smoothness, but in considering the general integrals we are interested in, we generically have to consider manifolds with singularities, which may not be homologous to smooth manifolds, so we need to explore further generalizations.

*6.3. Lefschetz Theorems, Hodge-Riemann Bilinear Relations and Perversities*

We want to go further in our tour of intersection theory by looking at finer structures. We will follow mainly [122], indicating some further lectures explicitly when necessary.

First, let us go back to the case of Riemann surfaces. These are complex curves of complex dimension 1, and smooth real surfaces of real dimension 2. They can be embedded in a projective space; for example, the elliptic curves can be embedded in $\mathbb{P}^2$ as the zero locus of a quartic or cubic polynomial, such as

$$y^2 = P_4(x), \tag{190}$$

in local non homogeneous coordinates. If in the above equation in $\mathbb{P}^2$ we replace $P_4$ with a polynomial $P_d$ of degree $d = 2g$ or $d = 2g - 1$; we get an hyperelliptic complex curve of genus $g$. They are therefore *algebraic varieties* that are zeros of polynomial functions in some real or complex space. When an algebraic variety can be embedded in a projective space $\mathbb{P}^N$ for some $N$, then it is called a *projective variety*. Furthermore, we can notice that the one forms on a Riemann surface $\Sigma$, in local complex coordinates $z$ and $\bar{z}$, have the general form

$$\omega = f(z,\bar{z})dz + g(z,\bar{z})d\bar{z}, \tag{191}$$

with differential

$$d\omega = (\partial_z g - \partial_{\bar{z}} f)dz \wedge d\bar{z}. \tag{192}$$

Any two form is

$$\Omega = \lambda(z, \bar{z}) dz \wedge d\bar{z}, \tag{193}$$

and it is closed, $d\Omega = 0$ for dimensional reasons. It is cohomologically trivial if it has the form (192). $\Sigma$ is also a Kähler manifold, which means the following. It is always possible to put on it a Hermitian metric

$$G = g_{z\bar{z}} dz \otimes d\bar{z}, \tag{194}$$

that defines at any point $p$ of $\Sigma$ a sesquilinear map

$$G_p : T_p \Sigma \times T_p \Sigma \longrightarrow \mathbb{C}, \tag{195}$$

such that

$$G_p(\alpha\xi, \zeta) = G(\xi, \bar{\alpha}\zeta) \tag{196}$$

for any $\alpha \in \mathbb{C}$. Its real part defines a Riemannian metric on $\Sigma$, while its imaginary part defines a two form

$$K = -\frac{i}{2} g_{z\bar{z}} dz \wedge d\bar{z}, \tag{197}$$

which is a closed form, called the Kähler form. The same construction can be made in higher dimensions, with the only difference being that in general the Kähler form is not automatically closed. When it is, the complex manifold is said to be a Kähler manifold. It is possible to prove that for a Kähler manifold $M$, in a local coordinate patch $U$, it is always possible to find a real-valued function $\mathcal{K} : U \to \mathbb{R}$, called the *Kähler potential*, such that

$$g_{z\bar{z}} = \partial_z \partial_{\bar{z}} \mathcal{K}. \tag{198}$$

An example is the complex projective manifolds $\mathbb{P}^N$, that, in non-homogeneous coordinates $z_1, \ldots, z_N$, admit the Kähler potential

$$\mathcal{K}_{FS} = \log \left( 1 + \sum_{j=1}^{N} |z_j|^2 \right), \tag{199}$$

generating the Fubini study Hermitian metric

$$G_{FS} = \frac{1}{(1 + \sum_{j=1}^{N} |z_j|^2)^2} \left[ (1 + \sum_{j=1}^{N} |z_j|^2) \sum_{k=1}^{N} dz_k \otimes d\bar{z}_k - \sum_{j,k=1}^{N} z_j \bar{z}_k dz_j \otimes d\bar{z}_k \right]. \tag{200}$$

Any projective manifold is a Kähler manifold, since it inherits a Kähler structure after restriction of the Fubini study of the ambient space $\mathbb{P}^N$ in which it is embedded.

The cohomology of a Kähler manifold of complex dimension $m$ has a nice structure, called a Hodge structure. For any $k = 0, 1, \ldots, 2m$, the cohomology group $H^k(M, \mathbb{C})$ admits a direct decomposition

$$H^k(M, \mathbb{C}) = \bigoplus_{p+q=k} H^{p,q}(M, \mathbb{C}), \tag{201}$$

where $p$ and $q$ run in $\{0, \ldots, m\}$, and the elements of $H^{p,q}(M, \mathbb{C})$ are represented by closed forms of the type

$$\omega_{p,q} = \sum_{j_1, \ldots, j_p, k_1, \ldots, k_q} \omega_{j_1 \ldots j_p k_1 \ldots k_q} dz^{j_1} \wedge \cdots \wedge dz^{j_p} \wedge d\bar{z}^{k_1} \wedge \cdots d\bar{z}^{k_q}. \tag{202}$$

For example, for connected Riemann surfaces of genus $g$ we have

$$H^0(\Sigma, \mathbb{C}) \simeq \mathbb{C}, \tag{203}$$

$$H^1(\Sigma, \mathbb{C}) = H^{1,0}(\Sigma, \mathbb{C}) \oplus H^{0,1}(\Sigma, \mathbb{C}) \simeq \mathbb{C}^g \oplus \mathbb{C}^g, \tag{204}$$

$$H^2(\Sigma, \mathbb{C}) = H^{2,0}(\Sigma, \mathbb{C}) \oplus H^{1,1}(\Sigma, \mathbb{C}) \oplus H^{0,2}(\Sigma, \mathbb{C}) \simeq 0 \oplus \mathbb{C} \oplus 0. \tag{205}$$

In particular, $H^{1,0}(\Sigma, \mathbb{C})$ is generated by $g$ holomorphic forms, which for an hyperelliptic curve in $\mathbb{P}^2$

$$y^2 = P(z) \tag{206}$$

are

$$\omega_j = \frac{z^j dz}{y}, \quad j = 0, \dots, g - 1. \tag{207}$$

$H^2(\Sigma, \mathbb{C})$ is generated by the Kähler form, which is also real valued, $K \in H^{1,1}(M, \mathbb{C}) \cap H^2(M, \mathbb{R})$.

Now, unfortunately, it is time to become more abstract. Given a cohomology group of any kind, let us assume that it as a finitely generated abelian group $H$ over $\mathbb{Z}$, and, following [122], let use the notation $H_{\mathbb{K}} = H \otimes_{\mathbb{Z}} \mathbb{K}$, $\mathbb{K} = \mathbb{Q}, \mathbb{R}, \mathbb{C}$. We say that $H$ carries a *pure Hodge structure* of weight $k$ if there is a decomposition of abelian groups

$$H_{\mathbb{C}} = \bigoplus_{p+q=k} H^{p,q}, \quad , i.e., \quad \overline{H^{p,q}} = H^{q,p}. \tag{208}$$

An equivalent formulation is to say that $H$ admits a decreasing Hodge Filtration, that is, a family of abelian subgroups $F^p$ of $H_{\mathbb{C}}$ with the properties

$$H_{\mathbb{C}} = F^0 \supset F^1 \supset \cdots \supset F^p \supset F^{p+1} \supset \cdots, \tag{209}$$

$$F^p \cap \overline{F^{k+1-p}} = 0, \tag{210}$$

$$F^p \oplus \overline{F^{k-p}} = H_{\mathbb{C}}. \tag{211}$$

Indeed, the equivalence is read by the relations

$$H^{p,q} = F^p \cap \overline{F^q}, \tag{212}$$

$$F^p = \bigoplus_{j \geq p} H^{j,k-j}. \tag{213}$$

On $H_{\mathbb{C}}$, there is a real action of $S^1 \subset \mathbb{C}_*$ given by $\rho(s)a = s^{p-q}a$ for $a \in H^{p,q}$ and extended by $\mathbb{R}$ linearity. In particular, $\rho(i) = w$ is called the Weyl map. A *polarization* on the pure Hodge structure is a real bilinear form $Q$ on $H_{\mathbb{R}}$, which is invariant under the action of $S^1$ and such that

$$B(a, b) := Q(a, wb) \tag{214}$$

is symmetric and positive definite. After extending $Q$ over $H_{\mathbb{C}}$, one gets that $i^{2k+p-q}Q(a, \bar{a}) > 0$ for a non vanishing $a \in H^{p,q}$.

Now, consider a non singular algebraic projective variety of dimension $m$, $X \subset \mathbb{P}^N$ (for instance a projective manifold). Let $H$ be a generic hyperplane in $\mathbb{P}^N$. Then, $H \cap X$ defines an element of $H_{n-2}(X, \mathbb{Z})$, whose Poincaré dual is $\eta \in H^2(X, \mathbb{Z})$, see [121]. We then define the Lefschetz map

$$L : H^k(X, \mathbb{Q}) \longrightarrow H^{k+2}(X, \mathbb{Q}), \quad a \mapsto L(a) = \eta \cup a. \tag{215}$$

The degree $k$ primitive vector spaces are the subspaces $P^k \subseteq H^k(X, \mathbb{Q})$ defined by $P^k := \ker L^{m-k+1}$. Finally, on $H^k(X, \mathbb{Q})$ let us define the quadratic form

$$Q_L^{(k)}([\omega], [\omega']) := (-1)^{\frac{k(k+1)}{2}} \int_X \eta^{m-k} \wedge \omega \wedge \omega'. \tag{216}$$

Then, $H^k(X, \mathbb{Z})$ is a pure Hodge structure of weight $k$ and $P^k$ is a rational hodge structure of weight $k$. Then we have the following important properties ([122], Th.5.2.1).

**Theorem 4** (Hard Lefschetz Theorem). *With the above notations, the maps*

$$L^k : H^{m-k}(X, \mathbb{Q}) \longrightarrow H^{m+k}(X, \mathbb{Q}) \tag{217}$$

*are isomorphisms.*

**Theorem 5** (Primitive Lefschetz decomposition). *For $0 \leq k \leq m$, one has the decomposition*

$$H^k(X, \mathbb{Q}) = \bigoplus_{j \geq 0} L^j P^{k-2j}. \tag{218}$$

*Moreover, for any given $j$, $V^j \equiv L^j P^{k-2j}$ is a pure Hodge substructure of weight $k$, and for any $j \neq j'$, $V^j$ and $V^{j'}$ are $Q_L^{(k)}$-orthogonal.*

**Theorem 6** (Hodge-Riemann bilinear relations). *For any $0 \leq k \leq m$, $Q_L^{(k)}$ is a polarization for the pure Hodge structure $P_{\mathbb{R}}^k$. In particular, after complexification,*

$$i^{p-q} Q_L^{(k)}([\omega], [\bar{\omega}]) > 0, \qquad \forall [\omega] \in P^k \cap H^{p,q}(X, \mathbb{C}), \tag{219}$$

*so it is strictly positive definite.*

This result seems to give us a very broad generalization of the properties we met for integrals along one-dimensional cycles on a Riemann surface. However, again, it is not enough, since we had to require for $X$ to be a nonsingular variety. The simplest example is $\mathbb{C}_* = \mathbb{C} \backslash \{0\}$. Its first cohomology group is obviously one-dimensional, and this fact is incompatible with the existence of a pure Hodge structure. Indeed, assuming $H^{2k+1}$ has a pure Hodge structure, then the relation $H^{p,q} = \overline{H^{q,p}}$ implies

$$\dim(H^{2k+1}) = 2 \sum_{j=0}^{k} \dim(H^{j,2k+1-j}) \in 2\mathbb{N}, \tag{220}$$

so it must be even-dimensional.

Since we have to work with general Feynman integrals, we cannot avoid working with singular varieties, so we need to overcome this difficulty. Luckily, this has been done throughout the years, passing from Weyl, Serre, Grothendieck, and finally solved by Deligne [123–125], who proved the existence of a *mixed Hodge structure* on the $H^k(X, \mathbb{Q})$ when $X$ is an arbitrary algebraic complex variety. This means that, given $X$ of complex dimension $m$ and $0 \leq k \leq 2m$, it is always possible to find two filtrations:

- A decreasing filtration $F^*$, called the *Hodge filtration*, such that

$$H^k(X, \mathbb{Q}) = F^0 \supseteq F^1 \supseteq \cdots \supseteq F^m \supseteq F^{m+1} = \{0\}; \tag{221}$$

- An increasing filtration $W_*$, called the *Weight filtration*, such that

$$\{0\} = W_{-1} \subseteq W_0 \subseteq \cdots \subseteq W_{2k-1} \subseteq W_{2k} = H^k(X, \mathbb{Q}); \tag{222}$$

which have the following property. On the complexification of each graded part of $W_*$, $F^*$ induces a decreasing filtration that endows the pieces

$$gr_j^W := W_j \otimes \mathbb{C}/W_{j-1} \otimes \mathbb{C} \tag{223}$$

with a pure Hodge structure of weight $j$. We do not intend to delve further into these concepts and direct the reader to [123] for details.

At this point, in order to define intersection theory, one has essentially to tackle the problem of how to define intersections of cycles that may contain singularities or pass through a singular region. To get an idea, let us follow the nice exposition in [121], Chapter 1. Let us consider a two-dimensional pinched torus $W$, which looks something like a würstel bent over and glued at the tips just at one point $p$. It is clear that on this surface, a loop around the hole is a non-trivial generator for $H_1(W)$ (but pass through the singularity), while a loop $L$ wounding the "würstel" shrinks down to a point when moved to the gluing point $p$ (the singularity), Figure 21.

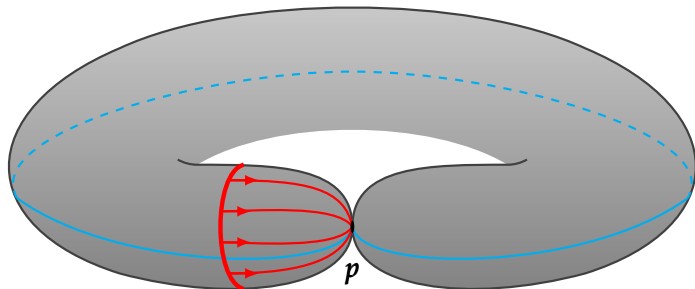

**Figure 21.** A singular torus.

So, $H^1(W)$ is one-dimensional and the pure Hodge structure is lost. A relevant observation is that if we think at the singular point $p$ as a zero-dimensional representative of $H_0(W)$, it meets the loop $L$ non-transversally. This situation repeats for higher dimensional varieties, where singularities can be much worse. A strategy is to construct more refined (co)homologies but starting from the simplicial ones and restricting the allowed (co)chains. One idea is first to generate a homology starting from a simplicial one where only triangulations giving locally finite chains are allowed (so that any point admits a small boundary that intersects only a finite number of cells of the chain). This construction defines the *Borel-Moore homology groups* $H_k^{BM}(X, \mathbb{Z})$ [121]. Tensoring with $\mathbb{C}$ gives the isomorphism $H_k^{BM}(X, \mathbb{C}) \simeq H_c^k(X, \mathbb{C})$ of the compactly supported cohomology, whose cochains (or closed $k$-forms) have compact support. Assume now that singularities define an increasing filtration of $X$, that is, a sequence of subvarieties

$$\varnothing = X_{-1} \subseteq X_0 \subseteq X_1 \ldots \subseteq X_{2m-2} = X_{2m-1} \subseteq X_{2m} = X, \tag{224}$$

where the $X_j$, $j < 2m$ are the real $j$-dimensional singular sub loci, called the *strata*, of which the largest one has at least real codimension 2, and $X \backslash X_{2m-2}$ is required to be dense in $X$. With this in mind, we can further refine the homology by allowing only triangulations, giving locally finite $k$-chains whose support intersects all strata transversally. We call $H_k^{tr}(X)$ the obtained homology group. A theorem by McCrory ([121], Th. 2.3.2, [126]), then provides the isomorphism $H_k^{tr}(X) \simeq H^{2m-k}(X, \mathbb{Z})$. Assuming that (in a suitable sense, see [121]) $X$ is oriented, we can use cap product with the fundamental class $[X]$ to map $H^{2m-k}(X, \mathbb{Z})$ into $H_k^{BM}(X, \mathbb{Z}) \simeq H_k^{lf}(X)$, the last term being the locally finite simplicial homology. However, since $X$ is singular, the Poincaré map is no longer an isomorphism. Therefore, $H_k^{lf}(X)$ and $H_k^{tr}(X)$ are not isomorphic, the reason essentially being that, in general, the locally finite cycles cannot be deformed, so to have transversal intersection with all strata. For example, this is what happened in our example of the "glued würstel".

Therefore, one needs to do a final refinement, in order to get a correct intersection theory together with a good Poincaré duality. This is obtained by restricting which chains can meet the singular locus, if not transversally.

The first important request is that any chain has to meet $X_{2m-2}$ transversally. For algebraic varieties of complex dimension 1, that is all. For example, in the case of our glued würstel, this just implies that the loop around the hole is forbidden. Thus, $H_1(Z) = 0$ and the perverse homology of the singular surface is the same as for the two dimensional sphere. Indeed, the singularity in this case is non-normal, in the sense that if we take a small neighborhood of the singular point $p$ and keep the point out, the neighborhood separates into two disconnected components. We can normalize the surface by adding the lacking point separately to each one of the two disconnected parts. In other words we unglue the würstel, which becomes a usual one, with the topology of $S^2$. These kinds of singularity are thus simply solved by normalization, and for the case of complex dimensions, two things remain relatively simple.

However, for higher-dimensional $X$, one has to consider also intersection with higher codimensional singularities. Here is where one introduces a *perversity*, [127–135]:

**Definition 1.** *A perversity is a function*

$$p : \mathbb{Z}_{\geq 2} \longrightarrow \mathbb{N}, \tag{225}$$

*satisfying the conditions*

$$p(2) = 0, \qquad p(j) \leq p(j+1) \leq p(j) + 1, \quad j \geq 2. \tag{226}$$

In particular, $p_t(j) = j - 2$ is called the top perversity, while $p_0(j) = 0$ is the zero perversity. Furthermore, $p_{lm}(j) = \lfloor (j-2)/2 \rfloor$, $j > 2$, is called the lower-middle perversity, and $p_{um} = p_t - p_{lm}$ is the upper-middle perversity.

Given a perversity $p$, an allowed $k$-chain $\sigma$ is a locally finite chain whose support $|\sigma|$ satisfies

$$\dim(|\sigma| \cap X_{2m-j}) \geq k - j + p(j), \tag{227}$$
$$\dim(|\partial\sigma| \cap X_{2m-j}) \geq k - j + p(j) - 1, \tag{228}$$

for all $j \geq 2$. The homology groups generated as such are called the *BM intersection homology groups* with perversity $p$ of $X$, $I^{BM}H_k^{(p)}(X, \mathbb{Z})$. If in place of locally finite chains one starts with finite chains, then one obtains the *homology groups* with perversity $p$, $IH_k^{(p)}(X, \mathbb{Z})$.

With these notions, one has the following version of the Poincaré duality ([121], Th.2.6.1):

**Theorem 7** (Poincaré duality). *If $X$ is an algebraic manifold of complex dimension $m$, whose singularities define a filtration as above, and $p$ and $\bar{p}$ are two complementary perversities, which means $p + \bar{p} = p_t$, then there is a a non-degenerate bilinear pairing*

$$\frown: IH_k^{(p)}(X, \mathbb{Z}) \times I^{BM}H_{2m-k}^{(\bar{p})}(X, \mathbb{Z}) \longrightarrow \mathbb{Q}. \tag{229}$$

This theorem provides the necessary definition of ($\mathbb{Q}$-valued) intersection number. A construction that proves homological invariance and independence of choices, as well as calculability, passes through sheaf theory and its generalizations. Analyzing such a formulation goes too far beyond the aim of our discussion. However, we want just to add some remarks for those with at least a basic knowledge on sheaf theory (a good introduction is found in [121], Chapter 4). In the sheaf-theoretical formulation, a particular role is played by Local Systems, which are locally constant sheaves, that is, any sheaf $\mathcal{L}$ over $X$, such that its local restriction to a small neighborhood $U_p$ of any given $p \in X$, $\mathcal{L}_p \simeq M_p$, a given $A$-module. Thus, it turns out to be necessary to work with *twisted homologies*, which are

homologies taking value in a Local System. This leads to work with complexes of sheaves. A main difficulty in doing these is that in such framework often diagrams commute only up to homotopies and not identically. To overcome this problem, one introduces the category $K(X)$, which is essentially the same of the one of complex of sheaves, up to identifications by homotopy. The problem is that one of the main computational instruments, that is, short exact sequences, are lost in passing from the category $C(X)$ of complexes of sheaves to $K(X)$. A remedy consists in finding a way to map short exact sequences in $C(X)$ into *distinguished triangles*, from which it is also possible to construct long exact sequences. This leads to the language of derived categories ([121], Chapter 4). This way, one can construct the intersection homology groups $IH^k(X)$.

In place of embarking into the formidably difficult program of analyzing such constructions further, we move on to illustrating a possible strategy to be applied to the more specific situation of computing integrals of the form (2), suggested by M. Kontsevich in a private communication to P. Mastrolia, T. Damour and S.L. Cacciatori, at IHES on January 2020.

*6.4. From Feynman Integrals to Intersection Theory*

Let us consider (2). In general, we assume that $\nu_j = 0$ for the sake of simplicity, and $\gamma \in \mathbb{C}\backslash\mathbb{Z}$.[10] Consider the map

$$B : \mathbb{C}_*^M \longrightarrow \mathbb{C} \supset \mathbb{C}_*. \tag{230}$$

This map has critical regions with critical values $t_\alpha \in \mathbb{C}$, $\alpha = 1, \ldots, n$. For each $t$, we can consider the algebraic variety

$$X_t = B^{-1}(t) \cap \mathbb{C}_*^M, \tag{231}$$

with $t \in \tilde{\mathbb{C}} \equiv \mathbb{C}^* - \{t_\alpha\}_{\alpha=1}^n$. For these $t$, $X_t$ is smooth and admits a compactification $\bar{X}_t$, with $\bar{X}_t$ projective, $X_t \subset \bar{X}_t$ open embedding, and $\bar{X}_t\backslash X_t$ is a normal crossing divisor, which means that at any point of $X_t$ it is described by local equations $\prod_{j=1}^r z_j$. Thus, starting from $H^{M-1}(X_t, \mathbb{Z})$, we get a (Deligne) weight filtration, starting from $\bar{X}_t$ (with $H^{M-1}(\bar{X}_t)$ of lowest weight 0). When $t$ moves in $\tilde{\mathbb{C}}$, we get a variation of mixed Hodge structures on $H^{M-1}(X_t, \mathbb{Z})$.

Now, these variations of mixed Hodge structures define a perverse sheaf over $\tilde{\mathbb{C}}$. Luckily, on complex curves, they can be understood quite easily; we refer to [136] for a nice presentation. A perverse sheaf is not exactly a sheaf but has properties similar to a sheaf, as it can be glued by local data. Furthermore, away from a singular point, it is effectively a sheaf, more precisely a Local System, so on $x$ non-singular, we can think of the relative stalks as a finitely generated module $V$. Around a singular point, one has to add a monodromy rule around the singular point $t_\alpha$:

$$T_\alpha : V \to V, \qquad \alpha = 1, \ldots, n. \tag{232}$$

This is the effect of our variation of mixed Hodge monodromy. Let $V^{T_\alpha} \subseteq V$ nd the submodule of invariant elements (eigenelements). The exact sequence $V^{T_\alpha} \hookrightarrow V \to V_{T_\alpha}$ defines the coinvariant module $V_{T_\alpha}$. Since the monodromy action depends only on the first homotopy group, we can see monodromies as representing the action of the first homotopy group $\pi_1(\tilde{\mathbb{C}})$. On $V_{T_\alpha}$, we get an irreducible representation of $\pi_1(\mathbb{C}^* - \{t_\alpha\}_{\alpha=1}^n)$ on a variation of a pure Hodge structure. Thus, it is endowed with a polarization via the Hodge–Riemann bilinear relations. This polarization is symmetric or skew-symmetric, according to the parity of $M - 1$, so we have representations

$$\pi_1(\tilde{\mathbb{C}}) \longrightarrow G, \tag{233}$$

with $G$ $\mathbb{Z}$-orthogonal or $\mathbb{Z}$-symplectic, according to the given parity. Any such irreps determines univocally a local system. Notice that from the Primitive Lefschetz decomposition one can read out the precise structure of the intersection product induced by the polarization.

At this point, one can also determine a basis for the homology underlying the original integral using this perverse extension. We can get a twisted sheaf over $C_*$, given by the Local System $\mathcal{L}$ twisted by the perverse monodromies, associated with $t^\gamma$. The original integral can be thought to be defined on a middle dimensional cycles in a complex $m$-dimensional variety $Y$. The cohomology $H^1(\mathbb{C}_*, \mathcal{L})$ determines a basis of real $m$-dimensional *Lefschetz thimbles* in $Y$, Figure 22, which indeed represent the perverse cohomology structure.

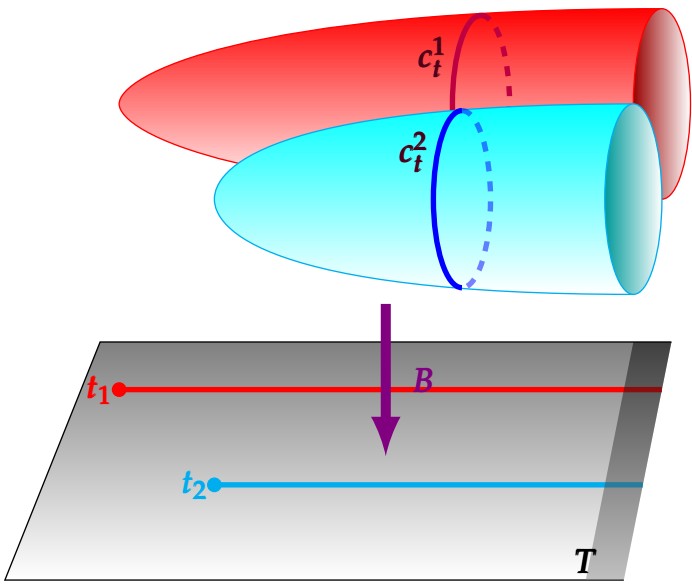

**Figure 22.** Thimbles drawn above the plane of $t = -\gamma \log B$.

On $\mathbb{C}$, each critical value $t_\alpha$ of $B$ determines a phase of $t_\alpha^\gamma$. Draw $n$ lines with such constant phases, passing through the $n$ critical values (so $\gamma \log B$ has constant imaginary part) starting from the critical values of $B$ to $\infty$ or to zero, according $\text{Re}(\gamma) < 0$, or $\text{Re}(\gamma > 0)$. To each line, it corresponds to a family of $(m-1)$-dimensional cycles $c_t^\alpha$, collapsing to a point in $t = t_\alpha$ (a collapsing cycle) and going towards a region where $B^{-\gamma}$ assumes larger and larger values. Such cycles are determined by the coinvariant elements in a way similar to the construction we gave in Section 5.4. Fix a large real value $T \gg \text{Re}(-\gamma \log t_\alpha)$, for all $\alpha$; then $-\gamma \log t > T$ corresponds to the region $Z \subset Y$ where $B^{-\gamma}|_Z > T$. The unions

$$\tau^\alpha = \bigcup_{t \left| \left( \begin{smallmatrix} \text{Im}(-\gamma \log \frac{t}{t_\alpha}) = 0, \\ \text{Re}(-\gamma \log \frac{t}{t_\alpha}) > 0 \end{smallmatrix} \right. \right)} c_t^\alpha, \tag{234}$$

are called Lefschetz thimbles, and result as a basis for the middle homology of $Y$ relative to $Z$. We remark that such constructions are quite familiar to physicists even if in a different form. Let us define the potential function

$$W = -\hbar \gamma \log B \tag{235}$$

so that the integrand becomes

$$e^{\frac{W}{\hbar}}. \tag{236}$$

One can evaluate the integral for very small $\hbar$ by using the stationary phase method. Therefore, such an integral is dominated by the stationary points, where

$$dW = 0 = \frac{dB}{B}. \tag{237}$$

These are the critical points of the polynomial. The valuation of the integral is then dominated by the paths that make the integrand oscillate as little as possible. These are the union of paths such that the phase of the integrand is constant, and that goes to infinity with $\mathrm{Re}(W) > 0$, so that the integrand goes to zero. This means $\mathrm{Re}(-\gamma \log \frac{t}{t_\alpha}) > 0$, and we get the thimbles.

Here, we end our mathematical excursus and go back to the practical constructions of Cho, Matsumoto, and others.

## 7. Computing Intersection Numbers: State of the Art and Open Problems

In this section, we mainly address the practical computation of the coefficients of the master integrals, the so-called *intersection numbers*.

Intersection numbers between two $n$-forms $\langle \phi_L |$ and $| \phi_R \rangle$, which have been introduced abstractly in Section 6, are defined by Cho and Matsumoto [74] as

$$\langle \phi_L | \phi_R \rangle_\omega := \frac{1}{(2\pi i)^n} \int_X \iota_\omega(\phi_L) \wedge \phi_R, \tag{238}$$

where $X$ is the whole complex space $\mathbb{C}^n$ deprived of the hyperplanes corresponding to the poles of $\omega$ and $\iota_\omega$ is a map that sends $\phi_L$ to an equivalent form with compact support. In our particular case, $\langle \phi_L |$ and $| \phi_R \rangle$ are *twisted* cocycles, which implies that the intersection number in Equation (238) is not an integer in general [137].

Notice that if we omitted $\iota_\omega$ in Equation (238), then the intersection number would vanish, as both $\phi_L$ and $\phi_R$ are holomorphic in the domain of integration $X$. To understand why this is true, we focus on a 1D description (there is only a single variable $z$) in order to have a simple notation, but there is no difference in considering a multidimensional example. We consider $z$ to be complex; hence $X$ is $\mathbb{C}$ without some points corresponding to poles. A certain function $f(x, y)$ can always be split into its real and imaginary part: $f(x, y) = u + iv$. Introducing a change of variables

$$\begin{cases} z = x + iy \\ \bar{z} = x - iy \end{cases} \tag{239}$$

allows to obtain a certain $\tilde{f}(z, \bar{z})$. If $f$ is holomorphic, in order to satisfy the Cauchy–Riemann conditions, after the change of variables, $\tilde{f}$ depends only on $z$: $\tilde{f}(z, \bar{z}) = f(z)$. In general a form $\phi$ can be decomposed as

$$\phi = \hat{\phi}_1 dz + \hat{\phi}_2 d\bar{z}. \tag{240}$$

When we wedge two forms as in Equation (240), only the mixed terms $dz \wedge d\bar{z}$ survive; but if the forms involved are holomorphic, the wedge product is null.

Strictly focusing on the twisted cohomology framework, we present in this section a summary of what has been obtained for the computation of intersection numbers so far; in Section 7.4, we give a review of still open problems plus some ideas on how to tackle them.

### 7.1. Univariate Case

In this section, we show how the intersection number can be evaluated exactly in the case when integrals are defined over only one complex variable. We will refer to the simple derivation that was presented in [76]. Let us consider a 1-form $\phi_L$ having poles at some points $z_i \in \mathbb{C}$. In order to compute Equation (238), $\iota_\omega(\phi_L)$ must be constructed explicitly. The key point lies in defining circular regions around each $z_i$ point: to fix ideas,

we call $V_i$ and $U_i$ two discs centered in $z_i$ such that $V_i \subset U_i$; these discs are defined such that $U_i \cap U_j = \varnothing$ for $i \neq j$. We then introduce for each $i$:

1: A holomorphic function $\psi_i$ such that

$$\nabla_\omega \psi_i = \phi_L \ \text{ on } \ U_i \setminus \{z_i\}; \tag{241}$$

2: A function $h_i$ such that

$$h_i = \begin{cases} 1 \ \text{ on } \ V_i; \\ 0 \leq h_i \leq 1 \ \text{ smooth interpolation on } \ U_i \setminus V_i; \\ 0 \ \text{ out of } \ U_i. \end{cases} \tag{242}$$

Then $\iota_\omega(\phi_L)$ can be written as

$$\iota_\omega(\phi_L) = \phi_L - \sum_i \nabla_\omega(h_i \psi_i) = \phi_L - \sum_i (dh_i \psi_i + h_i \nabla_\omega \psi_i). \tag{243}$$

Notice that in Equation (243), as $h_i = 0$ out of $U_i$, we are actually not modifying $\phi_L$ in that region. Inside $V_i$, $h_i = 1$, meaning that the whole $\phi_L$ is subtracted and $\iota_\omega(\phi_L) = 0$ in the innermost region around the singular points; on the other hand, in the outer ring $U_i \setminus V_i$, $\phi_L$ is subtracted smoothly. Finally, $-dh_i \psi_i$ is an extra term existing only in the $U_i \setminus V_i$ ring. Notice that $\phi_L$ and $\iota_\omega(\phi_L)$ lie in the same cohomology class, as they are identical up to a covariant derivative of a function.

What is left now is to find the explicit form of $\psi_i$ obeying Equation (241).

**Lemma 1.** *Unique existence of $\psi_i$.*
  *$\exists! \ \psi_i$ such that $\psi_i$ is holomorphic on $U_i \setminus \{z_i\}$ and $\nabla_\omega \psi_i = \phi_L$ on $U_i \setminus \{z_i\}$.*

**Proof.** Consider a 1-form $\phi_L$ having a pole of order $N$ at $z = z_i$; $\omega$ is a 1-form sharing a pole of $\phi_L$, $\psi_i$ is required to be holomorphic: hence, in terms of the local coordinate variable $z$ near $z_i$ (up to a change of coordinates, we can consider $z_i = 0$ without loss of generality), and it is possible to write

$$\phi_L = \sum_{m=-N}^{\infty} b_m z^m dz, \qquad \omega = \sum_{q=-1}^{\infty} a_q z^q dz, \qquad \psi_i = \sum_{m=-N+1}^{\infty} c_m z^m. \tag{244}$$

Notice how in (244) $b_{-1} = \mathrm{Res}_{z=z_i}(\phi_L)$ and $a_{-1} = \mathrm{Res}_{z=z_i}(\omega) = \alpha_i$. Then $\nabla_\omega \psi_i$ becomes

$$(d + \omega \wedge) \sum_{m=-N+1}^{\infty} c_m z^m = d\left( \sum_{m=-N+1}^{\infty} c_m z^m \right) + \sum_{m=-N+1}^{\infty} \sum_{q=-1}^{\infty} a_q c_m z^{m+q} dz =$$
$$= \sum_{n=-N+1}^{\infty} \left( (n+1)c_{n+1} + \sum_{q=-1}^{n} a_q c_{n-q} \right) z^n dz, \tag{245}$$

in which we defined $n = m + q$: hence the sum $\sum_{m=-N+1}^{\infty} \sum_{q=-1}^{\infty}$ must satisfy the condition $q = n - m$, whose bigger value is obtained when $m$ is as small as possible (which is $m = -N + 1$). This implies that the sum over $q$ goes up to $n + N - 1$. We are thus led to the identification

$$(n+1)c_{n+1} + \sum_{q=-1}^{n+N-1} a_q c_{n-q} = b_n \xrightarrow{n=-1} \sum_{q=-1}^{N-2} a_q c_{-q-1} = b_{-1}. \quad \square$$

The intersection number (238) can be rewritten as

$$\langle \phi_L | \phi_R \rangle_\omega = \frac{1}{2\pi i} \int_X \left[ \phi_L - \sum_i (dh_i)\psi_i - \sum_i h_i \nabla_\omega \psi_i \right] \wedge \phi_R = -\frac{1}{2\pi i} \sum_i \int_{U_i \setminus V_i} dh_i \psi_i \wedge \phi_R, \tag{246}$$

where the second equality can be obtained by recalling that the first term vanishes because $\phi_L \wedge \phi_R = 0$; the second term survives where $dh_i \neq 0$, i.e., in $U_i \setminus V_i$; the third term is again proportional to $\phi_L \wedge \phi_R$ and vanishes. Notice that it holds:

$$dh_i \psi_i \wedge \phi_R = d(h_i \psi_i \phi_R). \tag{247}$$

This is because the extra terms $h_i d\psi_i \wedge \phi_R$ and $h_i \psi_i d\phi_R$ vanish, as $d\psi_i$ and $\phi_R$ are both holomorphic and as $\phi_R$ is a closed form. By the Stokes theorem, it is possible to rewrite Equation (246) as

$$\langle \phi_L | \phi_R \rangle_\omega = -\frac{1}{2\pi i} \sum_i \int_{U_i \setminus V_i} d(h_i \psi_i \phi_R) = -\frac{1}{2\pi i} \sum_i \int_{\partial(U_i \setminus V_i)} h_i \psi_i \phi_R = \frac{1}{2\pi i} \sum_i \int_{\partial V_i} \psi_i \phi_R. \tag{248}$$

Because $\partial V_i$ is a closed path, we can always rewrite Equation (248) as a sum of residues:

$$\langle \phi_L | \phi_R \rangle_\omega = \sum_{i=1}^k \operatorname{Res}_{z=z_i}(\psi_i \phi_R). \tag{249}$$

### 7.2. Logarithmic n-Forms

Evaluation of intersection numbers is particularly simple for logarithmic forms. In the univariate case, one can check from Lemma 1 that if $\phi_L$ has a simple pole, then the function $\psi_i$ takes the form

$$\psi_i = \frac{\operatorname{Res}_{z=z_i}(\phi_L)}{\operatorname{Res}_{z=z_i}(\omega)} + \mathcal{O}(z - z_i) = \frac{\operatorname{Res}_{z=z_i}(\phi_L)}{\alpha_i} + \mathcal{O}(z - z_i). \tag{250}$$

In this case, Equation (249) leads to the following formula for the evaluation of the intersection number:

$$\langle \phi_L | \phi_R \rangle_\omega = \sum_i \frac{\operatorname{Res}_{z=z_i} \phi_L \operatorname{Res}_{z=z_i} \phi_R}{\alpha_i}. \tag{251}$$

Equation (251) is valid for logarithmic 1-forms, but it can be generalized to logarithmic $n$-forms. We give only the final result for this case. Let us consider the twisted one-form $\omega$ in the following form:

$$\omega = \sum_{i=0}^{s+1} \alpha_i d\log f_i, \tag{252}$$

where the $f_i$ are linear functions in the given variables (hence each $f_i = 0$ defines a certain hyperplane where the twist $\omega$ shows a logarithmic singularity), while $\sum_i \alpha_i = 0$. The cocycles $\phi_{L,R}$ instead assume the form

$$\phi_I = d\log\left(\frac{f_{i_0}}{f_{i_1}}\right) \wedge \cdots \wedge d\log\left(\frac{f_{i_{n-1}}}{f_{i_n}}\right) \tag{253}$$

with $I = (i_o, \cdots, i_n)$; the indexes are ordered such that $0 \leq i_0 < \cdots < i_n \leq s+1$. In other words, $\phi_I$ belongs to the $n$-th twisted cohomology group $H^n(X, \nabla_\omega)$, where $X = \mathbb{CP}^n \setminus \{\cup_i f_i = 0\}$.

With these hypotheses, a recent work [46] has shown that the intersection number between two logarithmic $n$-forms can also be written as

$$\langle \phi_L | \phi_R \rangle_\omega = \frac{1}{(-2\pi i)^n} \oint_{\wedge_{a=1}^n \{|\omega_a| = \epsilon\}} \frac{\phi_L \hat{\phi}_R}{\prod_{a=1}^n \omega_a}, \tag{254}$$

where $\hat{\phi}$ is the component of $\phi = \hat{\phi}\, dz_1 \wedge \cdots \wedge dz_n$, $a$ enumerates the dimensions involved (the number of variables), and $\omega_a$ represents the components of the twist $\omega$ along each $dz_a$.

### 7.3. Multivariate Case: Recursive Method

The main aim now is to be able to compute any intersection number, independently of the order of the poles appearing and the number of dimensions involved. A first attempt follows a recursive approach in the number of variables (we refer to [65,66]). In order to compute an intersection number in $z_1, \cdots, z_n$ variables, the problem can be divided into two steps: calculating the intersection number in $z_1, \cdots, z_{n-1}$ variables and then a generalized intersection number depending only on $z_n$, which will be introduced later. In the 1D case, $\phi_{L,R}$ and $\omega$ are 1-forms, which we can regard as some sort of *scalars*. Taking into account the existence of $\nu$ independent equivalence classes, it is possible to build *vectorial* ($\phi_{L,R,j}$) and *tensorial* ($\Omega_{ij}$) objects, with "gauge" transformations written as

$$\hat{\phi}'_{L,j} = \hat{\phi}_{L,j} + \partial_z \xi_j + \xi_i \Omega_{ij}, \qquad \hat{\phi}'_{R,j} = \hat{\phi}_{R,j} + \partial_z \xi_j - \Omega_{ji}\xi_i. \tag{255}$$

The next key idea is to study, for each $i$, the twisted cohomology group associated to the fibration

$$\nabla_\omega = d + \sum_{j=1}^{n} \omega_j dz_j = \sum_{j=1}^{i} dz_j \left( \frac{\partial}{\partial z_j} + \omega_j \right) + \sum_{j=i+1}^{n} dz_j \left( \frac{\partial}{\partial z_j} + \omega_j \right), \tag{256}$$

and define

$$\omega^{(i)} := \sum_{j=1}^{i} \omega_j dz_j. \tag{257}$$

In Equation (256) $z_{i+1}, \cdots, z_n$ are treated as fixed parameters. Starting from $i = n-1$, the goal is to express a cohomology class $\left\langle \phi_L^{(n)} \right| \in H_\omega^n$ using a basis of $H_\omega^{n-1}$, which is supposed to be known: such a basis is formed by $\nu_{n-1}$ elements $\left\langle e_j^{(n-1)} \right|$, with dual basis $\left| d_j^{(n-1)} \right\rangle$ such that $\left\langle e_j^{(n-1)} \middle| d_k^{(n-1)} \right\rangle = \delta_{jk}$. Inserting an identity of the form $I = \sum_{j=1}^{\nu_{n-1}} \left| d_j^{(n-1)} \right\rangle \left\langle e_j^{(n-1)} \right|$ leads to

$$\left\langle \phi_L^{(n)} \right| = \sum_{j=1}^{\nu_{n-1}} \left\langle \phi_{L,j}^{(n)} \right| \wedge \left\langle e_j^{(n-1)} \right|, \qquad \left| \phi_R^{(n)} \right\rangle = \sum_{j=1}^{\nu_{n-1}} \left| d_j^{(n-1)} \right\rangle \wedge \left| \phi_{R,j}^{(n)} \right\rangle, \tag{258}$$

where

$$\left\langle \phi_{L,j}^{(n)} \right| = \left\langle \phi_L^{(n)} \middle| d_j^{(n-1)} \right\rangle, \qquad \left| \phi_{R,j}^{(n)} \right\rangle = \left\langle e_j^{(n-1)} \middle| \phi_R^{(n)} \right\rangle. \tag{259}$$

Notice how the coefficients defined in Equation (259) are not unique objects but classes themselves. For example, the coefficient $\left\langle \phi_{L,j}^{(n)} \right|$ is invariant with respect to the choice within the cohomology class $\left| d_j^{(n-1)} \right\rangle$ of $H_\omega^{*\,n-1} = H_{-\omega}^{n-1}$: a change $d_j^{n-1} \to d_j^{n-1} + \nabla_{-\omega^{(n-1)}} \psi$ with $\psi$ being a 2-form would make no difference. This equivalence class is smaller than $\left\langle \phi_L^{(n)} \right|$, which is composed of all the transformations $\phi_L^{(n)} \to \phi_L^{(n)} + \nabla_\omega \xi$ with $\xi$ being a generic $(n-1)$-form: thus, the coefficients in Equation (259) represent new equivalence classes overall.

We introduce a new matrix $\Omega^{(n)}$ with dimension $\nu_{n-1} \times \nu_{n-1}$

$$\Omega_{ij}^{(n)} := \left\langle (\partial_{z_n} + \omega_n) e_i^{(n-1)} \middle| d_j^{(n-1)} \right\rangle, \tag{260}$$

which implies

$$\sum_i \Omega_{ij}^{(n)} \left\langle e_j^{(n-1)} \right| = \left\langle (\partial_{z_n} + \omega_n) e_i^{(n-1)} \right|. \tag{261}$$

The matrix in Equation (260) allows us to write the gauge transformations of the coefficients in the following form (the derivation can be found in [66]):

$$\hat{\phi}^{(n)}_{L,j} \to \hat{\phi}^{(n)}_{L,j} + g_i \left( \overleftarrow{\partial_{z_n}} \delta_{ij} + \Omega^{(n)}_{ij} \right), \qquad \hat{\phi}^{(n)}_{R,j} \to \hat{\phi}^{(n)}_{R,j} + \left( \delta_{ij} \overrightarrow{\partial_{z_n}} - \Omega^{(n)}_{ji} \right) g_i, \qquad (262)$$

where the $g_i$ are arbitrary functions. With a shorter notation, Equation (262) can be rewritten as

$$\hat{\phi}_j \to \hat{\phi}_j + (\delta_{jk} \partial_{z_n} + \Omega_{jk}) g_k, \qquad (263)$$

where the $L$ and $R$ cases are restored taking $\Omega \equiv (\Omega^{(n)})^{\mathsf{T}}$ and $\Omega \equiv -\Omega^{(n)}$, respectively. In a number of practical cases, the matrix $\Omega^{(n)}$ can often be chosen to have only simple poles by the application of the *Moser algorithm*, whose details we do not present in this work but can be found in [138,139].

This property, along with Equation (263), allows us to find coefficients $\hat{\phi}_j$ that are more convenient than the original ones, namely coefficients that show only simple poles in $z_n$. We start looking for the transformation that lowers the order of a pole appearing in $\hat{\phi}_j$. We outline the procedure when a pole is located at a finite point. To fix ideas, we call $q$ the irreducible polynomial appearing at the denominator of $\hat{\phi}_j$, raised to a certain power $o$. The correct transformation is given by the choice

$$g_j(z_n) = \frac{1}{q^{o-1}} \sum_{k=0}^{\deg(q)-1} c_{jk} z_n^k, \qquad (264)$$

where the coefficients $c_{jk}$ have yet to be determined. With the introduction of Equation (264), Equation (263) assumes the form

$$\hat{\phi}_j \to \hat{\phi}_j + \delta_{jk} \left[ \partial_{z_n} \left( \frac{1}{q^{o-1}} \right) \sum_{s=0}^{\deg(q)-1} c_{ks} z_n^s + \frac{1}{q^{o-1}} \sum_{s=0}^{\deg(q)-1} c_{ks} \partial_{z_n} z_n^s \right] + \frac{1}{q^{o-1}} \sum_{s=0}^{\deg(q)-1} \Omega_{jk} c_{ks} z_n^s$$

$$= \hat{\phi}_j - (o-1) \frac{\partial_{z_n} q}{q^o} \underbrace{\sum_{s=0}^{\deg(q)-1} c_{js} z_n^s}_{1} + \underbrace{\frac{1}{q^{o-1}} \sum_{s=0}^{\deg(q)-1} s c_{js} z_n^{s-1}}_{2} + \underbrace{\frac{1}{q^{o-1}} \sum_{s=0}^{\deg(q)-1} \Omega_{jk} c_{ks} z_n^s}_{3}. \qquad (265)$$

Notice how the term labeled 1 in Equation (265) is the one involving the power $q^0$ in the denominator: these terms (which are of the type $z_n^k / q^o$ with $k \in \{0, \cdots, \deg(q) - 1\}$) can be eliminated by solving $\deg(q)$ equations. The solution of this system determines $c_{jk}$ for a fixed $j$. Considering all the $j$ values, all the appropriate $c_{jk}$ coefficients are determined by solving $\nu_{n-1} \cdot \deg(q)$ equations. A similar procedure exists when the pole is not located at a finite point but at infinity.

Notice that, thanks to Equation (258), it is possible to write the intersection number as

$$\langle \phi_L | \phi_R \rangle_\omega = \sum_{z_0 \in S_n} \sum_{j=1}^{\nu_{n-1}} \text{Res}_{z_n = z_0} \left( \hat{\psi}^{(n)}_{L,j} \left\langle e^{(n-1)}_j \Big| \phi_R \right\rangle \right), \qquad (266)$$

where $S_n$ is the set comprehensive of all the singular points of $\Omega^{(n)}$ in $z_n$, while $\hat{\psi}^{(n)}_{L,j}$ is a function that locally solves

$$\partial_{z_n} \hat{\psi}^{(n)}_{L,j} + \hat{\psi}^{(n)}_{L,i} \Omega^{(n)}_{ij} = \hat{\phi}^{(n)}_{L,j}. \qquad (267)$$

Notice how this situation closely resembles the one outlined by Equation (251) and Equation (241). Similarly to the result given by Lemma 1, the solution of Equation (267) is given by

$$\hat{\psi}^{(n)}_{L,j} = \sum_i \text{Res}_{z_n = z_0} \hat{\phi}^{(n)}_{L,i} \left( \text{Res}_{z_n = z_0} \Omega^{(n)} \right)^{-1}_{ij} + \mathcal{O}(z_n - z_0). \qquad (268)$$

Notice that only the first term in Equation (268) contributes to the residue in Equation (266); hence, recalling that $\hat{\phi}_{L,i}^{(n)}$ and $\Omega^{(n)}$ have at most simple poles in $z_n$, it is possible to substitute

$$\hat{\psi}_{L,j}^{(n)} \to \sum_i \hat{\phi}_{L,i}^{(n)} \left(\Omega^{(n)}\right)_{ij}^{-1}. \tag{269}$$

Remembering that adj $\Omega^{(n)} = \det(\Omega^{(n)}) \cdot (\Omega^{(n)})^{-1} = (\Omega^{(n)})^{-1} P/Q$ (we write $\det \Omega^{(n)} = P/Q$), we can finally obtain

$$\langle \phi_L | \phi_R \rangle_\omega = \sum_{z_0 \in S_n} \sum_{i,j=1}^{\nu_{n-1}} \mathrm{Res}_{z_n = z_0} \left( \frac{Q}{P} \hat{\phi}_{L,i}^{(n)} \left(\mathrm{adj}\,\Omega^{(n)}\right)_{ij} \hat{\phi}_{R,j}^{(n)} \right)$$

$$= \frac{1}{2\pi i} \int_\gamma dz_n \sum_{i,j=1}^{\nu_{n-1}} \frac{Q \hat{\phi}_{L,i}^{(n)} \left(\mathrm{adj}\,\Omega^{(n)}\right)_{ij} \hat{\phi}_{R,j}^{(n)}}{P}, \tag{270}$$

where by $\gamma$ we mean the contour of small counterclockwise circles around the $z_0 \in S_n$ points.

Let us take a deeper look at Equation (270). In the univariate case, the twist is a certain $\omega = \omega_1 dz_1$ with $\omega_1 = P/Q$, where $P$ and $Q$ are certain polynomials in the only variable $z_1$. Calling $C_1 = \{z_1 \in \mathbb{C} : P(z_1) = 0\}$ and $I_1 = <P>$ the ideal generated from $P$, then $C$ is the algebraic variety generated from $I_1$: $C = V(I_1)$. If $\phi_{L,R}$ have only simple poles, then we can express the intersection number between such forms as a global residue:

$$\langle \phi_L | \phi_R \rangle_\omega = -\mathrm{Res}_{<P>}(Q \hat{\phi}_L \hat{\phi}_R). \tag{271}$$

The notation in Equation (271) is as follows [140]: given a certain form

$$\tilde{\omega} = \frac{h(z) dz_1 \wedge \cdots \wedge dz_n}{f_1(z) \cdots f_n(z)}, \tag{272}$$

in which the variables are $z = (z_1, \cdots, z_n)$ and $f \equiv f_1(z), \cdots, f_n(z)$ are $n$ holomorphic functions in the neighborhood of the closure $\overline{U}$ of the ball $U = \{||z - \xi|| < \epsilon\}$ which have a unique isolated zero given by the point $\xi$ ($f^{-1}(0) = \xi$), then the residue of $\tilde{\omega}$ at the point $\xi$ is defined as

$$\mathrm{Res}_{\{f_1, \cdots, f_n\}, \xi}(\tilde{\omega}) = \left(\frac{1}{2\pi i}\right)^n \oint_\Gamma \frac{h(z) dz_1 \wedge \cdots \wedge dz_n}{f_1(z) \cdots f_n(z)}. \tag{273}$$

In Equation (273), the contour $\Gamma$ is given by $\Gamma = \{z : |f_i(z) = \epsilon_i|\}$ with orientation given by $d(\arg f_1) \wedge \cdots \wedge d(\arg f_n) \geq 0$. The subscript $<P>$ in Equation (271) refers to the fact that the residue in Equation (273) does not depend strictly on the $f$ functions in the denominator, but more in general to the ideal they generate $<f_1, \cdots, f_n>$, as we will see in more detail later in Section 7.4. Notice that Equation (271) is equivalent to Equation (251). With this new formalism, Equation (270) can be similarly rewritten as

$$\langle \phi_L | \phi_R \rangle_\omega = -\mathrm{Res}_{<P>}(Q \hat{\phi}_{L,i}(\mathrm{adj}\,\Omega^{(n)})_{ij} \hat{\phi}_{R,j}), \tag{274}$$

in which the contour given by the many counterclockwise circles of Equation (270) is now deformed into a single clockwise contour (hence the minus sign) partially encircling a singular point, going towards infinity, coming back from infinity, partially encircling the next singular point, etc. until a closed path is formed. Equation (274) generalizes Equation (271) in many dimensions. Notice how in Equation (260), a generalized twist $\Omega^{(n)}$ was built, with properties very similar to the ones of the usual twist $\omega$. By writing its determinant $\det \Omega^{(n)} = P/Q$, one can build the set $C_n = \{z_n \in \mathbb{C} : P(z_n) = 0\}$ (set of the points where $\Omega^{(n)}$ does not have full rank) and $I = <P>$, the ideal generated by $P$, similarly to what happened when writing the univariate Equation (271).

In conclusion, the procedure described in this section outlines an algorithm which, given as an input two cohomology classes $\phi_{L,R}$, computes their intersection number recursively: starting from $i = n - 1$, it expands $\phi_{L,R}$ on the basis of $H^{(n-1)}_{\omega,-\omega}$ and computes the matrix $\Omega^{(n)}$. By using transformations of the form (263), it transforms the coefficient vectors $\phi_{L,R,j}$ into equivalent coefficients but with only simple poles in $z_n$ and therefore can finally compute the residue (274). The procedure is then repeated for all $i$, until one has only an intersection number depending on the last variable.

*7.4. Moving Onwards: An Open Problem*

Recursive approaches to the computation of intersection numbers are known, but the time needed for their computation can be extremely long because of the many steps comprising the algorithm. A new question arises: is there any way to compute (238) without having to introduce some type of recursivity (whether in the number of variables or in the reduction of the order of the poles appearing in the twisted cocycles)? The fact that a recursive solution to the problem exists suggests there might be some underlying property of intersection numbers that could lead to a more fundamental method of computation. Being able to understand this could make the computation faster, but it would also imply getting a deeper understanding of the problem: in the end, it would be possible to understand what Feynman integrals represent—both in an algebraic and geometric perspective, not only by a straightforward computation problem point of view.

As a first note, we observe that in every case computation can be carried out, intersection numbers are always written in terms of a residue of the form Equation (273): we find it instructive to focus on Equation (273) and highlight some of its properties.

1:  Non-degenerate case.
    If $f = f_1, \cdots, f_n$ is non-degenerate (i.e., its Jacobian evaluated in 0 is $J_f(0) \neq 0$), then Equation (273) can be evaluated by the introduction of a change of variables $w := f(z)$. Using the usual Cauchy formula leads to

$$\left(\frac{1}{2\pi i}\right)^n \int_\Gamma \tilde{\omega} = \left(\frac{1}{2\pi i}\right)^n \int_{|w_i|=\epsilon_i} h(f^{-1}(w)) \frac{dw_1 \cdots dw_n}{J_f(w)} \frac{1}{w_1} \cdots \frac{1}{w_n} = \frac{h(f^{-1}(0))}{J_f(0)}. \tag{275}$$

2:  $h \in \mathrm{I}(f)$ ideal generated from the $f_i$.
    In this case the residue is 0. If, for example, $h(z) = g(z)f_1(z)$, then $\tilde{\omega}$ is holomorphic in a bigger set, which we call $U_1 := U \setminus (D_2 + \cdots D_n)$.
    Then the contour $\Gamma_1 = \{z \text{ such that } |f_1(z_1)| \leq \epsilon_1, |f_i(z_i)| = \epsilon_i \text{ for } i \neq 1\}$, which is an element of $U_1$, has a boundary $\delta\Gamma_1 = \pm\Gamma$: by the Stokes theorem, the residue of $\tilde{\omega}$ along $\Gamma$ is then 0.

While the general degenerate case is hard to compute, in [141] is suggested that if each of the $z_i$ appearing in $\tilde{\omega}$ depends only on a single variable $z_i$, it is possible to factorize Equation (273) as

$$\mathrm{Res}_{(0)}(\tilde{\omega}) = \left(\frac{1}{2\pi i}\right)^n \oint \frac{dz_1}{f_1(z_1)} \cdots \oint \frac{dz_n}{f_n(z_n)} h(z). \tag{276}$$

How can this be achieved? Algebraic geometry [140] shows that actually Equation (273) can be reinterpreted in terms of sheaf cohomology, which allows us to obtain an important theorem:

**Theorem 8.** *Transformation Theorem*
*Let $f = \{f_1, \cdots, f_n\}$ and $g = \{g_1, \cdots, g_n\}$ be holomorphic maps $f, g : \bar{U} \to \mathbb{C}^n$ such that $f^{-1}(0) = g^{-1}(0) = 0$ and $g_i(z) = \sum_j A_{ij}(z)f_j(z)$, with $A_{ij}$ being a holomorphic matrix. Then*

$$\mathrm{Res}_{(0)}\left(\frac{h(z)dz_1 \wedge \cdots \wedge dz_n}{f_1(z) \cdots f_n(z)}\right) = \mathrm{Res}_{(0)}\left(\frac{h(z)dz_1 \wedge \cdots \wedge dz_n}{g_1(z) \cdots g_n(z)} \det A\right). \tag{277}$$

Given $n$ functions $g_i(z) = g_i(z_i)$ obeying the hypotheses of Theorem 8, then the residue in Equation (273) would take the form of Equation (276). Calling $R = \mathbb{K}(z_1, \cdots, z_n)$ the ring over a field $\mathbb{K}$ with $n$ variables $z_1, \cdots, z_n$, we notice that the ideal $I = <f_1, \cdots, f_n> \subset R$ generated by the $f_1, \cdots, f_n$, defined as $I = \{\sum_i h_i f_i \mid h_i \in R\}$ implies that the ideal $J$ generated by the $g_i$ polynomials is a subset of $I$:

$$J = \left\{ \sum_i \tilde{h}_i g_i = \sum_{ij} \tilde{h}_i A_{ij}(z) f_j = \sum_i \left( \sum_j \tilde{h}_j A_{ji}(z) \right) f_i \mid \tilde{h}_i \in R \right\} \subset I. \qquad (278)$$

This observation implies that Theorem 8 can be reinterpreted in the following way: the residue does not depend on the specific generators of the ideal $I$: it is possible to use the generators of an ideal that is a subset of the original one. As suggested in [141], there is a way to generate such an ideal: the exploitation of Gröbner bases, which can be regarded as a set of polynomials with special properties. In Appendix B, we give a brief introduction to the topic (for more details, see for example [142]).

Going back to the problem of the computation of the residue and following the idea from [141], we recall that the aim is to find some new $g_i(z) = A_{ij}(z) f_j(z)$ that are more convenient than the original $f_i$ polynomials. In order to find $g_i(z) = g_i(z_i)$ for a fixed $i$, we define a Lexicographic order in which the $i$-th variable $z_i$ is the "least important variable":

$$z_{i+1} \succ z_{i+2} \succ \cdots \succ z_n \succ z_1 \succ z_2 \succ \cdots \succ z_i. \qquad (279)$$

Starting from the original $f_1, \cdots, f_n$ polynomials and constructing a Gröbner basis while keeping this order in mind, one fact is inevitable: one polynomial in the basis must depend only on the $z_i$ variable. In fact, the obtained Gröbner basis can be reduced with respect to itself (i.e., each polynomial in the basis can be divided with respect to the other ones): because of the chosen order, this process progressively eliminates the variables $z_1, \cdots, z_{i-1}, z_{i+1}, z_n$, leaving at least one polynomial depending only on the last variable $z_i$. This polynomial is $g_i(z) = g_i(z_i)$. Notice that, by the Buchberger's algorithm, the many elements of the basis are constructed as a linear combination of the former polynomials: all of them are of the form requested by the Transformation Theorem hypotheses. Repeating this process for all the $n$ possible lexicographic orders, at the end, a set $\{g_1(z_1), \cdots, g_n(z_n)\}$ is created. The computation of Gröbner bases is achieved by the software *Macaulay2* [143]: here we give a brief example of how this procedure works.

**Example 1.** *As an example, we study the residue for $\tilde{\omega} = \frac{z_2 dz_1 \wedge dz_2}{z_1^2 (z_2 - z_1)}$.*

*Here we use $h(z) = z_2$, $f_1(z) = z_1^2$, $f_2(z) = z_2 - z_1$. The Jacobian is degenerate in $(0, 0)$; hence we compute the residue of $\tilde{\omega}$ around $(0, 0)$.*

*First, we look for $g_1(z_1)$: in this case there is no need to start creating a Gröbner basis associated with the ordering $z_2 \succ z_1$: $f_1(z)$ already depends only on $z_1$. Hence, we simply consider $g_1(z_1) = f_1(z) = z_1^2$.*

*We now switch to the problem of finding $g_2(z_2)$, using the order $z_1 \succ z_2$: it is clear that it is not possible to obtain a function depending only on $z_2$ by dividing each $f_i$ by the other remaining polynomial, so we build the Gröbner basis. Notice how this suggests that $\{f_1, f_2\}$ cannot form a complete Gröbner basis for the chosen order.*

$$S(f_1, f_2) = lcm(z_1^2, z_1) \left( \frac{f_1}{z_1^2} + \frac{f_2}{z_1} \right) = z_1 z_2. \qquad (280)$$

*Dividing $S(f_1, f_2)$ with respect to the set $\{f_1, f_2\}$ with the goal to obtain a remainder that is "smaller" in the sense of the specified order leads to*

$$S(f_1, f_2) = z_1 z_2 \xrightarrow{+z_2 f_2} z_2^2, \qquad (281)$$

*which is irreducible. We then add $S(f_1, f_2)$ (or better yet, its remainder, which has already gone through the division process) to the original set: we obtain $\{f_1, f_2, f_3\}$ with $f_3 = z_2^2$. Without checking if this new set is a Gröbner basis or not (which it actually is), we identify $g_2(z_2) = f_3(z) = z_2^2$.*

$$\begin{pmatrix} g_1 \\ g_2 \end{pmatrix} = \begin{pmatrix} A_{11} & A_{12} \\ A_{21} & A_{22} \end{pmatrix} \begin{pmatrix} f_1 \\ f_2 \end{pmatrix} \; with \; \begin{pmatrix} A_{11} & A_{12} \\ A_{21} & A_{22} \end{pmatrix} = \begin{pmatrix} 1 & 0 \\ 1 & z_1 + z_2 \end{pmatrix} \Rightarrow \det A = z_1 + z_2 \tag{282}$$

*By Theorem 8, the residue can be written as*

$$Res_{\{f_1,f_2\}(0,0)} \tilde{\omega} = Res_{\{g_1,g_2\}(0,0)} \left( \frac{z_2(z_1 + z_2)dz_1 \wedge dz_2}{z_1^2 z_2^2} \right) = \left( \frac{1}{2\pi i} \right)^2 \oint \frac{dz_1}{z_1^2} \oint dz_2 \frac{z_2(z_2 + z_1)}{z_2^2} = 1. \tag{283}$$

We wonder if it is always possible to write an intersection number in terms of a residue of the kind in Equation (273), not only when simple poles are involved. Regarding the emerging importance of the global residue theorem in the computation of intersection numbers, we mention a recent work [62]. It is argued that intersection numbers between forms can be computed by an expansion in the parameter $\gamma$ appearing in the Baikov polynomial $B^\gamma$, both for big and small $\gamma$: it appears that every term in the expansion can be written as a residue. We briefly focus on this result in order to give an intuitive idea of the reasons behind it without entering into too much detail.

We start by considering the manifold $M = \mathbb{CP}^m \setminus \cup_{i=1^k} H_i$, where $H_i$ represents some hyperplanes defined by certain equations $f_i = 0$. With these functions it is possible to build the holomorphic function

$$W = \sum_{i=1^k} \alpha_i \log f_i, \tag{284}$$

which only has logarithmic singularities along the $H_i$ hyperplanes. Equation (284) also defines a holomorphic 1-form $dW$, which in the language introduced in Section 2 corresponds to $d\log(B)$. Similarly, it is possible to construct the operator

$$\nabla_{dW} = d + \gamma dW \wedge, \tag{285}$$

which corresponds to our $d + \omega \wedge$ (remember that $\omega = d\log(B^\gamma) = \gamma d\log(B)$). We consider the space $\Omega^k(M)$ of the k-forms on $M$ and construct two sequences:

$$0 \xrightarrow{dW\wedge} \Omega^1(M) \xrightarrow{dW\wedge} \cdots \xrightarrow{dW\wedge} \Omega^m(M) \xrightarrow{dW\wedge} 0 \tag{286}$$

and

$$0 \xrightarrow{\nabla_{dW}} \Omega^1(M) \xrightarrow{\nabla_{dW}} \cdots \xrightarrow{\nabla_{dW}} \Omega^m(M) \xrightarrow{\nabla_{dW}} 0. \tag{287}$$

Notice how both sequences are well defined, as $dW \wedge dW = 0$ and $\nabla_{dW}^2 = 0$. The associated cohomology groups $H^k(M, dW\wedge)$ and $H^k(M, \nabla_{dW})$ are, respectively, defined as

$$H^k(M, dW\wedge) := \{\phi_k \in \Omega^k(M) | dW \wedge \phi_k = 0\} / \{\phi_k \in \Omega^k(M) | \phi_k = dW \wedge \phi_{k-1}, \text{with } \phi_{k-1} \in \Omega^{k-1}(M)\} \tag{288}$$

and

$$H^k(M, \nabla_{dW}) := \{\phi_k \in \Omega^k(M) | \nabla_{dW}\phi_k = 0\} / \{\phi_k \in \Omega^k(M) | \phi_k = \nabla_{dW}\phi_{k-1}, \text{with } \phi_{k-1} \in \Omega^{k-1}(M)\} \tag{289}$$

All cohomology groups are trivial for each $k \neq m$: only $H^m(M, dW\wedge)$ and $H^m(M, \nabla_{dW})$ contain useful information. We define two pairings, the first between $H^m(M, dW\wedge)$ and itself and the second between $H^m(M, \nabla_{dW})$ and $H^m(M, \nabla_{-dW})$. In the first case, taking $\phi_\pm \in H^m(M, \nabla_{dW})$, it is possible to define [46]

$$(\phi_- | \phi_+)_{dW,0} = Res_{dW=0} \left( \frac{\hat{\phi}_- \hat{\phi}_+ d^m z}{\partial_1 W \cdots \partial_m W} \right), \tag{290}$$

where the ˆ notation strips the forms from their $d^m z$ part and where $\mathrm{Res}_{dW=0}$ represents a sum of residues around each critical point given by $\partial_i W = 0$, for $i = 1, \cdots, m$.

In the second case, taking $\phi_\pm \in H^m(M, \nabla_{\pm dW})$ allows us to define

$$\langle \phi_- | \phi_+ \rangle_{dW} = \left( \frac{\gamma}{2\pi i} \right)^m \int_M \phi_- \wedge \phi_+^c, \tag{291}$$

where $\phi_+^c$ represents the compact version of $\phi_+$ belonging to the same equivalence class as $\phi_+$, similarly to the usual definitions from Cho and Matsumoto [74] (in Equation (238), we take the compact version of $\phi_-$, but it is just a matter of convention): Equation (291) represents an intersection number. It is argued that, just as (288) looks like a limit for large $\gamma$ of (289), the pairings in Equation (290) and Equation (291) must be linked. It is found that Equation (291) can be expanded as follows:

$$\langle \phi_- | \phi_+ \rangle_{dW} = \sum_{k=0}^{\infty} \gamma^{-k} (\phi_- | \phi_+)_{dW,k} \tag{292}$$

where each $(\phi_- | \phi_+)_{dW,k}$ is a *higher residue pairing* [144] and has the form of a $\mathrm{Res}_{dW=0}$ of certain functions that only contain $\hat{\phi}_-, \hat{\phi}_+$ in the numerator and the derivatives $\partial_i W$ both in numerator and denominator.

Notice that, as can be seen in Example 1, any residue of the kind we met can be calculated independently of the order of the poles appearing. If—as suggested by [62]—intersection numbers were always found to be expressed only in terms of residues, then a procedure involving Gröbner bases can be exploited. The biggest difference between the residue in Equation (273) and the integral in Equation (3) is the type of cohomology involved. Consider an $(n-1)$-form $\tilde{\xi}$ of the same type as in Equation (272):

$$\tilde{\xi} = \frac{\tilde{h}(z) dz_1 \wedge \cdots \wedge dz_{n-1}}{\tilde{f}_1(z) \cdots \tilde{f}_{n-1}(z)}. \tag{293}$$

The form $\tilde{\xi}$ is holomorphic everywhere in $U = \left\{ z \in \mathbb{C}^{n-1} \text{ such that } ||z|| < \epsilon \right\}$ except from the set $\tilde{D}$, defined as $\tilde{D} = \tilde{D}_1 + \cdots + \tilde{D}_{n-1}$, where each $\tilde{D}_i = (\tilde{f}_i)$ is the divisor of $\tilde{f}_i$ (namely, it is the formal sum of the points sent to 0 by applying $\tilde{f}_i$). This implies that its external derivative is null: $d\tilde{\xi} = 0$, so that $\int_\Gamma d\tilde{\xi} = 0$, where $\Gamma$ is the contour appearing in Equation (273). As the addition of a term of the type $d\tilde{\xi}$ gives no contribution to Equation (273), the residue of $\tilde{\omega}$ depends on the de Rham cohomology class $[\tilde{\omega}] \in H^n_{\mathrm{dR}}(U \setminus D)$, where $D = D_1 + \cdots + D_n$, with $D_i = (f_i)$. On the other hand, the integral (3) similarly depends on the twisted cohomology class of $\phi$. This difference is due to the presence of $u = B^\gamma$ inside the integral: as we have seen earlier, the effect of $u$ is the introduction of some cuts in the complex space: we wonder if, in a sense, $u$ can be considered as part of the geometry and can be "transferred inside the sign of integral". The integral $I$ on the cut complex space can also be thought of as an integral on a certain Riemann surface where the cuts have been glued together. We wonder if, in this sense, it is possible to rewrite the integral (3) as multi-variable residue with a form similar to Equation (273), or at least as a sum of terms of this kind as suggested in [62]. In this case, a similar approach could be followed: while Theorem 8 emerges from the interpretation of the residue by means of sheaf cohomology—in which the cohomology of interest is the de Rham cohomology—another similar theorem for the twisted cohomology group should exist, telling us how one can adapt the function inside the integral while obtaining the same residue. As a first guess, it should not differ too much from the original theorem. Once having derived how the integrand can be modified to obtain a better integral, the focus could remain the same: having functions in the denominator that depend only on a single variable, so that the computation of the integral can finally be performed.

Of course, while we stress that these observations only outline a possible way to tackle the problem of computing an intersection number as a whole, we highlight how certainly

the path has to lie in looking at Equation (238) not only with straightforward computation in mind, but with the objective of understanding a deeper structure: the path certainly lies in the abstract procedure outlined in Section 6. In this section, we introduced many of the known practical tools: some have already been exploited for the actual evaluation of intersection numbers (Sections 7.1–7.3), while in Section 7.4 we introduced some concepts and definitions that may become part of the practical instruments needed to pursue, in the end, the computation of the Feynman integral.

## 8. An Explicit Example of Feynman Integral

We want to conclude this review by showing how to explicitly apply intersection theory to the computation of Feynman integrals. The present example is due to Manoj Mandal and Federico Gasparotto. Let us consider the double box diagram for massless particles, see Figure 23:

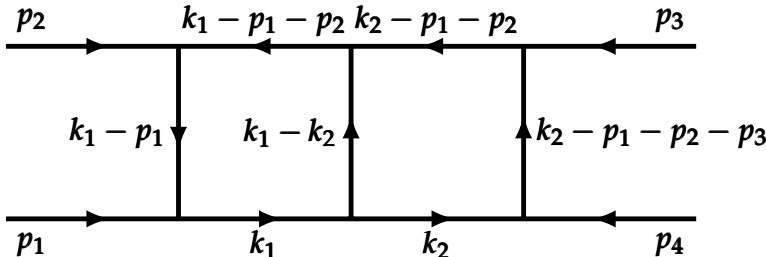

**Figure 23.** Double box diagram.

After reducing to the maximal cut and choosing a basis of master integrals, we will show how to use intersection theory for determining the differential equations for the MIs as functions of a Mandelstam's variable. Of course, momentum conservation requires $p_4 = -p_1 - p_2 - p_3$, so we have $E = 3$ independent external momenta, while $L = 2$, so we need

$$M = EL + \frac{L(L+1)}{2} = 9 \tag{294}$$

Baikov's variables. Seven are provided by the true denominators

$$D_1 = k_1^2, \qquad D_2 = (k_1 - p_1)^2, \qquad D_3 = (k_1 - p_1 - p_2)^2, \qquad D_4 = (k_1 - k_2)^2,$$
$$D_5 = (k_2 - p_1 - p_2)^2, \qquad D_6 = (k_2 - p_1 - p_2 - p_3)^2, \qquad D_7 = k_2^2, \tag{295}$$

while the remaining two are given by the fake denominators

$$D_8 = (k_2 - p_1)^2, \qquad D_9 = (k_1 - p_1 - p_2 - p_3)^2. \tag{296}$$

Using the Mandelstam's variables,[11]

$$s = 2p_1 \cdot p_2, \qquad t = 2p_2 \cdot p_3, \tag{297}$$

the Baikov polynomials $B$ in the variables $z_j = D_j$, $j = 1, \ldots, 9$, is given by

$$
\begin{aligned}
\frac{16B}{(-st(s+t))^{\frac{4-d}{2}}} =\ & st^2z_1z_3 + s^2t^2z_4 - st^2z_1z_4 - st^2z_2z_4 - st^2z_3z_4 + (s^2t + st^2)z_4^2 - st^2z_1z_5 + t^2z_1^2z_5 + stz_1z_2z_5 \\
& - t^2z_1z_3z - st^2z_4z_5 - 2stz_1z_4z_5 - t^2z_1z_4z_5 + stz_2z_4z_5 + t^2z_3z_4z_5 + t^2z_1z_5^2 + tz_1^2z_5^2 \\
& - tz_1z_2z_5^2 - s^2tz_2z_6 + stz_1z_2z_6 + s^2z_2^2z_6 - 2stz_1z_3z_6 + stz_2z_3z_6 - s^2tz_4z_6 + stz_1z_4z_6 \\
& - s^2z_2z_4z_6 - 2stz_2z_4z_6 + stz_3z_4z_6 + stz_1z_5z_6 - tz_1^2z_5z_6 + stz_2z_5z_6 + (s+t)z_1z_2z_5z_6 \\
& - sz_2^2z_5z_6 + tz_1z_3z_5z_6 - tz_2z_3z_5z_6 + s^2z_2^2z_6^2 - sz_1z_2z_6^2 + sz_2^2z_6^2 + sz_1z_3z_6^2 \\
& - sz_2z_3z_6^2 - st^2z_3z_7 - t^2z_1z_3z_7 + stz_2z_3z_7 + t^2z_3^2z_7 - st^2z_4z_7 + t^2z_1z_4z_7 + stz_2z_4z_7 \\
& - 2stz_3z_4z_7 - t^2z_3z_4z_7 + st^2z_5z_7 - t^2z_1z_5z_7 - 2stz_2z_5z_7 + tz_1z_2z_5z_7 + sz_2^2z_5z_7 \\
& - t^2z_3z_5z_7 - 2tz_1z_3z_5z_7 + tz_2z_3z_5z_7 + stz_2z_6z_7 - tz_1z_2z_6z_7 - sz_2^2z_6z_7 + stz_3z_6z_7 \\
& + tz_1z_3z_6z_7 + (s+t)z_2z_3z_6z_7 - tz_3^2z_6z_7 + t^2z_3z_7^2 - tz_2z_3z_7^2 + tz_3^2z_7^2 - 2stz_1z_3z_8 \\
& - s^2tz_4z_8 + stz_1z_4z_8 + s^2z_2z_4z_8 + stz_3z_4z_8 + stz_1z_5z_8 - tz_1^2z_5z_8 - sz_1z_2z_5z_8 \\
& + tz_1z_3z_5z_8 + s^2tz_6z_8 - 2stz_1z_6z_8 + tz_1^2z_6z_8 - s^2z_2z_6z_8 + sz_1z_2z_6z_8 - 2stz_3z_6z_8 \\
& - 2(s+t)z_1z_3z_6z_8 - sz_2z_3z_6z_8 + tz_3^2z_6z_8 + stz_3z_7z_8 + tz_1z_3z_7z_8 - sz_2z_3z_7z_8 \\
& - tz_3^2z_7z_8 + sz_1z_3z_8^2 + s^2tz_2z_9 - s^2tz_4z_9 + stz_1z_5z_9 - 2stz_2z_5z_9 + stz_4z_5z_9 \\
& - tz_1z_5^2z_9 + tz_2z_5^2z_9 - s^2z_2z_6z_9 + s^2z_4z_6z_9 - sz_1z_5z_6z_9 + sz_2z_5z_6z_9 - 2stz_2z_7z_9 \\
& + stz_3z_7z_9 + stz_4z_7z_9 - 2stz_5z_7z_9 + tz_1z_5z_7z_9 - 2(s+t)z_2z_5z_7z_9 + tz_3z_5z_7z_9 + sz_2z_6z_7z_9 \\
& - sz_3z_6z_7z_9 + tz_2z_7^2z_9 - tz_3z_7^2z_9 - s^2tz_8z_9 + stz_1z_8z_9 - s^2z_2z_8z_9 + stz_3z_8z_9 \\
& - s^2z_4z_8z_9 - 2stz_4z_8z_9 + stz_5z_8z_9 + sz_1z_5z_8z_9 + tz_1z_5z_8z_9 + sz_2z_5z_8z_9 - tz_3z_5z_8z_9 \\
& - s^2z_6z_8z_9 + sz_1z_6z_8z_9 - 2sz_2z_6z_8z_9 + sz_3z_6z_8z_9 + stz_7z_8z_9 - tz_1z_7z_8z_9 + sz_2z_7z_8z_9 \\
& + (s+t)z_3z_7z_8z_9 + s^2z_8^2z_9 - sz_1z_8^2z_9 - sz_3z_8^2z_9 + sz_5z_7z_9^2 + s^2z_8z_9^2 - sz_5z_8z_9^2 \\
& - sz_7z_8z_9^2 + sz_8^2z_9^2.
\end{aligned}
\tag{298}
$$

However, we will not work with the complete diagram but on the maximal cut, which correspond, to set $z_j = 0$, $j = 1, \ldots, 7$ so that only the denominators $D_8$ and $D_9$ survive. Using $w_1 = z_8$ and $w_2 = z_9$, we then get for the maximal cut

$$
B(w_1, w_2) = \frac{(-st(s+t))^{\frac{4-d}{2}}}{16}\left[-s^2tw_1w_2 + s^2w_1^2w_2 + s^2w_1w_2^2 + sw_1^2w_2^2\right].
\tag{299}
$$

The twisting section is thus

$$
u = B^\gamma
\tag{300}
$$

with

$$
\gamma = \frac{d - E - L - 1}{2} = \frac{d - 6}{2}.
\tag{301}
$$

The associated connection 1-form is

$$
\omega = \gamma\frac{-st + 2sw_1 + sw_2 + 2w_1w_2}{-stw_1 + sw_1^2 + sw_1w_2 + w_1^2w_2}dw_1 + \gamma\frac{-st + sw_1 + 2sw_2 + 2w_1w_2}{-stw_2 + sw_1w_2 + sw_2^2 + w_1w_2^2}dw_2.
\tag{302}
$$

The equation $\omega = 0$ has two solutions

$$
w_1 = w_2 = w_\pm \equiv -\frac{3}{4}s \pm \sqrt{\frac{9}{16}s^2 + \frac{1}{2}st},
\tag{303}
$$

so there are two MIs. Since we have to compute bivariate intersection numbers, we consider $w_1$ as internal integration variable. The number of corresponding MIs is given by

the solutions of $\omega_1 = 0$ for $w_1$, where $\omega = \omega_1 dw_1 + \omega_2 dw_2$. Therefore, we need just one MI for the internal integration.

As MIs we choose $I_{[1,1,1,1,1,1,1,0,0]}$ and $I_{[1,1,1,1,1,1,1,-1,0]}$, which correspond to the forms

$$\langle e_1| \equiv \langle 1|, \qquad \langle e_2| \equiv \langle w_1|, \tag{304}$$

respectively. Considering them as functions of the external parameter $s$, they have to satisfy the differential equation system

$$\langle \partial_s e_j + (\partial_s \log u)e_j| = \sum_{k=1}^{2} O_{jk}\langle e_k| \tag{305}$$

where the matrix $\boldsymbol{O}$ is obtained as follows (see [61], Section 3.2). Let $\phi_{sj} := \partial_s e_j + (\partial_s \log u)e_j$, $F_{jk} = \langle \phi_{sj}|d_k \rangle$ and $C_{jk} = \langle e_j|d_k \rangle$ define the matrices $\boldsymbol{F}$ and $\boldsymbol{C}$. Then,

$$\boldsymbol{O} = \boldsymbol{F}\boldsymbol{C}^{-1}. \tag{306}$$

We will then show how to compute $\boldsymbol{F}$ and $\boldsymbol{C}$, being very explicit for $C_{11}$ and quoting the results for all the other components. Notice that in our specific case $\partial_s e_j = 0$ and

$$\boldsymbol{\phi}_s = \frac{\gamma}{s} \begin{pmatrix} \frac{2s(t-w_1-w_2)-w_1 w_2}{s(t-w_1-w_2)-w_1 w_2} \\ \frac{w_1(2s(t-w_1-w_2)-w_1 w_2)}{s(t-w_1-w_2)-w_1 w_2} \end{pmatrix}. \tag{307}$$

We choose $d_j = e_j$.

### 8.1. Computation of $\boldsymbol{C}$

Let us start by computing $\langle 1|1 \rangle$ very explicitly. According to the general construction, we need to start with the internal integration; therefore, we choose an internal basis (for $w_1$ only). Recalling that the dimension is 1, we choose

$$\langle \varepsilon| = \langle 1/w_1| \tag{308}$$

and $\delta = \varepsilon$ for the dual. The critical point of $d_{w_1} \log u$ is

$$w_1 = \bar{w} \equiv \frac{s}{2}\frac{t-w_2}{s+w_2}. \tag{309}$$

Since $\varepsilon$ is a $d$ log-form, the reduced intersection matrix $\boldsymbol{C}_{red} \equiv \langle \varepsilon|\delta \rangle$ can be computed with the simple method of univariate intersection numbers for $d$ log-forms. From (254), we get

$$\boldsymbol{C}_{red} = -\text{Res}_{w_1=\bar{w}}\left( \frac{1}{w_1^2}\frac{1}{\omega_1} \right) = -\frac{1}{\bar{w}^2}\frac{1}{\partial_{w_1}\omega_1|_{w_1=\bar{w}}} = \frac{1}{2\gamma}. \tag{310}$$

To proceed iteratively, as we have seen for the recursive calculation of the multivariate intersection numbers, we have now to determine the new connection (260), which is now[12]

$$\Omega^{(2)} = \langle \partial_{w_2}\varepsilon + \omega_2\varepsilon|\delta \rangle \boldsymbol{C}_{red}^{-1}. \tag{311}$$

Using again (254), we get

$$\Omega^{(2)} = -2\gamma\text{Res}_{w_1=\bar{w}}\left( \frac{\omega_2}{w_1^2}\frac{1}{\omega_1} \right) = \frac{\gamma}{w_2} - \frac{\gamma}{w_2+s} + \frac{2\gamma}{w_2-t}. \tag{312}$$

Notice that $\Omega^{(2)}$ has only simple poles and its zeros are given by (303).

Next, we need to project $\langle \phi_L | \equiv \langle e_1 | = \langle 1 |$ to $\langle \varepsilon |$:

$$\left\langle \phi_L^{(2)} \right| := \langle \phi_L | \delta \rangle \mathbf{C}_{red}^{-1}. \tag{313}$$

This computation is not as direct as the previous ones, since $\phi_L$ is not of $d$ log-type, and it has a double pole at infinity: for $w_1 \to 1/w_1$ we have $\langle 1 | \to \langle -1/w_1^2 |$. However, we can use the cohomological property of the intersection numbers and shift $\phi_L$ by a $\nabla_{\omega_1}$-exact form to get a representative in the same class with only simple poles. For example

$$\tilde{\phi}_L = \phi_L - \nabla_{\omega_1} \frac{w_1}{1 + 2\gamma} = \frac{s\gamma}{1 + 2\gamma} \frac{w_2 - t}{-st + sw_1 + sw_2 + w_1 w_2} \tag{314}$$

has two simple poles, one at $w_1 = \infty$ and the other one at

$$w_1 = s \frac{t - w_2}{s + w_1}.$$

Since cohomologous formes give equal intersection numbers, we can write

$$\left\langle \phi_L^{(2)} \right| = \langle \tilde{\phi}_L | \delta \rangle \mathbf{C}_{red}^{-1} \tag{315}$$

and use once again (254) to get

$$\left\langle \phi_L^{(2)} \right| = -2\gamma \operatorname{Res}_{w_1 = \bar{w}} \left( \frac{\tilde{\phi}_L}{w_1} \frac{1}{\omega_1} \right) = \frac{s\gamma}{1 + 2\gamma} \frac{t - w_2}{s + w_2}. \tag{316}$$

In a similar way, we have to project $| \phi_R \rangle \equiv | e_1 \rangle = | 1 \rangle$ on $| \delta \rangle$:

$$\left| \phi_R^{(2)} \right\rangle := \mathbf{C}_{red}^{-1} \langle \varepsilon | \phi_R \rangle. \tag{317}$$

As before, $\phi_R$ has a double pole at infinity so we replace it with the representative[13]

$$\tilde{\phi}_R = \phi_L - \nabla_{-\omega_1} \frac{w_1}{1 - 2\gamma} = \frac{s\gamma}{(2\gamma - 1)} \frac{w_2 - t}{-st + sw_1 + sw_2 + w_1 w_2}, \tag{318}$$

to get

$$\left| \phi_R^{(2)} \right\rangle = -2\gamma \operatorname{Res}_{w_1 = \bar{w}} \left( \frac{\varepsilon \tilde{\phi}_R}{\omega_1} \right) = -\frac{s\gamma}{1 - 2\gamma} \frac{t - w_2}{s + w_2}. \tag{319}$$

Before considering the final step of the computation of $C_{11}$, which involves the intersection between $\phi_L^{(2)}$ and $\phi_R^{(2)}$ twisted with $\Omega^{(2)}$, we notice that the last two again have a double pole at infinity. Once again, we change the cohomology representatives in order to work with forms having only simple poles. Since

$$\phi_L^{(2)} = -\frac{s\gamma}{1 + 2\gamma} + \frac{\gamma s(s + t)}{(1 + 2\gamma)(s + 2w_2)}, \tag{320}$$

we conveniently define

$$\tilde{\phi}_L^{(2)} = \phi_L^{(2)} - \nabla_{\Omega^{(2)}} \left( -\frac{s\gamma}{1 + 2\gamma} \frac{w_2}{1 + 2\gamma} \right) = \frac{2st\gamma^2}{(1 + 2\gamma)^2 (w_2 - t)} + \frac{\gamma s(s + t) + 3s^2\gamma^2 + 2\gamma^2 st}{(1 + 2\gamma)^2 (s + w_2)}. \tag{321}$$

Similarly we set

$$\tilde{\phi}_R^{(2)} = \phi_R^{(2)} - \nabla_{-\Omega^{(2)}} \left( \frac{s\gamma}{1 - 2\gamma} \frac{w_2}{1 - 2\gamma} \right) = \frac{2st\gamma^2}{(1 - 2\gamma)^2 (w_2 - t)} + \frac{-\gamma s(s + t) + 3s^2\gamma^2 + 2\gamma^2 st}{(1 - 2\gamma)^2 (s + w_2)}. \tag{322}$$

We can finally use (270) to compute the final step[14]

$$
\begin{aligned}
\boldsymbol{C}_{11} &= \langle 1|1 \rangle = -\mathrm{Res}_{w_2=w_+}\left(\frac{\tilde{\phi}_L^{(2)}\tilde{\phi}_R^{(2)}}{\Omega^{(2)}}\boldsymbol{C}_{red}\right) - \mathrm{Res}_{w_2=w_-}\left(\frac{\tilde{\phi}_L^{(2)}\tilde{\phi}_R^{(2)}}{\Omega^{(2)}}\boldsymbol{C}_{red}\right) \\
&= s^2\frac{3(1-8\gamma^2)(s+t)^2 - 3\gamma^2 s^2 + 4\gamma^2 t^2}{4(1-4\gamma^2)^2}.
\end{aligned}
\tag{323}
$$

In the same way, we can compute the remaining matrix elements of which we just quote the final results

$$
\begin{aligned}
\boldsymbol{C}_{12} &= \langle 1|w_1 \rangle = \frac{s^2}{8(1-\gamma)(1-4\gamma^2)^2}\Big[2t^3\gamma(1-4\gamma^2) - 2st^2(3-7\gamma-22\gamma^2+44\gamma^3) \\
&\quad -3s^2t(4-7\gamma-33\gamma^2+54\gamma^3) - s^3(6-9\gamma-54\gamma^2+81\gamma^3)\Big],
\end{aligned}
\tag{324}
$$

$$
\begin{aligned}
\boldsymbol{C}_{21} &= \langle w_1|1 \rangle = -\frac{s^2}{8(1+\gamma)(1-4\gamma^2)^2}\Big[2t^3\gamma(1-4\gamma^2) + 2st^2(3+7\gamma-22\gamma^2-44\gamma^3) \\
&\quad +3s^2t(4+7\gamma-33\gamma^2-54\gamma^3) + s^3(6+9\gamma-54\gamma^2-81\gamma^3)\Big] = \boldsymbol{C}_{12}(\gamma\to-\gamma),
\end{aligned}
\tag{325}
$$

$$
\begin{aligned}
\boldsymbol{C}_{22} &= \langle w_1|w_1 \rangle = \frac{s^3}{16(1-4\gamma^2)^2(1-\gamma^2)}\Big[-16t^3\gamma^2(1-4\gamma^2) + 12st^2(1-13\gamma^2+30\gamma^4) \\
&\quad +12s^2t(2-23\gamma^2+45\gamma^4) + 3s^3(4-45\gamma^2+81\gamma^4)\Big].
\end{aligned}
\tag{326}
$$

### 8.2. Computation of *F*

This is computed exactly in the same way, now choosing $\phi_L = \phi_{sj}$ and $\phi_R = d_k$. Again, we will not repeat all passages here, but limit ourselves to quoting the final results. Of course, the interested reader is invited to reproduce all the details carefully. We get

$$
\begin{aligned}
\boldsymbol{F}_{11} &= \langle \phi_{s1}|1 \rangle = \frac{s}{4(1-4\gamma^2)^2}\Big[3s^2(2+5\gamma-18\gamma^2-45\gamma^3) + t^2(3+11\gamma-20\gamma^2-68\gamma^3) \\
&\quad +3st(3+9\gamma-24\gamma^2-68\gamma^3)\Big],
\end{aligned}
\tag{327}
$$

$$
\begin{aligned}
\boldsymbol{F}_{12} &= \langle \phi_{s1}|w_1 \rangle = \frac{s}{8(1-\gamma)(1-4\gamma^2)^2}\Big[2t^3\gamma(1+3\gamma-4\gamma^2-12\gamma^3) - 3s^3(4+4\gamma-51\gamma^2-36\gamma^3+135\gamma^4) \\
&\quad -2st^2(3+3\gamma-50\gamma^2-26\gamma^3+160\gamma^4) - 3s^2t(6+7\gamma-82\gamma^2-57\gamma^3+234\gamma^4)\Big],
\end{aligned}
\tag{328}
$$

$$
\begin{aligned}
\boldsymbol{F}_{21} &= \langle \phi_{s2}|1 \rangle = \frac{s}{8(1+\gamma)(1-4\gamma^2)^2}\Big[2t^3\gamma(-1-3\gamma+4\gamma^2+12\gamma^3) + 4st^2(-3-12\gamma+8\gamma^2+79\gamma^3+80\gamma^4) \\
&\quad +3s^3(-6-19\gamma+39\gamma^2+171\gamma^3+135\gamma^4) + 3s^2t(-10-35\gamma+50\gamma^2+273\gamma^3+234\gamma^4)\Big],
\end{aligned}
\tag{329}
$$

$$
\begin{aligned}
\boldsymbol{F}_{22} &= \langle \phi_{s2}|w-1 \rangle = \frac{s^2}{16(1-\gamma)(1-4\gamma^2)^2}\Big[4t^3\gamma(-1-6\gamma-9\gamma^2+24\gamma^3+52\gamma^4) \\
&\quad + 12st^2(2+3\gamma-26\gamma^2-45\gamma^3+60\gamma^4+114\gamma^5) + 6s^2t(10+17\gamma-115\gamma^2-197\gamma^3+225\gamma^4+396\gamma^5) \\
&\quad +3s^3(12+20\gamma-135\gamma^2-225\gamma^3+243\gamma^4+405\gamma^5)\Big].
\end{aligned}
\tag{330}
$$

### 8.3. Computation of *O*

With these elements, we can finally compute the matrix $\boldsymbol{O} = \boldsymbol{F}\boldsymbol{C}^{-1}$ defining the differential equation for the MIs:

$$
\boldsymbol{O} = \frac{1}{s(s+t)}\begin{pmatrix} -(2s+(d-6)t) & 4-d \\ (4-d)\frac{t}{2} & (-16+3d)\frac{s}{2}+(-6+d)t \end{pmatrix}.
\tag{331}
$$

Of course we could proceed to work out the present example, but our scope, illustrating how to apply intersection theory to Feynman calculations, is already reached, so

we leave further computation to the interested readers. Of course, exploiting all these calculations by hand is lengthy and requires a lot of time, but the major advantage of these methods is that they can be implemented on a computer making it very quickly.

### 9. Final Comments

The strategy we followed here is receiving growing interest at the time we are writing the present review. The ability of recognizing the correct cohomology underlying the Feynman Integrals and the corresponding intersection product would allow both to gain deeper understanding of such integrals and to systematize their computation. In particular, after determining the cohomology space, one is left with the choice of a convenient basis, i.e., the master integrals, with respect to which one can project the given amplitude, through the intersection product.

In Section 6.3, we argued the necessity of introducing the vector space $IH_M^{(p)}$ of twisted cohomology with perversity, in order to correctly deal with singular forms; in Section 3 we showed different ways in which the dimension $\nu$ of such space can be computed and interpreted. Such interpretations mostly rely on the Poincaré duality between cycles and cocycles, which in the case of singular varieties is restored by the introduction of perverse sheaves. While in principle any set of $\nu$ independent forms in $IH_M^{(p)}$ could serve as a basis, the determination of a preferred one suitable for practical computation is still an open problem. We remark that a natural basis of cycles is represented by the Lefschetz thimbles, as we described at the end of Section 6.4. This allows us in principle to define a basis also for the cohomology, thanks to Poincaré duality. Here our excursus ends: the practical realization of such duality has yet to be performed in a form apt to get a systematic approach in the problem of the calculation of Feynman Integrals.

**Author Contributions:** S.L.C., M.C. and S.T. have contributed equally to the present review. All authors have read and agreed to the published version of the manuscript.

**Funding:** This research received no external funding

**Institutional Review Board Statement:** Not applicable.

**Informed Consent Statement:** Not applicable.

**Acknowledgments:** First of all, we are indebted to Pierpaolo Mastrolia and the group of Padua (Manoj Mandal, Federico Gasparotto, Luca Mattiazzi, Vsevolod Chestnov, Hjalte Frellesvig and Co.) for introducing us to the topic of intersection theory methods in the Feynman integrals landscape. We also thank Pierpaolo for helping us in considerably improving our manuscript and adding content to it. We also thank Maxim Kontsevich for their lesson related to Section 6.4 and Thibault Damour for helpful discussions. In particular, M.C. is grateful to Roberta Merlo for sharing her deep knowledge about spheres with handles. We are particular indebted to Manoj Mandal and Federico Gasparotto for providing us with the example of Section 8, including all the detailed calculations shown there.

**Conflicts of Interest:** The authors declare no conflict of interest.

### Appendix A. Baikov Representation

For the sake of completeness, in this Appendix we provide a short derivation of the Baikov Formula (2). The idea is to rewrite the integral using the independent scalar products between momenta as integration variables. They are

$$M = LE + L(L+1)/2, \tag{A1}$$

where the first term is the number of scalar products $q_i \cdot p_j$ between one loop and one external momentum, while the second represents the scalar products $q_i \cdot q_j$ between loop momenta.

In total, one has $m = E + L$ total independent momenta. It is useful to introduce the complete vector $k$ of all $m$ momenta:

$$k = (\underbrace{q_1, \cdots q_L}_{K_i = q_i \ i \le L} \ , \ \underbrace{p_1, \cdots p_E}_{K_i = p_{i-L} \ i > L}). \tag{A2}$$

To perform this change in variables, we start by decomposing $q_1$ as

$$q_1 = q_{1\parallel} + q_{1\perp}, \tag{A3}$$

where $q_{1\parallel}$ represents the projection of $q_1$ onto the space generated by all the other $m - 1 = E + L - 1$ momenta $\{q_2, \cdots q_L, p_1, \cdots, p_E\}$, while $q_{1\perp}$ represents the orthogonal component with respect to said space. This is done for every variable: for each $i$, the corresponding $q_i$ is projected onto the space generated by the momenta that come next in the vector $k$: $\{q_{i+1}, \cdots q_L, p_1, \cdots, p_E\}$. In the last step, one projects the last $q_L$ along the space of the $E$ external momenta $\{p_1, \cdots, p_E\}$. This decomposition leads to

$$d^D q_1 \cdots d^D q_L = (d^{E+L-1} q_{1\parallel} \ d^{D-E-L+1} q_{1\perp}) \ \cdots \ (d^E q_{L\parallel} \ d^{D-E} q_{L\perp}). \tag{A4}$$

We introduce the Gram matrix

$$G(k) = \begin{pmatrix} s_{11} & \cdots & s_{1m} \\ \vdots & & \vdots \\ s_{m1} & \cdots & s_{mm} \end{pmatrix}, \tag{A5}$$

in which the entries are $s_{ij} = k_i k_j$. The square root of the determinant (which we will address as $G$ instead of $\det(G)$ for brevity) of the matrix (A5) represents the volume of the parallelotope generated by the elements of $k$. With this interpretation, one can write

1:     Parallel component $q_{i\parallel}$.

$$d^{E+L-i} q_{i\parallel} = \frac{ds_{i,i+1} \cdot ds_{i,i+2} \cdots ds_{i,E+L}}{G^{1/2}(q_{i+1}, \cdots, q_L, p_1, \cdots, p_E)} = \prod_{j=i+1}^{E+L} \frac{ds_{ij}}{G^{1/2}(q_{i+1}, \cdots, q_L, p_1, \cdots, p_E)}. \tag{A6}$$

In the numerator of Equation (A6), we perform the scalar product of $q_i$ along the space generated by the vectors that come next (starting from $q_{i+1}$): this allows us to find the projections of $q_i$ along such vectors. The denominator is the necessary normalization that allows us to get the correct dimension.

2:     Perpendicular component $q_{i\perp}$.

Introducing polar coordinates and separating the angular part from the radial part, we get

$$d^{D-E-L+i} q_{i\perp} = \Omega_{D-E-L+i-1} |q_{i\perp}|^{D-E-L+i-1} d|q_{i\perp}| = \frac{\Omega_{D-E-L+i-1}}{2} |q_{i\perp}|^{D-E-L+i-2} d|q_{i\perp}|^2, \tag{A7}$$

where $d|q_{i\perp}|^2$ is $ds_{ii}$. Notice that $|q_{i\perp}|$ can be seen as the height of a parallelotope with base $q_{i+1}, \cdots, p_E$, so it can be computed as the whole volume divided by the volume of its base: hence, we write

$$d^{D-E-L+i} q_{i\perp} = \frac{\Omega_{D-E-L+i-1}}{2} \left( \frac{G(q_i, \cdots q_L, p_1, \cdots, p_E)}{G(q_{i+1}, \cdots q_L, p_1, \cdots, p_E)} \right)^{\frac{D-E-L-2+i}{2}} ds_{ii}. \tag{A8}$$

Using Equation (A6) and Equation (A8) and recalling

$$\frac{\Omega_{D-E-L+i-1}}{2} = \frac{\pi^{\frac{D-E-L+i}{2}}}{\Gamma(\frac{D-E-L+i}{2})}, \tag{A9}$$

it is possible to obtain

$$\frac{d^D q_i}{\pi^D/2} = \frac{\pi^{\frac{-E-L+i}{2}}}{\Gamma\left(\frac{D-E-L+i}{2}\right)} \int_{\Gamma_i} G^{-1/2}(q_{i+1}, \cdots, p_E) \prod_{j=i}^{E+L} d(q_i k_j) \left( \frac{G(q_i, \cdots, p_E)}{G(q_{i+1}, \cdots p_E)} \right)^{\frac{D-E-L-2+i}{2}}, \tag{A10}$$

where $\Gamma_i$ is the contour determined by $|q_{i\perp}|^2 > 0$. The whole Feynman integral (1) becomes

$$I_{\nu_1, \cdots, \nu_N} = \frac{\pi^{\frac{L-M}{2}}}{\prod_{l=1}^L \Gamma\left(\frac{D-E-L+l}{2}\right)} \int_{\Gamma} \prod_{i=1}^L G^{-1/2}(q_{i+1}, \cdots, p_E) \left( \frac{G(q_i, \cdots, p_E)}{G(q_{i+1}, \cdots p_E)} \right)^{\frac{D-E-L-2+i}{2}} \frac{\prod_{j=i}^{E+L} d(q_i k_j)}{\prod_{a=1}^N D_a^{\nu_a}}. \tag{A11}$$

By explicit computation of the product over $i$ in Equation (A11), only a few terms survive and Equation (A11) becomes

$$I_{\nu_1, \cdots, \nu_N} = \frac{\pi^{\frac{L-M}{2}} G(p_1, \cdots, p_E)^{-\frac{L}{2}}}{\prod_{i=1}^L \Gamma\left(\frac{D-E-L+i}{2}\right)} \int_{\Gamma} \left( \frac{G(q_1, \cdots, p_E)}{G(p_1, \cdots p_E)} \right)^{\frac{D-E-L-1}{2}} \frac{\prod_{i=1}^L \prod_{j=1}^{E+L} d(q_i k_j)}{\prod_{a=1}^N D_a^{\nu_a}}$$

$$\equiv C \int_{\Gamma} B^{\gamma} \frac{\prod_{i=1}^L \prod_{j=1}^{E+L} d(q_i k_j)}{\prod_{a=1}^N D_a^{\nu_a}}. \tag{A12}$$

In the second line of Equation (A12), $B$ is the *Baikov polynomial* (similar to the Jacobian associated to the change of variables) and is raised to the power $\gamma = (D - E - L - 1)/2$, while $C$ is an overall constant. Equation (A12) can be further simplified by noticing that in general the propagators $D_a$ in the denominator usually depend linearly on the scalar products between momenta; hence, one can perform a change of variables $z_a = D_a$. Notice however that the number of denominators $N$ and the the number of independent momenta scalar products $M$ are different in general: for this reason, it is sufficient to add $N - M$ fake denominators $D_a$, each one raised to a certain power $\nu_a$. The original integral can be recovered by putting $\nu_a = 0$ for $a = N + 1, \cdots, M$. With this procedure, one readily obtains the final form (2), where the constant $K = C/\det A$, and $A$ is the Jacobian matrix between the scalar products and the propagators $D_a = A_a^{ij} q_i k_j + m_a^2$.

## Appendix B. An Introduction on Gröbner Bases

In this Appendix, we introduce some basic tools, along with some notations and definitions, necessary to understand how to work with Gröbner bases. We start by calling $R = \mathbb{K}(z_1, \cdots, z_n)$ the ring over a field $\mathbb{K}$ with $n$ variables $z_1, \cdots, z_n$, with $I \subset R$ being an ideal in the ring $R$.

**Definition A1.** *Combination of polynomials.*

*Given a set of polynomials, an expression consisting of the sum of polynomial multiples of the elements in the set is called a combination.*

For example, $(3z_1 + z_2^2)(z_1 + z_3) + z_1(z_1^2 + 5z_2^2 z_3^2)$ is a combination of the polynomials $\{z_1 + z_3, z_1^2 + 5z_2^2 z_3^2\}$.

**Definition A2.** *Power product.*

*A power product is a polynomial that can be obtained only by multiplication of the variables.*

For example, $z_1 z_2^2$ is a power product.

**Definition A3.** *Term order.*

*A term order on the monomials of a ring $R$ is an order $\prec$ with the following properties:*

1: $M \prec N \Leftrightarrow MP \prec NP$
2: $M \preccurlyeq MP$

For example, the following order is called a *Lexicographic order* on the variables $z_1, z_2$:

$$1 \prec z_1 \prec z_1^2 \prec z_1^3 \prec \cdots \prec z_2 \prec z_1 z_2 \prec z_1^2 z_2 \prec \cdots z_2^2 \prec z_1 z_2^2 \prec z_1^2 z_2^2 \prec \cdots \tag{A13}$$

**Definition A4.** *Initial term (or leading power product).*
*Given a ring R with a term order $\prec$, the initial term $in_{\succ}(f)$ of a polynomial $f \in R$ (or leading power product $LLP_{\succ}(f)$) is the largest monomial in f with respect to the given order, together with its coefficient.*

For example, if $z_1 \prec z_2$ and $f = 3z_1 z_2 + 2z_2^2$, then $in_{\succ}(f) = 2z_2^2$.

**Definition A5.** *Initial Ideal.*
*Given an ideal $I \subset R$, $in_{\succ}(I)$ is the monomial ideal generated by the initial terms $\{in_{\succ}(f) : f \in I\}$.*

**Definition A6.** *Gröbner basis.*
*Given an ideal $I \subset R$ with a term order $\prec$ and a set of polynomials $G = \{g_1, \cdots, g_s\} \subset I$, if $in_{\succ}(I)$ is generated by $\{in_{\succ}(g_1), \cdots, in_{\succ}(g_s)\}$ then G is called a Gröbner basis with respect to the order.*

Definition A6 implies that, given a $f \in I$ with $f \neq 0$, $in_{\succ}(f)$ must be divisible by at least one of the $in_{\succ}(g_i)$ in the basis.

We also introduce some concepts that will be useful in our context. First, we introduce the algorithm of *division of a polynomial by a given set of polynomials*. This concept arises from the following question: given an ideal $I \subset R$, when is an $h \in R$ also $h \in I$? Suppose we have a Gröbner basis for $I$: if $h \in I$, then its leading power product must be divisible by one of the leading power products appearing in the basis (the one belonging to a certain polynomial $g_i$ to fix ideas). The subtraction of an appropriate multiple of $g_i$ from $h$ leads to a new $\tilde{h}$, where the old leading term has been canceled. If $h \in I$, then also $\tilde{h} \in I$: the process can be reiterated until 0 is obtained (hence $h \in I$ is verified). If the iterated process leads to a polynomial $\neq 0$ that cannot be further reduced, then $h \notin I$. The result of this procedure of division of a polynomial $f$ by a set of polynomials $G$ is called the *remainder $R_G(f)$*. Notice that this procedure must be finite, as the definition of a term order $\prec$ implies a descending path in the variables that has to end (this is not true for a generic set of polynomials that is not a Gröbner basis for $I$, as one should check infinite types of combinations of polynomials in principle).

Notice that the concept of remainder can be used to determine if two polynomials $f, g$ in $R$ are equal modulo $I$ (i.e., $f \sim g$ if $f +$ combination of generators of $I = g$): in general, one should check infinite combinations of the generators, but with a Gröbner basis for $I$:

- $G$ is a Gröbner basis for $I \Leftrightarrow R_G(f) = 0 \; \forall \; f \in I$;
- If $G$ is a Gröbner basis for $I$, then $f, g \in R$ are equal modulo $I \Leftrightarrow R_G(f) = R_G(g)$.

With these tools, it is possible to construct a practical way that tells us how to build Gröbner bases.

**Definition A7.** *S-polynomial.*
*Given two polynomials $f, g$, the S-polynomial between f and g is defined as:*

$$S(f,g) = lcm(\tilde{in}_{\succ}(f), \tilde{in}_{\succ}(g)) \left( \frac{f}{in_{\succ}(f)} - \frac{g}{in_{\succ}(g)} \right), \tag{A14}$$

*where the notation $\tilde{in}_{\succ}(f)$ means $in_{\succ}(f)$ deprived of its coefficient.*

Notice that the S-polynomial of two polynomials $f, g$ is a combination of $f$ and $g$: because of the previous observations, if a set $G = \{g_1, \cdots, g_s\}$ is Gröbner basis, then $\forall \; i \neq j$ $R_G(S(g_i, g_j)) = 0$. This requirement is the fundamental of *Buchberger's Algorithm* [145],

which, starting from a certain set of polynomials $\{f_1, \cdots, f_t\}$ that generate an ideal $I$, returns a Gröbner basis for the same ideal, at a certain given order. The algorithm works as follows: given $F = \{f_1, \cdots, f_t\}$, it computes $S(f_1, f_2)$; if $R_F(S(f_1, f_2)) = 0$, and then it proceeds to the next couple of elements ($f_1$ and $f_3$, for instance). If $R_F(S(f_1, f_2)) \neq 0$, $F$ is not a Gröbner basis, it then adds $S(f_1, f_2)$ - or better yet, its remainder $R_F(S(f_1, f_2))$, as a new element of $F$, and starts again. Notice that the addition of the new term to $F$ ensures that in the next iteration $R_F(S(f_1, f_2)) = 0$. At the end of the process, $F$ forms a Gröbner basis.

## Notes

[1] Here one has to be careful: if we take a sphere with a hole, its boundary is not a boundary of the hole, but it is for its complement. In the above example this is not so.

[2] If we cut before passing the saddle point, we get two cylinders, which correspond to trivial pieces.

[3] $\gamma$ runs along $[-1, 1]$ twice, so a further factor $\frac{1}{2}$ appears.

[4] $(x, y)$ are non homogeneous coordinates on $\mathbb{P}^2$ that can be related to homogeneous ones $(z_0 : z_1 : z_2)$ by $x = z_1/z_0$, $y = z_2/z_0$ on the patch $z_0 \neq 0$.

[5] For example, assume $P(x) = a(x - x_1)(x - x_2)(x - x_3)(x - x_4)$, and take the transformation

$$x(z) = \frac{x_1 z - b}{z - d}, \qquad z(x) = \frac{xd - b}{x - x_1}, \tag{82}$$

which sends $x_1$ to $\infty$. Then, the integral (76) takes the form

$$I_1 = 2\sqrt{\frac{x_1 d - b}{a \prod_{j=2}^4 (x_1 - x_j)}} \int_{z(x_2)}^{\infty} \frac{dz}{\sqrt{4(z - z(x_2))(z - z(x_3))(z - z(x_4))}}. \tag{83}$$

Finally, fixing $b, d$ such that $\sum_{j=2}^4 z(x_j) = 0$, we get the Weierstrass normal form.

[6] Notice that this cannot be the general case, or at least not assuming $b$ and $d$ remain orthogonal to the equipotential lines.

[7] In addition, they do not change deforming the path $a_j$ and $b_j$ continuously.

[8] If we considered the invariants, the result at the point would be that the points meet with total coefficient $1 + 1 = 2$.

[9] Therefore, $[N]$ is defined by any given simplicial decomposition of $N$.

[10] Indeed $\gamma = N/2 + \epsilon$ because of dimensional regularization.

[11] Notice also that the massless condition and momentum conservation imply $s + t = -2p_1 \cdot p_3$.

[12] Notice that we are not using an orthonormal frame, so the inverse matrix $C_{red}^{-1}$ appears.

[13] Notice that we have changed the connection to $-\omega_1$.

[14] once again taking into account of the non othonormality through $C_{red}$.

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
