# Peer review of "Co-Homology of Differential Forms and Feynman Diagrams"

_universe, doi:10.3390/universe7090328_

Round 1
Reviewer 1 Report
The paper "Cohomology of Differential Forms and Feynman Diagrams" (Universe-1341782) by Sergio L. Cacchatori, Maria Conti and Simone Trevisan provides an overview that provides a detailed analysis of the relationship between Feynman integrals and cohomology theories in the light of recent developments.
The article is very long, with a complex topic, but well written and the topic is
very interesting and useful, because it contains results that are not very well
known to many physicists. The results considered can be used by both the authors and other researchers in future calculations of the Feynman integrals.
Therefore, this article can be recommended as a review of the most modern methods for calculating massive Feynman integrals. Indeed, in addition to the standard Feynman parameter method, the article describes in detail (and with a strong mathematical bias) modern methods of working mainly with diagrams containing elliptic forms, the consideration of which has become popular recently.
While studying the article, I did not find any mistakes or shortcomings in the formulas and text.
Thus, I believe that the article "Cohomology of Differential Forms ..." (Universe-1341782) can be published in the journal Universe in its original form.
Author Response
We thank the referee for his/her comments.
We just report here the changes we made to answer the other two referees:
-we have replaced the short abstract with an extended version; -we have made the following changes to avoid overlapping with arXiv:2008.04823: Partially modified the text in lines 126–137 and 144–150 of the Introduction. Partially modified the text lines 208–222, 252, 253, 274–281 of section 3; -we fixed all typos listed by referee 3 plus few more we have found checking the manuscript once again; -we added section 8 with an explicit example of application of intersection theory to Feynman integrals; -we improved Acknowledgements.Reviewer 2 Report
Co-homology of Differential Forms and Feynman diagrams
Sergio L. Cacciatori 1,2, Maria Conti 1,2, Simone Trevisan
In this review, the authors provide a systematic derivation of fundamental concepts in algebraic topology which have been recently found to control the vector space structure of Feynman integrals.
Morse theory, De Rahm co-homology, and more generally the integration of differential forms on Riemann surfaces constitute the mathematical bases of the so called Intersection Theory.
Multiloop Feynman integrals are crucial ingredients of Field Theories, that emerge in the evaluation of scattering amplitudes, and related physical quantities, like cross sections, interaction potentials, impact parameters, scattering angles, and so on.
Intersection Theory for twisted De Rham co-homology offers the proper mathematical framework for studying Feynman integrals in dimensional regularization, and the functional equations they satisfy: linear relations, differential equations, difference equation and quadratic relations.
The study of (co)-homolgical properties of Feynman integrals started already in the mid sixties, with the works of Pham, Hwa, Tepliz, Lefshetz, and many others.
Only recently these formal properties inspired a novel computational method.
This review contains a clear presentation of advanced mathematical concepts which allow a physics-oriented audience to approach what it is considered as a known topic, such as Feynman calculus, under a novel vest, and to familiarise with concepts such as
thimbles, perversites, singular and simpicial co-homologies, and intersection numbers,
to name a few. Using them, the analyticity of scattering amplitudes appear more closely related to geometry, as dictated by graph polynomials associated to Feynman diagrams.
At the same time, special methods, developed in the context of pure mathematics appeared to have direct application in the physical context of elementary particle scattering.
This work clearly present the direct relation of Riemann (twisted) periods and Feynman integrals, in Baikov representation, where the latter expose their structure as Aomoto-Gel'fand integrals.
The review is an important introduction to a novel research channel in theoretical physics, combining Algebraic Geometry, Differential Geometry, Topology, Number Theory, Complex Analysis and Quantum Field Theory, which is currently missing in the literature.
It brings the reader through non-trivial concepts, from the formal developments to the computational tools, till the description of currently open problems, and suggesting interesting future directions.
I consider it worth to be published as Editorial of the Universe scientific journal.
I have just a couple minor comments for the authors, which I hope they could account for, before publication:
1.
Given the relevant content, I would suggest to extend the abstract, which in the current form is very minimal.
2.
Since the review deals with Feynman integrals, it might be useful to add one simple application to a Feynman graph.
Author Response
We thanks the referee for his/her comments. In particular, we are grateful for the suggestion of adding an explicit example of applications to an explicit Feynman graph, which allowed us to get a more complete version of the review.
In particular:
- We extended the abstract, we hope in a satisfactory way.
- We have added a new section 8 containing applications of intersection theory to a massless double box Feynman diagram. We have shown a part of the calculation very explicitly, proposing the remaining part as possible exercises for the readers.
Moreover, we also point out that we included few further changes in order to satisfy the requests of a another referee:
-we have made the following changes to avoid overlapping with arXiv:2008.04823: Partially modified the text in lines 126–137 and 144–150 of the Introduction. Partially modified the text lines 208–222, 252, 253, 274–281 of section 3; -we fixed all typos listed by referee 3 plus few more we have found checking the manuscript once again; Finally, we improved the Acknowledgements.Reviewer 3 Report
This manuscript is a review of the recent developments in the area of Feynman integrals and cohomology of differential forms.
One of the main practical bottlenecks in using Feynman integrals to make high-precision predictions for particle experiments is the shear number of integrals to be performed. Because of this, it is more advantageous to first identify and compute a smaller set of master integrals (MIs) and then relate the remaining Feynman integrals to MIs with integration-by-parts (IBP) identities. The cohomology approach that the authors are reviewing is meant to put this on a solid mathematical ground as a "vector space of Feynman integrals" and directly compute the coefficient of the IBP expansion as intersection numbers. Recent progress indicates that this method has a potential applicability in deriving IBP relations, differential equations, etc. for Feynman integrals in practice. It also features connections to multiple fields of mathematics, including algebraic and differential geometry.
The authors wrote a comprehensive review of these ideas, which also features a lot of background material that will be very helpful for researchers wanting to understand this topic. This is a very mathematical subject and it's great to see the authors organized the review in a clear way. They often introduce an idea on simple examples coming from classic geometry and then explain the "twisted" version relevant for Feynman integrals. Parts of the review are also dedicated to explaining open problems and challenges. The manuscript is written in clear English.
Section 2 introduces Feynman integrals in parametric representation and formulates them in the language of twisted cohomology. Section 3 discusses twisted cohomology, counting the number of MIs and quadratic relations for Feynman integrals. Section 4 gives a brief introduction to the topics of cohomology, Morse theory and computing the topological Euler characteristic. Section 5 explains the connection between integrals and intersection theory in classic geometry followed by a generalization to twisted cohomology. Section 6 gives a background on simplicial homology, cup products, Poincare dualities and related mathematical topics. Section 7 discusses different ways for computing intersection numbers in practice with some examples. The appendices give more details about Baikov representation and Groebner bases.
I noticed that the arXiv submission of this manuscript has a comment: "arXiv admin note: text overlap with arXiv:2008.04823 by other authors." Before I can make a decision on this manuscript I will ask the authors to rewrite the parts of their review that caused this problem, so that there's no appearance of impropriety.
Reading along I caught several typos listed here:
- In the abstract: "on the light" should be "in the light"
- 5th paragraph of the introduction: "analiticity" should be "analyticity"
- Above eq. (10) "Mis" should be "MIs"
- Above eq. (12) "Pomeranski" should be "Pomeransky"
- In the paragraph between eqs. (46) and (47): "unit matix" should be "unit matrix"
- Above eq. (53): "spheric caps" should be "spherical caps"
- Last paragraph on p. 30: "pretence" should be "pretense"
- Below eq. (183): "Witney" should be "Whitney"
- Below eq. (216): "pure hodge structure" should be "pure Hodge structure"
- Last paragraph of sec. 7.4: "trasformations" should be "transformations"
- Below eq. (A5): "adress" should be "address"
Author Response
We thanks the referee for his/her comments. Indeed there was an overlap of 1.7% with arXiv:2008.04823. In order to fix it we performed the following changes:
-we have partially modified the text in lines 126–137 and 144–150 of the Introduction, and part of the text lines 208–222, 252, 253, 274–281 of section 3; -we also fixed all typos listed by the referee plus few more we have found checking the manuscript once again; We point out that we included further changes to the file in order to satisfy the requests of another referee: -we have replaced the short abstract with an extended version; -we added section 8 with an explicit example of application of intersection theory to Feynman integrals, in particular, to a massless double box diagram, making the relevant calculations very explicit step by step; Finally, we improved the Acknowledgements.Round 2
Reviewer 3 Report
The authors addressed all my comments in a satisfactory way. In addition, they expanded on parts of the text, including the new section 8 featuring intersection numbers computation for the massless double-box Feynman integral on the maximal cut, contributed by Gasparotto and Mandal.
In the present form, this review article will be a great addition to Universe and I strongly recommend it for publication.